# Effects of sea-level rise on tides and sediment dynamics in a Dutch tidal bay

Long Jiang[1,2], Theo Gerkema[2], Déborah Idier[3], Aimée B. A. Slangen[2], and Karline Soetaert[2]

[1] College of Oceanography, Hohai University, Nanjing, China
5 [2] NIOZ Royal Netherlands Institute for Sea Research, Department of Estuarine and Delta Systems, and Utrecht University, P.O. Box 140, 4400 AC Yerseke, The Netherlands.
[3] BRGM, 3, avenue C. Guillemin, 45060 Orléans cedex 2, France.

*Correspondence to*: Long Jiang (ljiang@hhu.edu.cn)

**Abstract.** Sea-level rise (SLR) not only increases the threat of coastal flooding, but also may change tidal regimes in 10 estuaries and coastal bays. To investigate such nearshore tidal responses to SLR, a hydrodynamic model of the European Shelf is downscaled to a model of a Dutch coastal bay (the Oosterschelde, i.e., Eastern Scheldt) and forced by SLR scenarios ranging from 0 to 2 m. This way, the effect of SLR on tidal dynamics in the adjacent North Sea is taken into account as well. The model setup does not include meteorological forcing, gravitational circulation, and changes in bottom topography. Our results indicate that SLR up to 2 m induces larger increases in tidal amplitude and stronger nonlinear tidal distortion in the 15 bay compared to the adjacent shelf sea. Under SLR up to 2 m, the basin shifts from a mixed flood- and ebb-dominant state to complete ebb-dominance. We also find that tidal asymmetry affects an important component of sediment transport. Considering sand bed-load transport only, the changed tidal asymmetry may lead to enhanced export. In this case study, we find that local impacts of SLR can be highly spatially-varying and nonlinear. The model coupling approach applied here is suggested as a useful tool for establishing local SLR projections in estuaries and coastal bays elsewhere. Future studies 20 should include how SLR changes the bed morphology as well as the feedback effect on tides.

## 1 Introduction

Sea-level rise (SLR) poses an increasing flood risk on global shorelines (FitzGerald et al., 2008; Haigh et al., 2014). In addition to the direct increment in water levels (Church et al., 2013; Oppenheimer et al., 2019), SLR induces changes in global and regional tidal regimes (Pelling et al., 2013b; Devlin et al., 2017; Pickering et al., 2017). Understanding these tidal 25 changes is an imperative step towards projecting future water levels and designing shoreline protection works (Katsman et al., 2011; Haasnoot et al., 2019). Shifts in estuarine and coastal tidal regimes including tidal amplitude, residual currents, and tidal asymmetry can potentially modify sediment transport, which further influences shoreline morphology and the accretion (or erosion) of salt marshes and tidal flats (van Goor et al., 2003; Chernetsky et al., 2010; Nnafie et al., 2014). Furthermore, in estuaries and embayments, changes in tidal mixing and currents are expected to have implications for salt intrusion, 30 nutrient transport, primary production, and other ecosystem functions (Nienhuis and Smaal, 1994; Zhang et al., 2010;

Winterwerp et al., 2013; de Jonge et al., 2014). Therefore, in order to understand and mitigate the SLR hazards in estuaries and coastal bays, it is of high priority to study the SLR impact on tides.

SLR-induced tidal changes in estuaries and embayments are more complex than in the open ocean and shelf seas (Holleman and Stacey, 2014). Tidal waves propagating in nearshore regions are strongly deformed via factors like shoaling, damping, and reflection (resonance). As these processes respond to SLR to various extents, this triggers spatially heterogeneous modifications in tidal regimes (e.g., Pickering et al., 2012; Carless et al., 2016). Therefore, tidal responses to SLR vary among and within systems (Holleman and Stacey, 2014; Pelling and Green, 2014). For example, with SLR, the tidal amplitude may increase due to reduced friction (Arns et al., 2015; Idier et al., 2017) or decrease as a consequence of enhanced dissipation in the newly inundated areas (Pelling et al., 2013a; Ross et al., 2017). Overtides and tidal asymmetry can also be modulated distinctly among estuaries, with ramifications for residual sediment transport and morphodynamic development (Hoitink et al., 2003; van der Wegen, 2013; Gräwe et al., 2014; Dijkstra et al., 2019a). In estuaries and tidal bays, the direction and extent of tidal asymmetry depend on the tidal amplitude ($a$) to basin depth ($h$) ratio, $a / h$, as well as the volume ratio of intertidal regions ($V_s$) to channels ($V_c$), $V_s / V_c$ (Friedrichs and Aubrey, 1988). Large $a / h$ and $V_s / V_c$ ratios point to high tendency of flood and ebb dominance, respectively (Friedrichs and Aubrey, 1988). While the $V_s / V_c$ ratio decreases with SLR, how $a / h$ is influenced by SLR is less straightforward, as it depends on the specifics of the basin in question (Brown and Davies, 2010). Thus, given the various responses in different types of basins, tidal impacts of SLR should be assessed on a system- or site-specific basis.

Changes in tidal amplitude in coastal oceans and estuaries can be caused by anthropogenic activities (e.g, dredging, land reclamation) and climate change (Haigh et al., 2019; Talke and Jay, 2020). The impacts of SLR on estuarine and riverine tides are analogous to those of channel deepening owing to dredging and sand mining, as both are associated with increased water depth (Talke and Jay, 2020). Winterwerp and Wang (2013) found in a theoretical study that narrowing and deepening of small convergent estuaries tends to increase tidal amplitude and flood dominance resulting in a hyper-turbid state. The deepening-induced stronger tidal amplitude is reported in the Ems (de Jonge et al., 2014), Hudson (Ralston et al., 2019), and Loire Rivers (Winterwerp et al., 2013), and the Cape Fear River Estuary (Familkhalili and Talke, 2016), especially in the upper reach. Tidal responses to deepening also show intra- and inter-system variability due to the various deepening extents and characteristics of estuaries (Talke and Jay, 2020). For instance, the partially dredged Newark Bay exhibits spatially variable changes in tidal range (Chant et al., 2018). While the retrospective analyses on deepening-induced tidal alterations have implications for potential shifts under SLR circumstances, it is noteworthy that they are not the same. Dredging activities are mainly conducted at tidal channels and spatially nonuniform, whereas SLR occurs throughout the estuary, as well as in the adjacent coastal seas (Ralston et al., 2019).

Although challenging, tidal changes due to SLR have been modeled in both idealized and realistic estuaries and bays (e.g., Hong and Shen, 2012; Pelling et al., 2013b; Holleman and Stacey, 2014; Ensing et al., 2015; Passeri et al., 2016; Du et al., 2018). Generally, studies with idealized (simplified) basin shapes cannot fully address the spatially nonuniform shoaling, reflection, and damping in realistic systems. Furthermore, in many modeling studies, SLR is prescribed by simply

increasing the surface elevation at the open boundary, without taking into account the tidal changes in the adjacent shelf seas itself, which would radiate into the estuaries and bays (e.g., Hong and Shen, 2012; Holleman and Stacey, 2014; Ross et al., 2017). In fact, before entering estuaries, tidal waves on the shelf can be significantly modified in amplitude and phase or distorted (Jay et al., 2010; Idier et al., 2017). Chernetsky et al. (2010) found that the externally generated overtides play a key role in the sediment dynamics within an estuary. Additionally, the tidal changes in estuaries and shelf seas may exert a strong influence on tides in the ocean (e.g., Ray et al., 2006; Arbic and Garrett, 2010). This underlines the necessity of coupling the coastal and shelf processes with the dynamics of land-locked water bodies for SLR assessments (e,g., Zhao et al., 2014; Rasquin et al., 2019).

Here, we present a case study with regional SLR projections driving a hydrodynamic model of the European Shelf, which was then downscaled for the Oosterschelde (English name: Eastern Scheldt), a Dutch tidal bay adjacent to the North Sea (Fig. 1). We investigate the SLR impacts on the local tidal dynamics and the implications for residual sediment transport. This study also explores the necessity of the model coupling method when predicting SLR influences in other estuaries and coastal bays.

## 2 The study site

The Oosterschelde is a well-mixed tidal bay on the southwestern coast of the Netherlands (Fig. 1). Because of the limited freshwater input, the semidiurnal-dominant tides control the water renewal and material transport in the Oosterschelde (Jiang et al., 2019). A storm surge barrier (hereafter referred to as "barrier", Fig. 1) constructed at its mouth in the late 1980s reduced the tidal prism and amplitude by ~30% and 13%, respectively. As a result, the tidal flat and salt marsh areas and landward supply of sediments, nutrients, seston, and chlorophyll sharply declined, and the residence time was doubled (Nienhuis and Smaal, 1994). Due to a decreased sediment source and reduced tidal range, erosion of tidal flats is ongoing, disturbing ecosystem services in the bay (Vroon, 1994). Since the construction of the barrier, tides have not been substantially changed by changing bed morphology of the basin (de Pater, 2012).

In surrounding systems such as the Western Scheldt estuary and Rotterdam waterway, south and north to the Oosterschelde respectively, tides and sediment dynamics have been changing over the last century as a result of sand mining, dredging, and modifications of shorelines and navigation channels (Winterwerp et al., 2013; van Rijn et al., 2018; Cox et al., 2019; Dijkstra et al., 2019a). The tidal range in the southern North Sea has increased since the 1950s partially due to the engineering works on the Dutch Delta (Hollebrandse, 2005).

In the future, SLR at the Dutch coast may exceed 1 m between 2000 and 2100 under a high greenhouse gas emission scenario (Vermeersen et al., 2018). Shelf-study models indicate that, with SLR, the $M_2$ amphidromic point in the southern North Sea moves northeastwards, further away from the Dutch Delta but closer to the Wadden Sea, inducing a decreased (increased) $M_2$ amplitude on the northern (southern) Dutch coast (Pickering et al., 2012; Idier et al., 2017). The overall tidal amplitude (mostly semidiurnal components) adjacent to the Dutch Delta are projected to increase mainly due to

reduced friction and amphidrome movement (Pickering et al., 2012; Pelling et al., 2013a; Idier et al., 2017). As a result of the increased water depths on the European shelf, tidal phase speed is increased, leading to earlier arrival of semidiurnal tidal waves, i.e., reduced semidiurnal phases, in the southern North Sea (Idier et al., 2017). The SLR-induced increased tidal amplitude may reverse some of the above post-barrier declining trends, highlighting the need of exploring the local tidal

responses to SLR in the Oosterschelde.

## 3 Methods

We used the hydrodynamic models MARS (Model for Applications at Regional Scale, Lazure and Dumas, 2008) to model tides on the European Shelf at a horizontal resolution of 2 km, and, in a downscaling setup, GETM (General Estuarine Transport Model, www.getm.eu) to model tides in the Oosterschelde at a resolution of 300 m (Fig. 1). Our study applies a

one-way nesting technique that accounts for the communication from the larger (MARS) to smaller (GETM) domain, but not the other way. The description and setup of these two models were detailed by Idier et al. (2017) and Jiang et al. (2019), respectively. This section therefore focuses on the model coupling setup, SLR scenarios, and model calibration.

MARS was forced with all tidal components from the global tide model FES2004 (Lyard et al., 2006), i.e., Mf, Mm, Msqm, Mtm, O1, P1, Q1, K1, M2, K2, 2N2, N2, S2, and M4. The MARS domain (Fig. 1) covers the entire North-West

European continental shelf and extends to deep waters (> 200 m) so that the SLR-induced changes in tidal components at its open boundary is minimal. The year 2009 was used as the baseline scenario, and the observed water elevation and tidal components from 16 tide gauges were well reproduced by MARS for that year (Idier et al., 2017).

The SLR scenarios were implemented in MARS by uniformly increasing the open-boundary water level in the baseline scenario. Using the water level data from a regional SLR projection model (Slangen et al., 2014), Idier et al. (2017)

also tested a scenario with nonuniform open-boundary SLR and found insignificant differences of tides from the uniform scenarios in the southern North Sea. Thus, only the MARS results with uniform SLR scenarios were used for downscaling to the Oosterschelde model. The SLR scenarios used for the MARS model are 0.25 m, 0.5 m, 0.75 m, 1.0 m, 1.5 m, and 2.0 m. The regional projection at Vlissingen by Slangen et al. (2014) was used to estimate the SLR time scale in the Oosterschelde, which is included in some figures as an indication (the SLR rate does not feature in the model runs itself). In the MARS

domain, most low-lying land within 2 m height above the present sea level is located on the eastern shore of the North Sea, i.e., Belgium, the Netherlands, Germany, and Denmark (www.flood.firetree.net/), where the coastlines are well protected from flooding (Pelling and Green, 2014). Hence, the SLR scenarios were conducted without flooding of these shorelines in both MARS and GETM. Given the uncertainties of bottom topography in the future, a constant bathymetry was used in the baseline and SLR scenarios. Atmospheric forcing or baroclinic effects were not included in the model setup.

GETM was used to downscale MARS to the Oosterschelde in a one-way coupling and 2D barotropic mode. The barrier consists of two artificial islands and three tidal openings (Fig. 1), and each opening is facilitated by concrete pillars and steel gates that can be closed under severely stormy conditions. While our model resolution is incapable of representing

each pillar (< 4 m wide), the cross-sectional area of pillars, ~8.2% of the overall area, was compensated by reducing the depth by the same percentage at the "barrier" cells in the model. Given the uncertainties in future bed morphology and bottom roughness, we applied a spatially constant bottom roughness length scale of 1.7 mm in the baseline and all SLR scenarios as used in the Wadden Sea (Duran-Matute et al., 2014). Wetting/drying of tidal flats in the Oosterschelde was solved in GETM, while the shoreline (white areas in the GETM domain, Fig. 1) stayed unflooded. The Flather open boundary (Flather, 1988) was applied in GETM, in which the gravity-wave radiation condition requires prescribing both water elevation and current velocity as boundary forcing. In the baseline and SLR scenarios, these two variables in the vicinity of the GETM open boundary (Fig. 1) were extracted from MARS output every 15 min and linearly interpolated to each GETM open boundary node (255 nodes totally, Jiang et al., 2019). Every scenario was run for one year (2009) and the time series of modeled tidal elevation and currents were decomposed with the T_TIDE toolbox (Pawlowicz et al., 2002) to extract the major components for all grid cells excluding the tidal flats.

Observational water elevation data at three stations in the east, middle, and west of the Oosterschelde (Fig. 1) were obtained from the Dutch government agency Rijkswaterstaat website (www.rijkswaterstaat.nl/water), and compared with the sea surface height in the GETM baseline output. Note that the observed water elevation is influenced by tides as well as winds and gravitational circulation, while the model is only tide-forced. The correlation coefficients, standard deviations, and root mean square differences (RMSDs) between modeled and observed water level were calculated to plot a Taylor Diagram (Taylor, 2001). Harmonic analyses were conducted to both observed and simulated water elevation to compare the magnitude and phase of the resultant M2 and M4 components. The absolute error (simulation - observation) and relative error (absolute error / observation) were used to denote the fit for the phase and magnitude, respectively (Fig. 2).

Tidal asymmetry was estimated using the phase difference between the M2 and M4 currents following Friedrichs and Aubrey (1988): a positive, zero, and negative value of $\cos(2\phi_{UM2} - \phi_{UM4})$ indicates flood-dominance, symmetric tide, and ebb-dominance, respectively, where $\phi_{UM2}$ and $\phi_{UM4}$ are the phases of M2 and M4 current velocity, respectively. Changing tidal asymmetry may exert an alteration in the sediment budget of a system. Sediment transport is a complex function of sediment properties, tidal dynamics, gravitational circulation, and other processes (e.g., van der Wegen et al., 2011; Dijkstra et al., 2017; Schulz and Gerkema, 2018) and would require solving the basin-wide dynamic sediment transport of all grainsizes and geomorphological changes in the current and future conditions. Here we aim to gain insight into the potential implications for sediment transport due to changes in tidal asymmetry alone, i.e., by focussing on a component of sediment transport. Specifically, we used the analytical quantity $Q$ (kg m$^{-1}$ s$^{-1}$, Eq. (1)) proposed by Gräwe et al. (2014), which applies to systems with relatively weak stratification and estuarine circulation, such as the Oosterschelde (Burchard et al., 2013).

$$Q = \frac{3\alpha\kappa_v}{4w_s^2} U_{M2}^2 U_{M4} \cos(2\phi_{UM2} - \phi_{UM4}) \tag{1}$$

In Eq. (1), $\alpha$ is the sediment erosion parameter (kg s m$^{-4}$), $\kappa_v$ is the vertical diffusion coefficient (m$^2$ s$^{-1}$), $w_s$ is the settling velocity (m s$^{-1}$), and $U_{M2}$ and $U_{M4}$ are the magnitude of the M2 and M4 current velocity (m s$^{-1}$), respectively. Because most

of the post-barrier sediment in the Oosterschelde is sandy (Mulder and Louters, 1994), we focus on a typical kind of sand with values of $\alpha$ (0.001) and $w_s$ (0.001) as suggested by Burchard et al. (2013) and Gräwe et al. (2014) for weakly stratified estuaries and the North Sea. $Q$ represents sand bedload transport as it assumes a negligible settling time lag between the local suspended sediment concentration and currents. $U$, $\kappa_v$, and $\phi$ were calculated from different SLR scenarios. The analysis based on $Q$ saves substantial computational time compared to applying a sophisticated sediment transport model.

## 4 Results

### 4.1 The baseline scenario

Our modeled water elevation in the baseline scenario (SLR = 0) matches well with the observed water level at tide gauges with the overall correlation coefficients > 0.95 and RMSDs (dimensionless) < 0.1 (Fig. 2). For example, Figs. 2a and 2b shows good agreement between simulated and observed water elevation during days 175-185, a period with relatively weak winds (this period was chosen as the coupled models were run without atmospheric forcing). In addition to the water level, the model captures the observed magnitude and phase of M2 and its major overtide M4 with relatively small errors (Fig. 3a). The errors of observed and modelled M2-M4 phase difference ($2\phi_{M2} - \phi_{M4}$) are -6.2º, 13.5º and 21.3º at Roompot binnen, Stavenisse, and Bergse Diepsluis west, respectively (Fig. 3b), which may result from the meteorological forcing and gravitational circulation that is lacking in the model. The model overestimates the extent of flood and ebb dominance, but the direction of tidal asymmetry agrees with the observation (Fig. 3b). The spatial asymmetry of vertical tides, i.e., flood dominance in the west and ebb dominance in the middle and east, also matches the previous results with the 2008 observational data (de Pater, 2012).

The tidal range (TR) increases from 2.5 m to 3.4 m from the mouth to head (Fig. 4a). As the dominant component in the Oosterschelde, the M2 magnitude captures most of the TR spatial pattern and it takes ~30 min (phase difference ~15°) for the semidiurnal tidal wave to propagate from the western to eastern side (Fig. 5c). With the basin length $L = 40.8$ km, the average M2 phase speed of the basin ($c_{M2}$) is approximately 22.7 m/s. This phase speed is much faster than the shallow water wave speed in inviscid and frictionless systems, $c_0 = \sqrt{gh} = 8.3$ m/s, where $h$ is the average depth of the bay, 7.0 m. Based on our field measurements and model calculation, the phase difference between the horizontal and vertical tides is close to 90º, indicating that the tidal waves in the basin are nearly standing waves. The landwards increasing tidal amplitude, the amplified phase speed compared to $c_0$, and the properties of nearly standing waves are all important features of a convergent system rather than a frictional damping system (Hunt, 1964; Jay, 1991; Friedrichs and Aubrey, 1994; Lanzoni and Seminara, 1998; Savenije and Veling, 2005; van Rijn, 2011). The convergence in cross-sectional area is a result of narrowing in the west and shoaling (increased areas of tidal flats) in the east (Figs. 1 and 6). The sharply reduced water depth caused by the barrier at the mouth (Fig. 1) significantly weakens the tidal amplitude and delays the propagation of tidal waves (Figs. 4 and

5). This discontinuity in the tidal amplitude and phase between the North Sea and Oosterschelde is consistent with post-barrier observations (Vroon, 1994).

Fig. 7a shows the M2-M4 phase difference of the horizontal tides (current velocity) in the baseline scenario, which is consistent with that calculated from the vertical tides. That is, inside the bay, the western and eastern parts show flood and ebb dominance, respectively (Fig. 7a). The barrier marks a boundary of changed tidal asymmetry, separating the ebb and flood dominance outside and inside, respectively (Fig. 7a).

## 4.2 SLR effects on the tidal amplitude and phase

In the absence of sediment deposition and erosion, the tidal flats will be gradually inundated by the increasing sea level at an accelerated pace (Fig. 8a). With SLR, TR increases within the Oosterschelde, especially at the landward ends (e.g., Fig. 4b). The average TR over the entire bay increases nearly proportionally to SLR according to the linear regression $TR = 0.337 * SLR + 2.93$ ($r^2 > 0.995$), all in metres, indicating a 11.5% increase in TR per metre SLR (Fig. 8a). Our study considers SLR up to 2 m, in which range the above relationship and other findings in Sections 4.2 and 4.3 apply. It should be noted that $r^2$ only denotes the goodness of fit of the linear model. With the assumptions made in the SLR scenarios and simplifications of our 2D barotropic model, the uncertainty level of the regression is at least 10% given the 0.1 RMSDs between the modeled and observed water elevation (Fig. 2c). In the pre-barrier period (1900–1980), a 25 cm SLR increased the tidal amplitude by 3%–4% in the Oosterschelde (Vroon, 1994), which is comparable to our estimated rate (11.5% per meter SLR). This rate is much faster than that in the adjacent North Sea in the GETM domain, where $TR = 0.0544 * SLR + 3.12$ ($r^2 > 0.995$, Fig. 8a) as well as the entire southern North Sea (Idier et al., 2017). In the 1980s, TR declined by ~0.35 m due to the construction of the barrier, so that, based on our estimation, the pre-barrier magnitude will be restored at around 1 m SLR, which may occur by the end of the 21st century (Fig. 8a).

The increase of M2, S2, and M4 amplitudes in the bay is also proportional to SLR, with a slope of 0.157, 0.056, and 0.029 m/m SLR ($r^2 > 0.995$). These increasing magnitudes per metre SLR account for 11.3%, 15.7%, and 40.4% of the M2, S2, and M4 magnitude in the baseline scenario, respectively. The spatial patterns of the semidiurnal components M2 and S2 are similar to that of the overall TR, and accordingly they are more sensitive to SLR inside the bay compared to outside (e.g., Fig. 5b). In contrast, under SLR the M4 amplitude decreases outside, while it increases inside the Oosterschelde (Fig. 5f). We conducted a hypothetical model run, in which tides in the 1 m SLR scenario of the MARS model were prescribed to the GETM open boundary, while the sea level stayed the same as the baseline run. Results in this run indicate that the M4 amplitude decreases out of the bay similarly as shown in Fig 5f and hardly changes in the bay. Thus, the decrease in M4 amplitude out of the bay results mainly from the MARS domain, whereas the M4 amplitude in the basin is largely modulated by the changing sea level. Compared to the baseline scenario, the M4 amplitude in the bay increases (40.4% per metre SLR) much faster than TR and the main semidiurnal components.

As a result of the movement of the amphidromic point and reduced friction (Idier et al., 2017), the M2 tide in the southern North Sea arrives earlier under SLR (Fig. 5d). The convergent property of the Oosterschelde (Section 4.1) amplifies

the vertical and horizontal M2 tide (Figs. 5b and 5d). Consequently, the phases of semidiurnal components decrease with SLR, with a larger change in the bay than in the coastal sea and at the landward than at the seaward ends (e.g., Fig. 5d). This can explain the accelerated tidal currents under SLR (Fig. 9). In contrast, the M4 phase varies in a nonuniform and non-linear way, implying strong modifications of shallow-water tides under SLR (Fig. 5h).

### 4.3 SLR-induced shifts in tidal asymmetry and implications for sediment transport

The different responses of M2 and M4 under SLR conditions change the tidal asymmetry in the Oosterschelde. The entire bay shifts from a mixed ebb- and flood-dominance (Fig. 7a) to an increasingly ebb-dominant state (Fig. 7b). The switch of the entire bay already occurs at SLR below 0.25 m, as indicated by the M2-M4 phase difference (Fig. 8b). In the scenario with 1 m SLR, the flood-dominant western part becomes ebb-dominant, while the ebb-dominance in the eastern part is enhanced; in contrast, the M2-M4 phase difference in the adjacent North Sea is relatively insensitive to SLR (Fig. 7b).

The quantity $Q$, an indicator of sand bedload transport, is used to estimate the combined effects of tidal current velocity and asymmetry. With both tidal currents (Fig. 9b) and ebb-dominance (Fig. 7b) strengthened by SLR, the absolute value of $Q$ is amplified, and the direction becomes completely seaward (Fig. 7d). The increasing seaward $Q$ under SLR (Fig. 7d) is mainly through channels where tidal currents are strong (Figs. 7d and 9a). Tidal asymmetry is one important mechanism of sediment transport among others (density-driven transport, lateral transport, etc.). While $Q$ in this study cannot be used for quantifying the full sediment budget of the basin, it sheds light on possible transitions in sediment transport directions. In the pre-barrier period, the sand erosion and sedimentation of tidal flats were in equilibrium in the Oosterschelde (Mulder and Louters, 1994). The barrier acts as an obstacle to sand import, while the reduced tidal currents cannot resuspend and supply sufficient sand to the eroded tidal flats, creating a sand deficit for tidal flats (Eelkema et al., 2012). Our results imply that despite the fact that SLR can restore the pre-barrier TR (Section 4.2), the increased ebb dominance may have an adverse impact on sand import. Note, however, the fate of tidal flats remains uncertain given the unaddressed mud transport in this study and many details such as the shape of tidal flats and wave action (van der Wegen et al., 2017).

## 5 Discussion

This case study shows how future SLR may change the tidal regime and residual sediment transport in a tidal bay. Understanding tidal responses to SLR is fundamental to anticipating any ecosystem shifts and adjusting management strategies in global estuaries and bays. For instance, a 1 m SLR can increase the mean high water level in the Oosterschelde by 0.16 m, about half of the increase in tidal range (Fig. 8a). However, in this scenario, the increase in high water at spring tides can be up to 30 cm, which implies that at least an extra 30 cm needs to be accounted for in dike construction in addition to the 1 m SLR, as similarly reported for other coastal systems (Arns et al. 2015; Devlin et al., 2017). As TR increases under SLR, the turnover time, an indicator of water renewal efficiency, is significantly shortened in most parts of the basin (Fig.

10). This will put the system under a greater influence of the North Sea, likely increasing import of nutrients and organic matter and affecting the carrying capacity of shellfish culture (Jiang et al., 2019). In many other nearshore systems, tidal changes as a result of SLR were also found to alter the ecosystem functions (Bhuiyan e al., 2012; Hong and Shen, 2012).

One intriguing finding in our study is a much stronger response of tidal range to SLR in the bay compared to the adjacent coastal sea (Fig. 8a). This is not a consequence of enhanced resonance. In strongly dissipative basins, SLR reduces the phase speed and the ratio of basin length to wavelength and increases the resonance-induced amplification of tidal magnitude (Talke and Jay, 2020). In our study, if the classical phase speed $c_0$ is applied, the quarter-wavelength resonance period of our system ($T = 4L/\sqrt{gh}$, see e.g., Gerkema, 2019) in the baseline scenario is 5.5 hours. As a convergent basin, when the average M2 phase speed $c_{M2} = 22.7$ m/s (Section 4.1) is used, the quarter-wavelength resonance period $T = 4L / c_{M2}$ should be 2.0 hours in the Oosterschelde. The Helmholtz resonance, which assumes a uniform tidal phase in a tidal bay, does not apply to the Oosterschelde because of the along-channel semidiurnal phase difference (~15º, 30 min, Fig. 5c). Moreover, the Helmholtz resonance period ($T_H = 2\pi\sqrt{AL_b/(gBH)}$, where $A$ is the basin surface area and $L_b$, $B$, and $H$ are length, width, and depth of the channel connecting the bay and North Sea; e.g., Sutherland et al, 2005; Gerkema, 2019) for the Oosterschelde is no longer than 2.4 hours. Both $T$ and $T_H$ are much shorter than the semidiurnal tidal period. With h, $c_{M2}$, and $H$ increased with SLR and $L$, $L_b$, and $B$ unchanged, $T$ and $T_H$ will decrease and shift further away from semidiurnal resonance. With the resonance impacts ruled out, the increased TR due to SLR in the Oosterschelde can result from reduced friction and the amplifying nature of a convergent basin. In a dissipative basin, SLR and channel deepening can reduce the bottom friction and increase tidal amplitude (e.g., Winterwerp et al., 2013; de Jonge et al., 2014). With the declining friction, the system can even shift from a damping into amplifying basin, such as the Hudson River (Ralston et al., 2019). In a convergent basin, the responses of tidal amplitude to SLR is always stronger at the landward than the seaward ends, as found in our study (Figs. 5b and 5f) and by Ensing et al. (2015). Changing inlet cross-sectional area due to SLR, usually by inlet widening, can also alter tides in the bay due to changed choking effect (Passieri et al. 2016; Talke and Jay, 2020). In our case, the bathymetry and anthropogenic islands at the storm surge barrier are resistant to flooding so that the choking effect is not modified by SLR.

Another major finding of this study is the SLR-induced changes in tidal asymmetry with potential effects on sediment transport. Tidal distortion and asymmetry result from interactions between basin geometry and shallow-water tidal waves (Speer and Aubrey, 1985). Shallow waters with a high $a / h$ ratio are usually flood-dominant because tides propagate faster during high water than low water, while an extensive intertidal area flanking deep channels can slow down flood propagation and generate ebb-dominance (Friedrichs and Aubrey, 1988). With SLR, the water depth increases, and tidal flat area diminishes. These two processes render the system to a less flood-dominant and less ebb-dominant state, respectively. The shift of tidal asymmetry depends on these two competing effects (Friedrichs et al., 1990). The SLR-induced shift to ebb dominance in the Oosterschelde implies a stronger reduction in $a / h$ than $V_s / V_c$. In a strongly convergent and less strongly dissipative basin, the overtide magnitude increases landwards (Lanzoni and Seminara, 1998), such as in the Oosterschelde

(Fig. 5e). When the relative strength of convergence to dissipation increases, such systems will become more distorted (i.e., M4 is amplified more than M2) and ebb-dominant (Lanzoni and Seminara, 1998). Under SLR, the basin geometry in our study (Fig. 6) does not change while friction is reduced due to increased water depth; i.e., friction becomes weaker relative to convergence. The SLR-induced phenomena that M4 increases faster than M2 (Section 4.2) and that the ebb dominance is enhanced (Fig. 7b) are consistent with findings by Lanzoni and Seminara (1998). In contrast, the friction-dominated Ems estuary acquires a stronger flood-dominant signal with increasing water level (Chernetsky et al., 2010; Winterwerp et al., 2013), whereas tidal asymmetry is insensitive to SLR in the Ria de Aveiro lagoon (Lopes and Dias, 2015) and to deepening in the Western Scheldt (Winterwerp et al., 2013). Both suspended and bedload sediment transport are strongly associated with tidal asymmetry, especially in weakly stratified estuaries, and are important to the long-term basin geomorphology (van Maren et al., 2004; Burchard et al., 2013). For example, with the deepening of the Ems estuary, the increased flood dominance promotes the import of suspended sediment, causing a hyper-turbid state in the estuary (Talke et al., 2009; Winterwerp and Wang, 2013; Dijkstra et al., 2019b). In our study, the shift to ebb dominance changes an important component of sediment transport $Q$ (Fig. 7d), which may be detrimental to shoreline protection and salt marsh accretion.

Although our results are not generic to global estuaries and bays, this study shows the complicated interaction between basin geometry and tides and pinpoints the urgency of understanding estuarine tidal responses under changing SLR conditions. To this end, our case study highlights the following aspects to be fully considered in future studies.

Firstly, tidal responses to SLR can vary from system to system, and comparative studies (e.g., Passeri et al., 2016) are much needed. Because of the spatially varying coastline and bathymetry, shallow-water tides react to friction, reflection, and cross-sectional convergence to different extents among and within systems under SLR (e.g., Fig. 8a; Carless et al., 2016; Idier et al., 2017). Thus, compared to shelf-sea models, estuarine models with refined spatial resolution are required to capture the detailed features of bathymetry and coastlines, and hence nearshore tidal distortion.

Secondly, in addition to water height, SLR-induced tidal variations in the shelf seas have significant impacts on estuarine/embayment dynamics. Tides in shelf seas exhibit nonlinear and nonuniform responses to SLR (Idier et al., 2017; Pickering et al., 2012), and these effects may penetrate or amplify in estuaries and bays (Ensing et al., 2015). Nevertheless, many previous studies into SLR effects on estuaries simply increased the water level, neglecting changes in the tidal characteristics at their coastal boundary (e.g., Hong and Shen, 2012; Holleman and Stacey, 2014). Based on our results of a scenario increasing the open-boundary water height only, such a simplification overestimates (underestimates) the TR in the Oosterschelde (North Sea) by 9 (4) cm for 1 m SLR and completely misses the reduction of tidal phases. Without variations in externally generated M4 tides, calculation of estuarine tidal asymmetry and sediment transport can produce different results (Chernetsky et al., 2010). Therefore, given the relatively coarse resolution of shelf-sea models in estuaries and the inability of estuarine models to resolve the shelf-sea tidal variations due to SLR, model coupling is essential in examining nearshore SLR impacts. In addition, the tidal changes in estuaries and embayments may exert an influence on tides in the shelf seas (Ray et al., 2006). Note however that the feedback from the Oosterschelde to the North Sea is not simulated in our model coupling, which makes an interesting future study.

Thirdly, nearshore studies should be combined with regional SLR projections to steer efficient shoreline management strategies. Despite large uncertainties associated with emission scenarios and sea-level contributions (Slangen et al., 2014), regional SLR projections provide specific timelines for tidal changes and set the time window for required management actions in the coming decades. For example, the shift to ebb-dominance will likely occur before 2050, while nourishment of intertidal flats will be continuously needed well beyond 2100 (Fig. 8). Shoreline defense against SLR ranges from hard measures that prevent flooding, such as levees and dikes, as in the Netherlands, to soft measures which allow flooding, such as using marshes and newly inundated areas for dissipating tidal waves, for instance in parts of the Chesapeake (Lee et al., 2017) and San Francisco Bay (Holleman and Stacey, 2014). Including hard or soft coastal defense measures in numerical models can significantly affect the sensitivity of tides to SLR (Ensing et al., 2015; Idier et al., 2017; Ross et al., 2017). In our model setup, intertidal flats around channels were allowed to drown, while the shorelines with dikes were not (Fig. 1) so that the reduction of cross-sectional area in Fig. 6 does not vary much with SLR. In a convergent system as the Oosterschelde, if shoreline flooding is fully or partially allowed, the spatial convergence may change accordingly. Clearly, the projections resulting from our study might greatly change if the actual shoreline defense were to be implemented differently.

It should be noted that our coupled model does not account for winds and the gravitational circulation. According to recent studies, wind climate can contribute significantly to long-term variability of regional water elevation (Arns et al., 2015; Gerkema and Duran-Matute, 2017). Density-driven flow can also dominate local transport processes (Geyer and MacCready, 2014; Burchard et al., 2018; Schulz and Gerkema, 2018). Another limitation of the study is not considering changes in bed morphology and thus, bottom roughness. The field measurements on the effect of bed morphology on bottom roughness and shear stress (e.g., Cheng et al., 1999; Prandle, 2004) will improve the simulation and projection of tidal currents. While tides in the Oosterschelde were not strongly affected by bathymetric changes in the past decades (de Pater, 2012), it would be interesting to investigate the interaction between SLR and basin morphology by implementing a geomorphology component into our model. With such a component, sediment of all grain sizes can be considered and the influence of bed morphology changes on tides can be addressed. In addition to the natural development of bed morphology, future anthropogenic activities such as dredging may change the regional bathymetry in estuaries and bays (Ralston et al., 2019). Despite the similarities to SLR processes, dredging activities involve deepening of only the tidal channel and likely changes in bottom slopes, and therefore the tidal responses can differ (Ensing et al., 2015).

**6 Conclusions**

This study applies a one-way model nesting approach to investigate how SLR affects tidal range and asymmetry in the Oosterschelde, a coastal bay located in the Southwest Dutch Delta. The local model domain comprising the Oosterschelde and part of the adjacent North Sea imports water elevation from a European Shelf model (Idier et al., 2017) in the year 2009 and under SLR ranging from 0 to 2 m, which may possibly occur in this and the next century. There is no

feedback of changes from the Oosterschelde model to the shelf model. Neither model accounts for the meteorological forcing, baroclinic effects, and changes in bottom topography.

Under these assumptions, the "static" (no erosion and accretion) tidal flats are submerged under up to 2 m SLR. As a convergent basin, the Oosterschelde exhibits greater increases in tidal range and amplitude of semidiurnal and quaterdiurnal components than the North Sea under SLR up to 2 m. In the SLR scenarios applied, the tidal current is accelerated as indicated by a reduced M2 phase difference between the seaward and landward ends. Tidal asymmetry in the bay defined by the M2 and M4 phase difference shifts from a mixed flood- and ebb-dominant to ebb-dominant state under SLR below 0.25 m. Using a simplified approach, we showed the impact of tidal asymmetry and current velocity on one component of sediment transport, considering the dominant sediment type, sand. Our results show that the SLR-induced shift to ebb-dominance also changes the direction of sand bedload transport, favoring export under SLR.

Despite the simplifications and assumptions in our study, the findings indicate strong potential SLR-driven effects on the Oosterschelde ecosystem. With negligible freshwater input, the physical transport in the bay is greatly dominated by tides. SLR may increase the tidal amplitude to exceed that in the pre-barrier decades. While the fate of tidal flats needs intensive further studies since fine sediments are not addressed here, our study highlights the spatially variable features of tidal changes under SLR. Comparative studies among basins and sites are desired, and a one-way or two-way model coupling approach seems appropriate in such applications.

**Code and data availability**

The source code for the MARS and GETM model used in this study are available at https://wwz.ifremer.fr/mars3d/ and https://getm.eu/. Research data is archived on 4TU.Research Data (https://data.4tu.nl/) with the doi https://doi.org/10.4121/uuid:c6753aa0-d501-4cbe-9476-a2833d47bfc6.

**Author contributions**

LJ ran the GETM simulations, analyzed the results, and initiated the writing of the manuscript. TG and KS provided guidance and important insights into data interpretation. DI ran the MARS simulation. ABAS conducted the regional SLR projection. All authors participated in the writing and editing of the manuscript.

**Competing interests.**

No competing interests are present.

## Acknowledgments

This work was supported by the collaborative framework of Utrecht University and Royal Netherlands Institute for Sea Research.

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

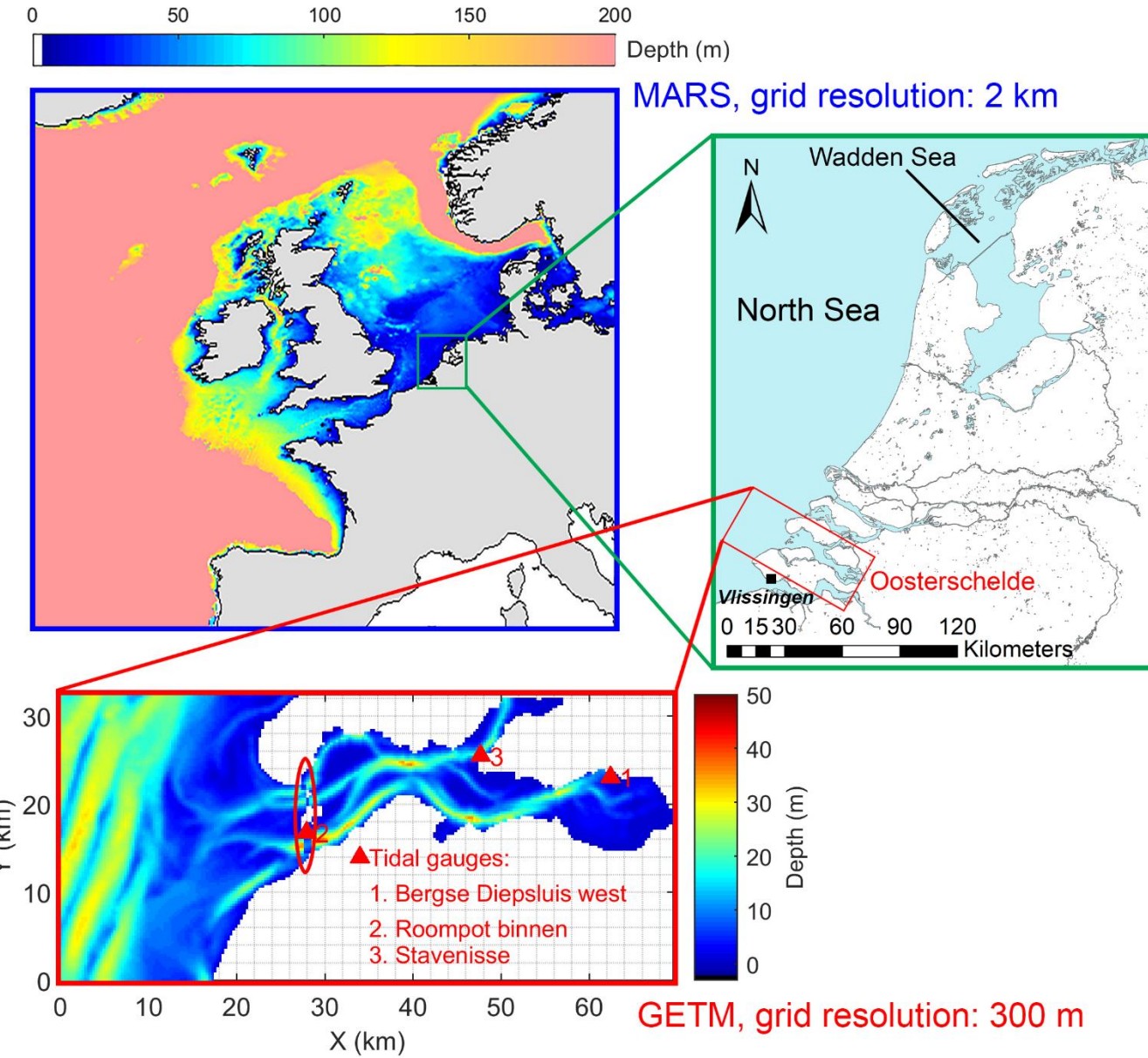

Figure 1: The study area and domains of the two hydrodynamic models representing the European Shelf (blue) and Oosterschelde (red). The green box shows locations of the Oosterschelde and local SLR projection, Vlissingen, in the zoom-in map of the Netherlands. In the Oosterschelde domain, tide gauges used for model calibration and the location of the storm surge barrier are marked with triangles and an ellipse, respectively; white areas are land segments protected by dikes where flooding is not allowed.

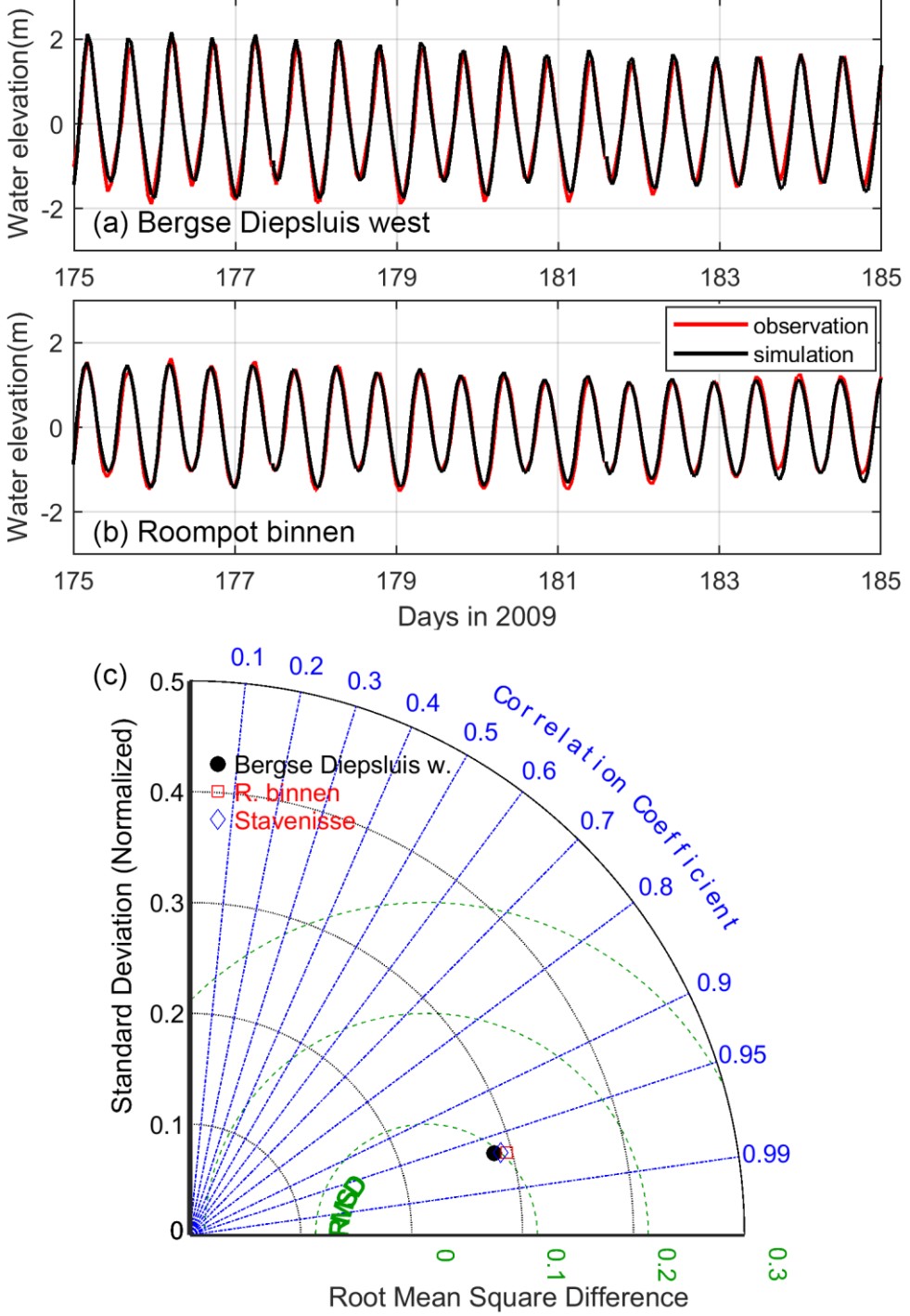

**Figure 2: (a and b) Time series of modeled and observed water elevation and (c) the Taylor Diagram of the water elevation comparison. See Fig. 1 for locations of data sites.**

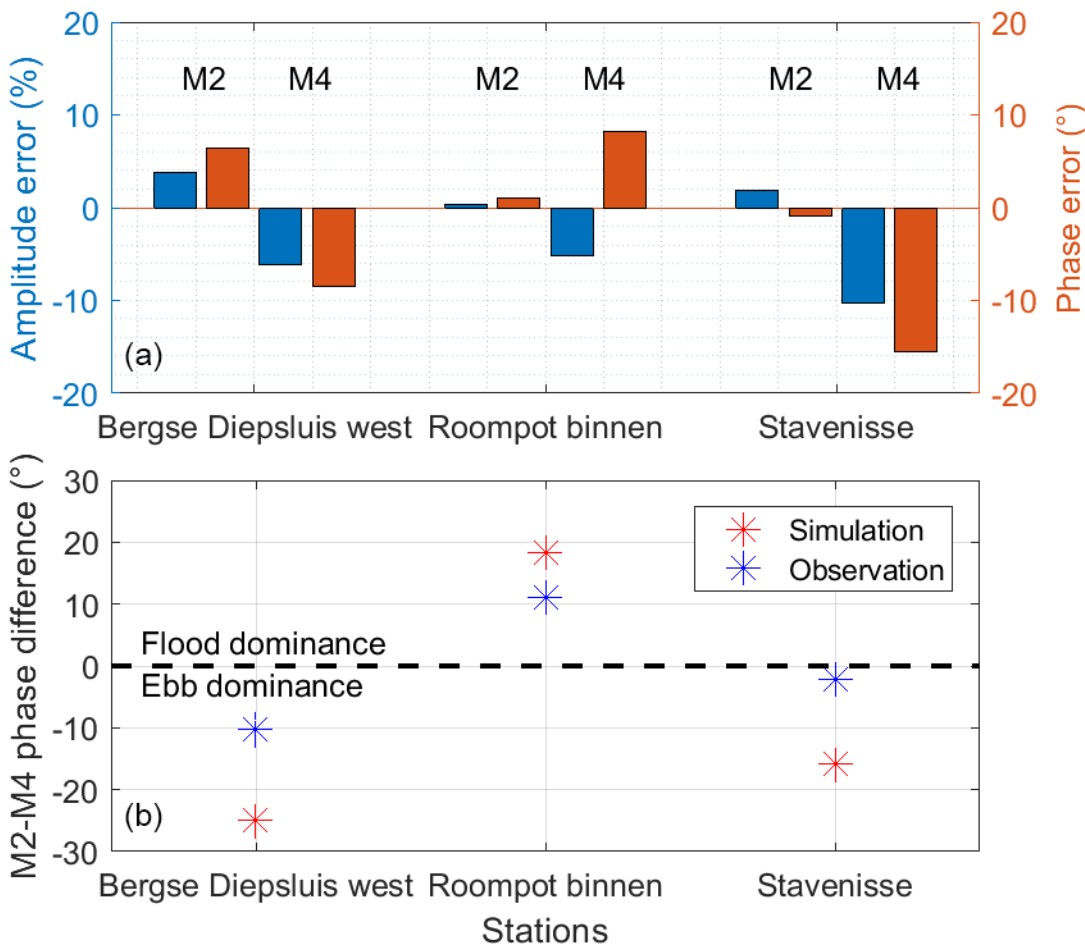

**Figure 3: The simulated (a) M2 and M4 amplitude and phase errors and (b) M2-M4 phase difference compared to observations. N Note that these M2 and M4 tides are calculated based on vertical tides rather than horizontal tides, i.e., different from those in Figs. 6 and 7. Tidal asymmetry is defined based on Friedrichs and Aubrey (1988). See Fig. 1 for locations of data sites.**

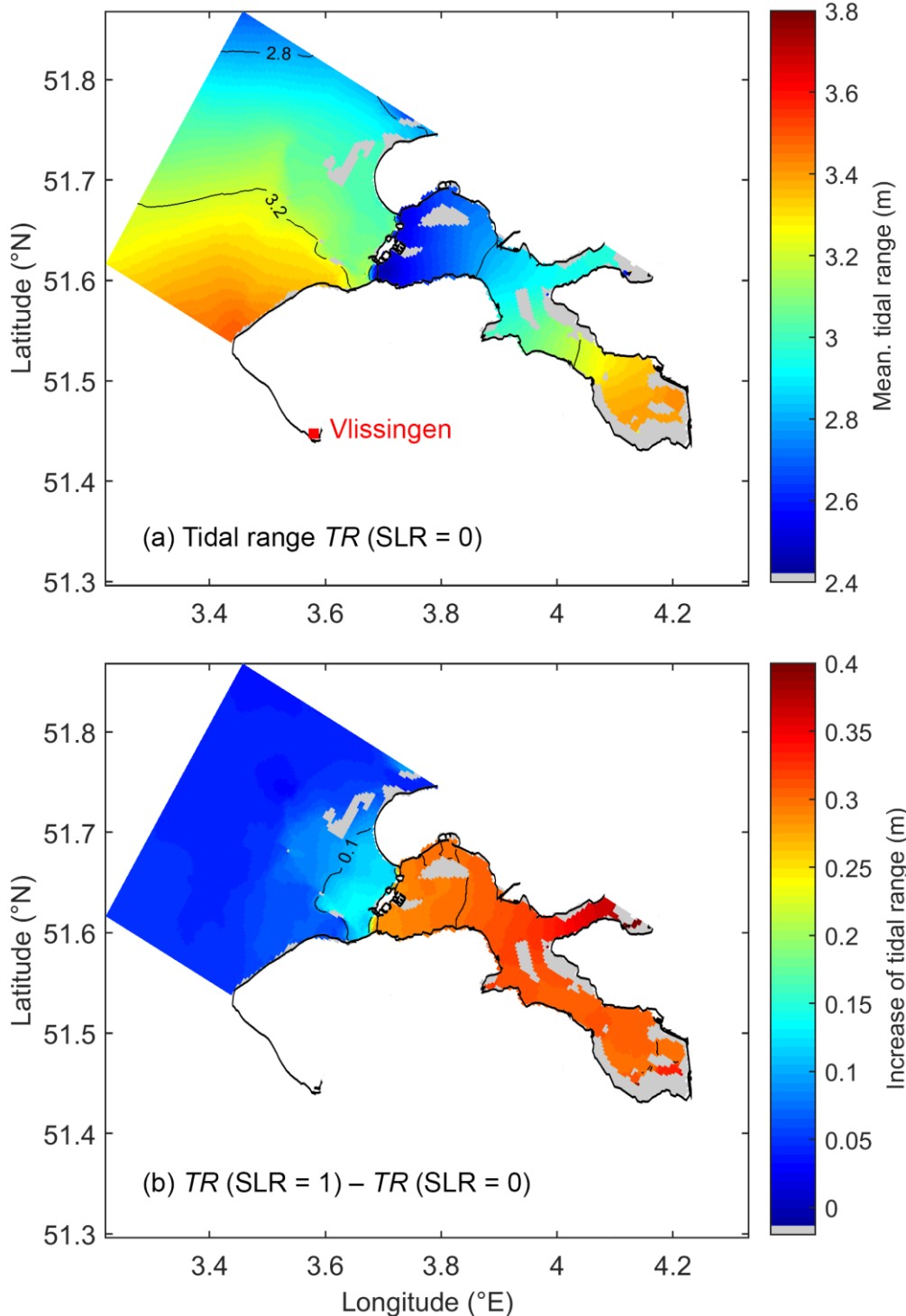

**Figure 4: (a) Tidal range in the baseline scenario and (b) the difference of tidal range between the 1 m SLR and baseline scenarios. Tidal flats are shown in grey. Tidal range is calculated as the annual average of the difference between high and low water in every tidal cycle.**

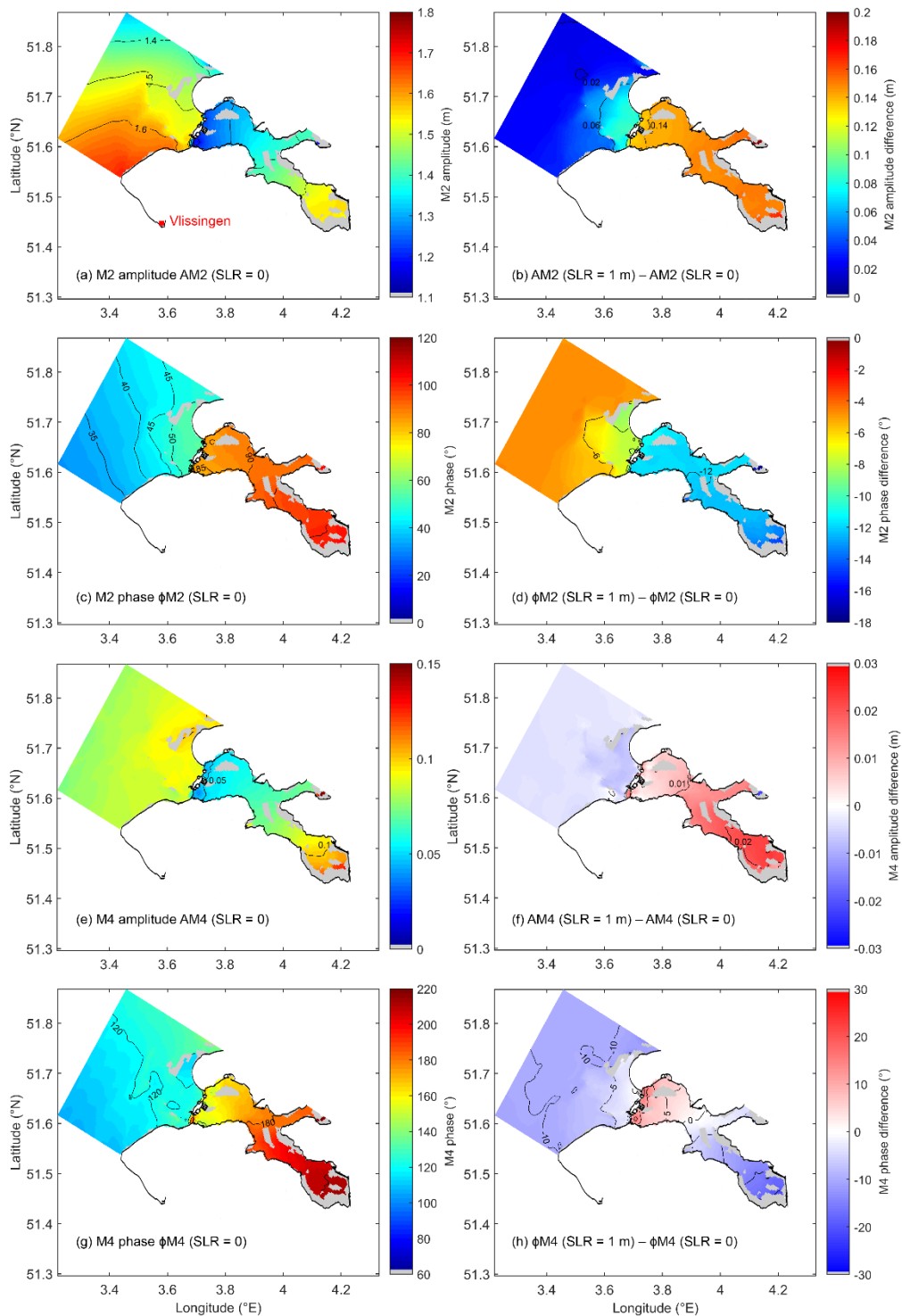

**Figure 5: The M2 (a) amplitude and (c) phase and M4 (e) amplitude and (g) phase in the baseline scenario and (b, d, f and h) the difference of these variables between the 1-m-SLR and baseline scenarios. Tidal flats are shown in grey.**

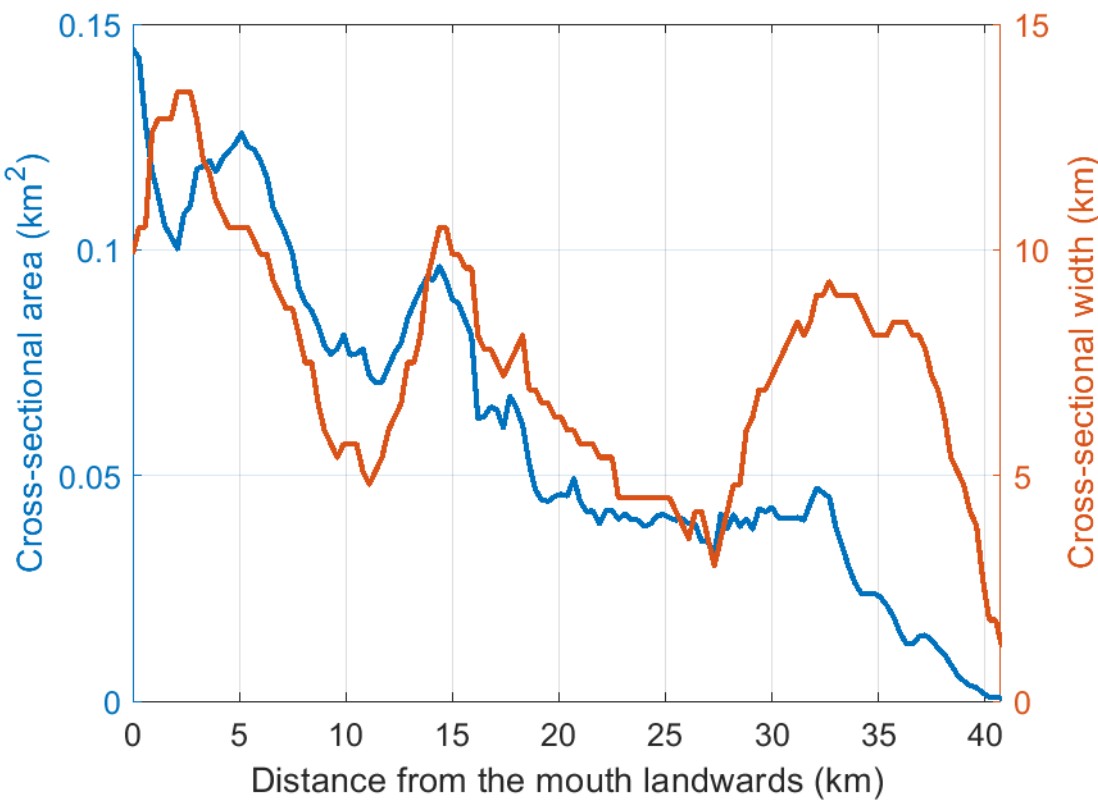

**Figure 6: The cross-sectional area and width of the Oosterschelde from the mouth to its eastern end. The northern branch (Fig. 1) is excluded from the calculation because of a different orientation of channels.**

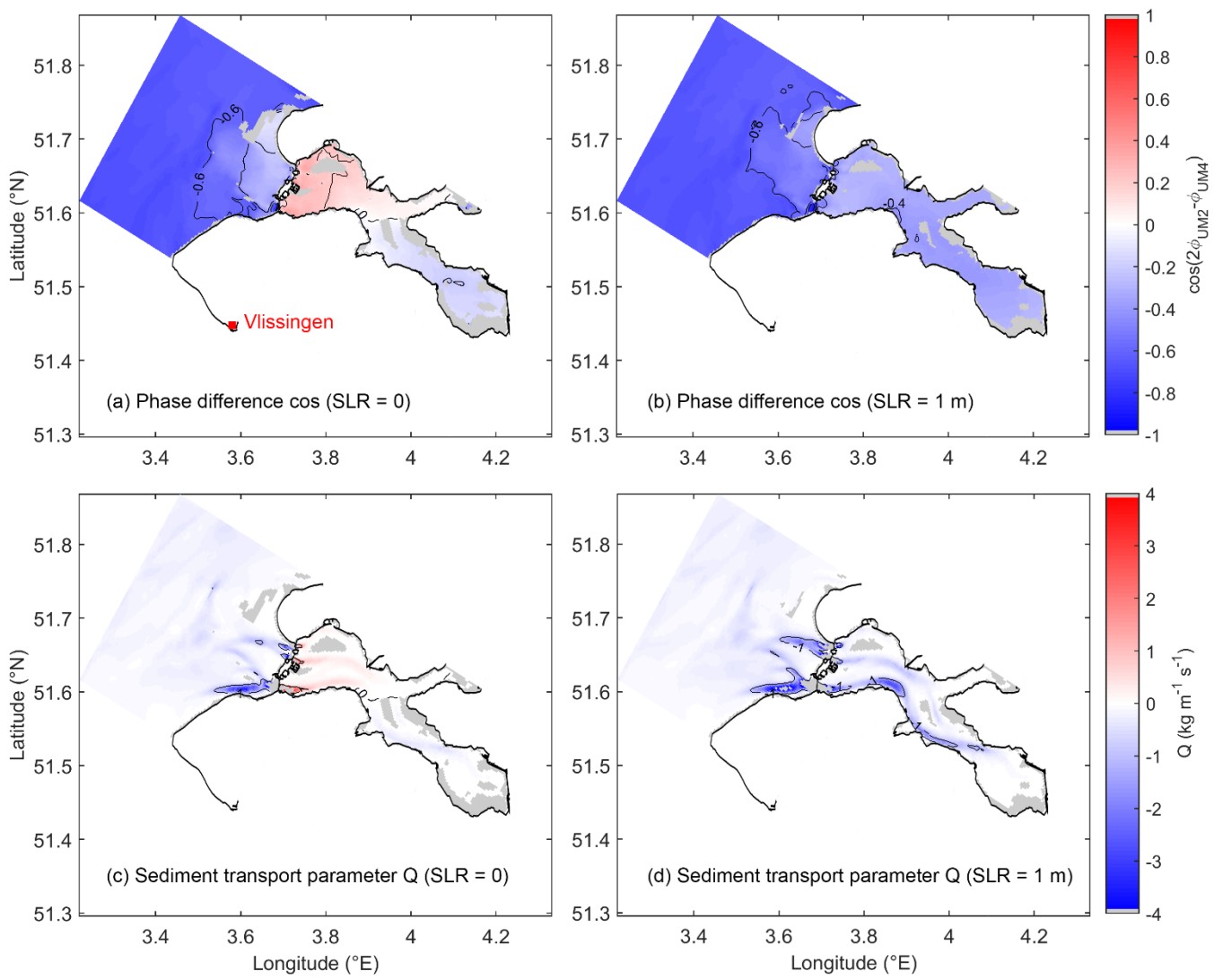

**Figure 7: (a and b) cosine of M2-M4 velocity phase difference and (c and d) sediment transport quantity $Q$ in the baseline and 1 m SLR scenarios. Positive (negative) $Q$ denotes landward (seaward) transport. Tidal flats are shown in grey.**

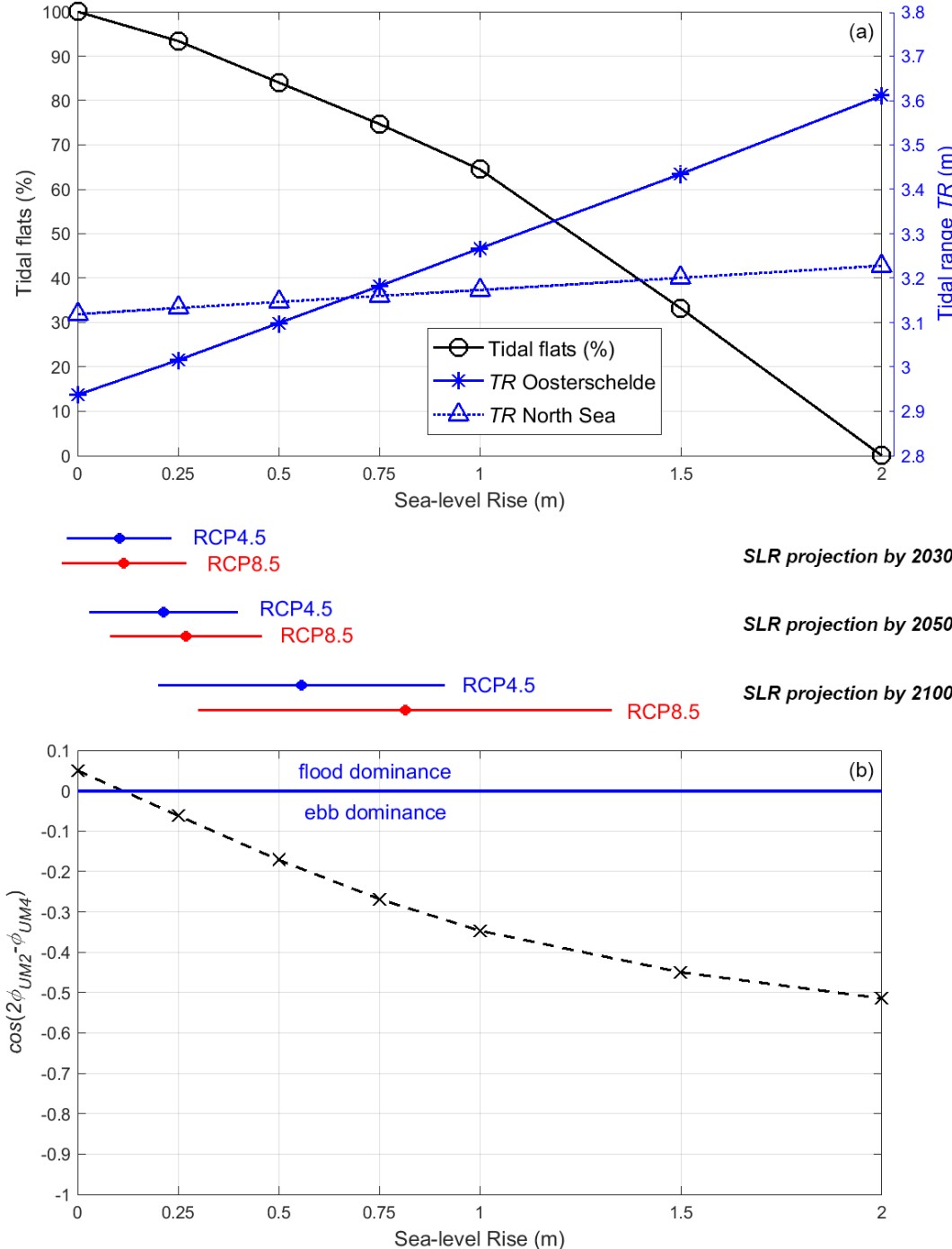

**Figure 8: (a)** Variations of the basin-average tidal flats percentage (defined as 100% in the baseline scenario) and average tidal range of the Oosterschelde and North Sea in the baseline and SLR scenarios; **(b)**, same as (a), but for the cosine of M2-M4 velocity phase difference averaged over the Oosterschelde. Scales between two panels are the local SLR projections (between the 5% and 95% confidence levels) in emission scenarios RCP4.5 and RCP8.5 in future decades, and dots on each scale denote projection medians.

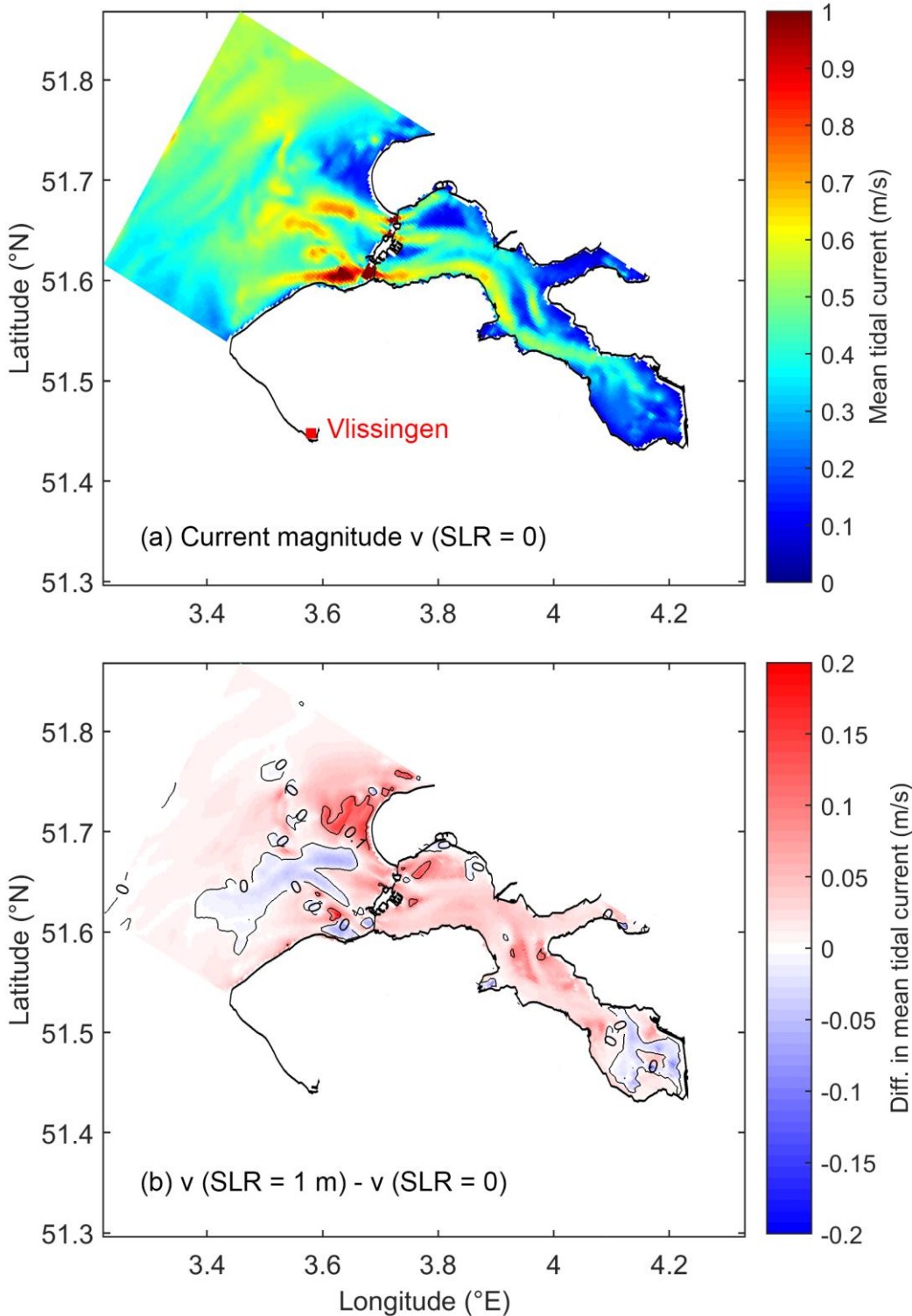

**Figure 9: (a) The annual average root mean square tidal currents $(u^2 + v^2)^{0.5}$ in the baseline scenario and (b) the difference between the 1-m-SLR and baseline scenarios.**

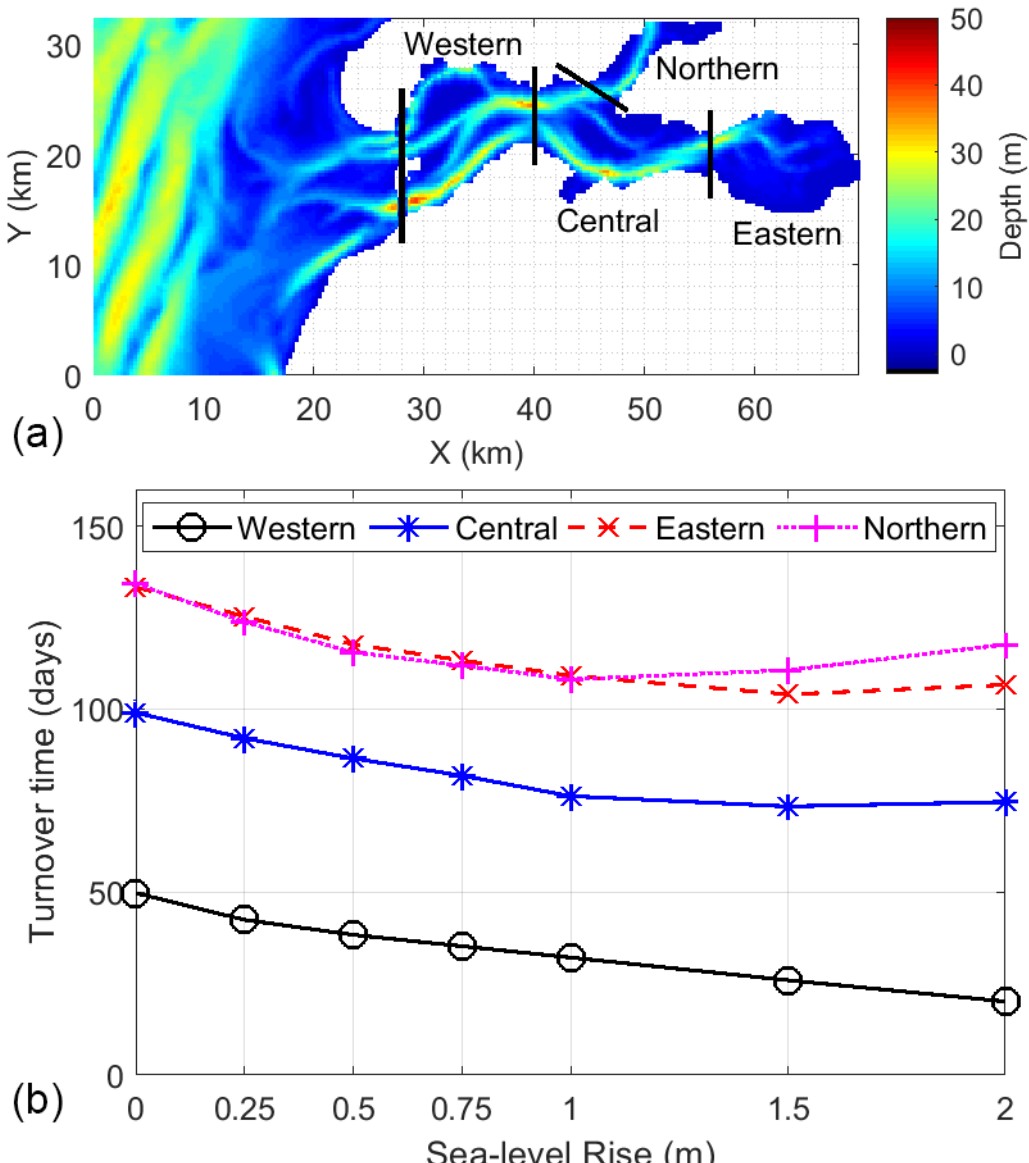

**Figure 10: (a) Division of four compartments in the Eastern Scheldt and (b) the turnover time of each compartment in the baseline and SLR scenarios. The calculation of turnover time is detailed in Jiang et al., 2020.**