# Peer review of "Effects of sea-level rise on tides and sediment dynamics in a Dutch tidal bay"

_Ocean Science, 2019_

## Short Comment (SC1) · 6 Sep 2019

Review of "Downscaling sea-level rise effects on tides and sediment dynamics in tidal bays", by Jiang et al.

In this manuscript, Jiang et al. describe a nested model in which a large regional model (2km resolution) is downscaled to an estuary in The Netherlands (the Eastern Scheldt). Sea-level scenarios are run and it is shown that tide changes are much bigger in the estuary than in the North Sea. Moreover, increasing sea-level is observed to shift the estuary towards ebb-dominated currents, with implications for sediment transport.

Overall, this is an interesting paper with some interesting results, and will be a good contribution to the growing literature on sea-level effects on the hydrodynamics of estuaries. However, the analysis and discussion of estuary tides and sediment transport could be improved, and many of the important papers and physical insights from the last decade or so could be referenced and used to help interpret the model results. The model is incompletely described, and more error statistics and discussion of sources of uncertainty would be good. In many places there are some additional analyses that could be done that would increase the novelty of the effort. Also, sediment transport in estuaries is complicated, and one usually should not ignore density/salinity effects; therefore, would suggest that the manuscript be more careful in how implications to sediment transport are described, and perhaps frame the discussion of results more in terms of hydrodynamic quantities (e.g., relative phase) that strongly suggest that important components of transport have changed.

Specific Comments:

Page 1 Line 18 "Global and regional tidal regimes" While regional tide changes have been observed or modelled, ocean scale changes to tides have not been. Would suggest removing "global"

Line 23 The Chernetsky reference is 2010, not 2011

Line 25 " Nienhuis and Smaal, 1994" This is a rather old reference. Can you find a few others? There are a number of references about tidal changes and effects on currents, transport, salinity, sediment concentration, oxygen concentration, etc for estuaries such as the Ems, Gironde, Loire, Hudson, Western Scheldt, etc.

Page 2 "ramifications for residual sediment transport and morphodynamic development"– Would suggest also referencing one of the more recent papers out of the Schuttelaars group (maybe the Dijkstra paper on the Western Scheldt). They have thought a lot about tidal asymmetry. Ton Hoitink probably may also have some relevant papers, if memory serves.

"Tidal changes due to SLR" Would suggest also referencing the very nice Ensing et

al. 2015 paper. There are many other papers on the effects of SLR on tides (another relevant one is Passieri et al., 2016). Somewhere, would suggest also referencing the forthcoming review papers on tide changes–Haigh et al., 2019 submitted to Annual Reviews of Geophysics, and Talke & Jay, 2020, Annual Review of Marine Science (https://www.annualreviews.org/doi/pdf/10.1146/annurev-marine-010419-010727 )

"without considering tidal changes in the shelf seas that may propagate into estuaries/ bays." This is a good point. However, it is also true that tide changes in a bay or estuary could affect basin tides–see the Godin paper on Bay of Fundy, and the similar papers by Arbic, Garret, et al. Would suggest also including this detail here, and acknowledging in the Methods that feedback effects into the ocean are not modeled (or are they?) with your downscaling approach.

"tidal waves on the shelf are significantly modified in amplitude and phase"– would replace "are" with "can be". When there is a steep shelf (e.g., US West Coast), there isn't very much modification that occurs.

Introduction, general comment: The introduction would be improved by surveying the local changes to tides that have been observed in the North Sea but also in theWestern Scheldt, the Rotterdam waterway, etc. See for example Winterwerp et al. 2013, Cai et al. 2013, Hollebrandse 2005, or van Rijn et al 2018. There is an analogy to be made between channel deepening and sea-level rise, though the analogy is not exact. See again the Ensing et al. paper for dynamical insights. In summary, there are many changes on historical dredging effects that could be referenced and provide validation that estuary tides can be very sensitive to bathymetry changes.

Page 3 "are projected to increase mainly due to reduced friction". Isn't the changing amphidrome also a factor? Would suggest commenting on its relative importance.

"tidal wave propagation can be Accelerated". Not sure this is the best wording, since this would suggest constantly changing phase speed. Maybe "tidal phase speed is increased"?

General comment: Use of acronym "ES" sometimes takes away from the understandability. You could consider just using the word Eastern Scheldt or Oosterschelde.

"MARS was forced". Could you comment in the text what guided the selection of these 14 constituents? Or in particular, why just one shallow water overtide? I presume M4 was quite small at the 200m isobaths, so is there any point in having it? Would be helpful to frame/discuss some of these issues, to help clarify the modeling methodology.

Remove the "The" in "the prescribing both water"

General comment: Am glad you considered variable MSL forcing on the boundary. Most studies do not do that.

Page 4 "Every scenario was run for one year" Can you comment on the consequences of missing the Sa and SSa constituents in your boundary forcing, which are probably larger than the fortnightly constituents you did include? Would be good to state what the magnitude of these constituents are, and what sorts of biases might be introduced by not including them. Or, stated differently, what do your sea-level rise scenarios suggest about seasonal variations in tide amplitudes, and at what point will sea-level rise effects be greater than seasonal variability?

Vertical eddy viscosity $K_v$– Why did you not use eddy viscosity from the model? Not sure using the same value everywhere makes sense. Also, is this a tidal average? Please specify. At the very least it would be good to ascertain that your modeled eddy viscosity is consistent with this value. What I would guess (and your results show) is that velocity decreases quite a bit into the estuary (since velocity goes to zero at the head of tides), such that a constant eddy viscosity is a poor representation of reality. This is also a factor that will change with sea-level rise. Hence, might suggest looking into spatial patterns of eddy viscosity, and how they change with SLR. This is usually an easy output in a model, and would be something new (and would give insights into changed frictional

Constant erosion parameter. While this is used in Graewe et al 2014, is the assumption of a constant erosion parameter justified in an estuary in which sediment properties can be highly variable? Also, is this formulation valid for the cohesive sediments found in estuaries, which behave quite differently than sand? Finally, semi-analytical models in estuaries include both an erosion parameter (somewhat analogous to the one here) and an erodability parameter that is a strong function of location. This is because estuary turbidity maxima form within estuaries, changing sediment availability (i.e., some places have mud banks, others don't). Please look into and discuss more thoroughly the validity of the Graewe formulation within estuaries, and carefully frame what is not included here and what the consequences of that are. There is probably also specific information about the Eastern Scheldt that can be found in the grey literature or similar about sediment sizes, erodability, etc. that could/should be discussed and referenced to help place your results in context.

General comment: there are other types of barotropic sediment transport that can be important besides tidal asymmetry (e.g., Tidal return flow, settling lag, etc). Would look at some the papers from the Schuttelaars group. Also, can you back up the assertion that gravitational circulation, internal asymmetry (now called "ESCO"; see one of the Dijkstra papers), and other types of tidal asymmetry are not important in the Eastern Scheldt, ideally with references or measurements? If there is a salinity gradient between ocean and freshwater, then it is at least somewhat important, in some places. There should be some information on this, and the salinity structure in the estuary should be discussed/referenced.

"the directional changes in residual sediment transport in different SLR scenarios." – Given the various caveats mentioned above, would frame this as sensitivity of one component of sediment transport to SLR scenarios.

General comment, methods: Did you account for the infrastructure at the Delta Works that caused tidal amplitude to decrease 13%, as stated earlier? Am not sure a resolution of 300m would be sufficient to model any bridge piers or storm surge structures in

an adequate way. However, it is essential to model this infrastructure in some way Tit would be incorrect to simply increase the drag coefficient in the entire estuary as a way to obtain realistic tides. In general, some description of the inlet infrastructure would be good (It looks like an Island was built, but there must be other structures as well).

A related note: Did not see any information about model calibration in section 3, even though section 3 promised (first paragraph) to discuss calibration. Information about tide stations used (and where to find data), statistics about root mean square error (for the different constituents), and so on is needed to assess how well the model is performing. Some of this information is given at the start of section 4, but it would be good to expand this.

General Comment: Was there wetting/drying in the model? This is very important for bathymetries in which there are intertidal flats, as there are here. For example, it can alter tidal amplitudes and tidal velocities. Please discuss whether you have wetting/drying, and the consequences if you do not (based off of known literature).

Section 4 Figure 2–Could you somewhere discuss the relative phase of the water levels (2M2 − M4) in your model, vs. the measured relative phase? This will give some indication about whether you are getting the tidal asymmetry correct.

Also, it would be useful if your discussion of the calibration discerns between errors at the ocean boundary and errors that are produced within the estuary. In other words, can you discern between the "external M4" and the "internal M4", as in Chernetsky et al. 2010? In that vein, in might be useful to extend your calibration and discussion to coastal gauges that are outside the estuary (e.g., Den Helder, Vlissingen, and some other nearby coastal gauges). Having only 3 calibration points is a rather small sample size, especially since the wider domain encompasses many tide gauges. It would be useful to know how well the larger model is doing (with comparison statistics).

General Comment: Can you let us know what the phase between tidal velocity and tidal elevation is at different locations (e.g., for M2), and discuss implications? The phase

provides insight into whether there is a Stokes Drift and an associated return flow (see e.g., Moftakhari et al. 2016).

Page 5 These results are interesting. However, projecting into the future is fundamentally a counterfactual. It's a "what if" scenario that cannot (yet) be proven, yet depends a lot on the assumptions made in the future projection (flooding vs. no flooding, for example, or the assumption of no morphological change). Also, would argue that the modeled future tides depend a lot on how friction was modeled in the estuary, whether and how wetting/drying is included, etc. Further, unmodeled small scale infrastructure (tide-gates) and small scale channels might (and probably do) matter. Some discussion of such uncertainties is needed. Again, the Ensing et al. paper has some insights, but see also the Lee et al. 2017 paper for the Chesapeake.

Obviously one cannot include everything, and the comment above doesn't just pertain to this paper. However, can you think of ways to address what the consequences of various modeling decisions are, and discuss how they impact results? For example, how might trends with MSL change if friction is changed by +/- 10%? What would be the consequence of random perturbations in bathymetry, or if only the channels (but not the flats) get deeper (i.e, an assumption of partial morphodynamic adjustment)? Finally, might suggest running the model with and without the storm surge barrier infrastructure, to see if your model is able to approximate the historical change to the model. As argued in the Talke & Jay review and references therein, doing a retrospective model run is helpful in terms of making sure that your model can at least reproduce past trends (thus increasing confidence in future trends).

A similar comment is that at present the trends are given to 3 significant figures (e.g., 0.337m per m sea-level rise), which is almost certainly not justified when sources of error are considered. It will help the long-term "staying power" of the paper if the quoted figure could have some sort of confidence or certainty interval. The quoted error statistics on the line fit are not the same as the actual uncertainty Tthe close correspondence to a line shows that, within the assumptions of the model, there is a linear system response (which is interesting). However, the model results themselves are not perfect, as shown in the calibration (the point I'm making is perhaps the difference between precision and uncertainty).

"under SLR the M4 amplitude decreases outside, while it increases inside ES" – Please explain why.

"Tidal waves in shallow waters propagate at a speed of sqrt(gh)" Actually, this is true only in the inviscid case (i.e., not your case), though it's not too far from this value in many cases. Would modify your text. Note that friction and convergence can strongly alter the phase speed. For your case, which is most likely weakly convergent and strongly (or moderately) frictional, would expect the phase speed to be somewhat less than sqrt(gh). Would suggest figuring out where in the parameter space mentioned above you are (e.g., by estimating your phase speed or by scaling), and discuss (the phase between velocity and water level also gives you an indication). In general, please look into the literature (e.g., Jay 1991, Friedrichs & Aubrey 1994, Lanzoni & Seminara 1998, and the many other idealized tide models) and discuss the processes in more detail, and how they affect results.

"decline in bottom friction favors faster wave propagation"–without explanation, this doesn't make sense. See comment above on frictional effects.

General comment–To what extent is reflection of the tide wave important? Do you see evidence of resonance, e.g., in the phase plots (in near resonance you get a fast phase speed)? It would seem that in addition to changes in friction (and convergence) caused by depth changes, you may have changes in reflection or partial reflection. See e.g., Winterwerp et al., 2013, Familkhalili & Talke 2016, or Ralston et al., 2019. In reflective estuaries, the biggest change in tides is usually seen at the boundary; in estuaries where depth/friction changes matter most and reflection is not important, the maximum tidal change is seen in mid-estuary (see again the Talke & Jay 2020 review).

Some discussion on resonance is found later, I see, but some more close analysis is

possible. One other idea would be to scale the relative importance of the convergence term and the friction term, to see if the rise in tide amplitude at the end of the estuary is due to friction that is weaker than convergence (e.g., Friedrichs & Aubrey, 1994).

General comment–Please explain why a transition to ebb dominance occurs. Perhaps the Friedrichs & Aubrey 1988 and Friedrichs & Madsen papers might have some insights.

"The quantity Q is used to estimate the combined effects of tidal current velocity and asymmetry". Before looking at Q, wouldn't it make sense to also plot out the M2 and M4 tidal currents (much like the amplitude plots)? It might also be interesting to see if the tidal ellipses change at all.

"the residual transport more than doubles" Again, would be careful about calling "Q" the residual transport. It is perhaps one type of residual barotropic transport, amongst many.

"this will not be accompanied by sufficient net sediment import as was in the past" Check grammar of this clause. Would also caution, again, about assuming that this is the only relevant source of transport. All coastal-plain estuaries that I've ever seen have a so-called estuary turbidity maximum that is caused by upstream transport. This is because baroclinic effects (ESCO, gravitational circulation) and settling lag effects are often so important. The paper would be helped by reviewing what is known about ETMs somewhere, both in general and in nearby estuaries (or ideally the Eastern Scheldt).

The results presented here (and the way they are framed) would suggest that no ETM forms, which is probably not the case and would likely be greeted with skepticism in the ETM community. For references, see the Burchard et al. 2018 review and references therein.

Discussion of resonance: Please give a general reference for Helmholtz resonance

beyond the one given later in the paragraph from one of the co-authors (in any case, Helmholtz resonance is usually just relevant for harbors). Also, 5.3 hours is not that far away from the M4 frequency, for which you see a big (and unexplained) amplification. M4 resonance is not unheard of, and occurs for example in Hecate strait (Foreman et al. 1993). In any case, please dig deeper into your results and try to figure out whether you see any markers of resonance or altered reflection properties for any of your constituents (see also comment above). Regardless of the conclusion, this will improve your discussion.

A related comment–please discuss how you came up with the time scale of 5.3 hours. Did you use average depth and length? Or did you stress test your model with different frequencies and see what happens? The latter would give you a more accurate estimate. In any case, idealized models show that in frictional systems, the tide wave propagates slower than sqrt(gh), such that the resonant time scale is modified (increased). Moreover, resonance with friction is broad-band, and there are a large range of frequencies that get amplified (again, see Talke & Jay 2020 and references therein). Do such considerations impact your analysis? (would seem not for M2, but the point is that using an inviscid quarter wavelength is only an approximation and potentially misleading, and that the paper would be improved by thinking about this in a more sophisticated way).

Page 7 "frictional damping increases the semidiurnal tidal amplitude by 0.03-0.05 m/m SLR in the study region"–Not clear what is meant by "study region". Please be specific.

General comment about bathymetric effects: Agree these are important. Would suggest that you reference some of the studies that have showed similar effects of convergence, depth variation, etc. in the past (including but not limited to Ensing et al., 2015).

"the Ems estuary may obtain a stronger flood-dominant signal" – There were differences between the "external" M2 and "internal M2" in the Ems. Basically, if memory

serves, the decrease in damping (in part caused by fluid mud, in part by depth change) reduced the damping of the external M4 more than the estuary M4 production was reduced. It would be helpful if you analyzed your results with this in mind. Also, what happens in the Western Scheldt? The Dijkstra et al. 2019 paper in ODYN discusses this estuary.

"Firstly, tidal responses to SLR can vary from system to system"–would say that this is already known. Perhaps modify conclusions, and make sure to include relevant references.

"and these effects may amplify in estuaries and bays." Again, would point to the Arbic et al. 2009 and Arbic& Garret 2010 papers. There is also the potential for changed estuary tides to feedback into the basin. Any evidence of that? Not sure the model framework can look into this (see earlier comments)

"for instance in parts of the Chesapeake Bay". Did you mean SF Bay? There are some interesting papers for the Chesapeake that should be referenced such as Lee et al., 2017 and Ross et al. 2017 (and Du et al. 2018).

"the gravitational force," Not sure what you mean by this. Do you mean Gravitational circulation/baroclinic effects?

"Density-driven flow can also dominate local transport processes" There are many other references, including reviews by Burchard et al. 2018 and Geyer and MacCready (2014) that address density gradient induced circulation and transport.

Figure 1– The surge barrier should be labeled, not just shown with an ellipse. In general, it would be more helpful to describe exactly how much of the channel crosssection is impeded by the storm surge barrier, and how this is modeled.

Figure 2 Can you explain why only these specific days of tidal modeling are shown? Without explanation it could be interpreted as "cherry picking" a period of time where the fit was good. In general, more statistics on calibration would be good.

Figure 3–How are you defining tidal range? There are different ways of doing that, so please specify.

Figure 4–The effect of the Delta Works is quite stark. Is there an effect of changing inlet cross-sectional area, i..e, as in Passieri et al. 2016? (That paper found variable changes to tides in back-barrier bays of the Gulf of Mexico, under sea-level rise scenarios. See also again the Talke & Jay 2020 review for discussion on and references for the "inlet choking effect".

Figure 7–Please provide information on how annual average was calculated. Is this based on peak velocity, rms velocity, average of the absolute value, or something else?

---

## Referee Comment (RC2) · Frank Koesters (Referee) · 29 Oct 2019

(plain text Version, see pdf of formatted version)

Review of 'Downscaling sea-level rise effects on tides and sediment dynamics in tidal bays' by Jiang et al. submitted to Ocean Science

General comments The paper by Jiang et al. presents a modelling study of a tidal bay, here the Eastern Scheldt, on the effects of sea-level rise on hydrodynamics and sediment transport. The study addresses an issue, which is of practical relevance to sediment management especially when considering future sea level rise. Overall the paper is clearly structured and written in an easily comprehensible but precise way. The authors apply results from a European Shelf model to force a regional Eastern Scheldt

model with different boundary conditions representing present-day and sea level rise (SLR) conditions. From these hydrodynamic modelling results they infer effects on sediment transport based on a simplified approach just taking the effect of M2 and M4 tides into account following Burchard et al. (2013). The authors conclude that the Eastern Scheldt will change with SLR from balanced flood-ebb current conditions to ebb dominance which will result in a loss of sediments of the Eastern Scheldt. The results of Jiang et al. add an interesting case study to previous assessments of SLR effects on coastal regions. Therefore, their work is relevant to the scope of Ocean Science. The modelling study applies state of the art techniques of model coupling to combine large-scale changes of hydrodynamics due to SLR and regional dynamics of a tidal bay. Based on their modelling results the authors draw the conclusion that the ES will change with SLR from balanced flood-ebb current conditions to ebb dominance and thus a potential loss of sediments. I find it hard to assess the model validity based on the presented results. I recommend a substantial improvement of model validation. I appreciate the approach to infer sediment transport from hydrodynamic quantities; here the ratio of M2 and M4 tides, but this certainly requires to carefully checking if results are consistent and plausible. Here the authors have to spent considerable more effort. In view of the complexity of hydrodynamic and sediment transport processes in tidal environ-ments, the paper's very simplified approach to asses residual sediment transport lacks a critical discussion of results.

Scientific comments

a) Numerical modelling - General Title and wording in the article are misleading when it comes to the term 'downscaling'. The authors imply that the apply a new approach but teir methods are neither new nor would I call it downscaling per se. However, their approach is appropriate to address the scientific question how tidal dynamics of the Eastern Scheldt (ES) will change with SLR. The term 'downscaling' in the title is in my view misleading but scope and methods are clarified in the abstract.

- Hydrodynamics One of the strong points of the paper is the analysis of possible

resonance effects even though they do not prove to be important here. A shortcoming of the modelling work is the model vali-dation. This holds for the general approach: not taking meteorological forcing into account is not well suited for a proper hydrodynamic model validation. This is especially important as the au-thors find that the presence of the storm surge barrier is relevant for SLR induced changes. Un-fortunately the storm surge barrier is not resolved therefore they have to proof that they can cover the effect under sea-level rise conditions properly. Please carry out a comparison of model results and measurements outside of the ES to show that the model captures the transformation due to the storm surge barrier narrowing. Moreover, I would suggest carrying at least out a sen-sitivity study for high mean water levels in the North Sea but for an open storm surge barrier in order to assess if the model adequately resolves higher water levels in the ES.

Further issues concerning the representation of hydrodynamics are: • Deviations of measurements and model results are rather large (up to 10 % in M4 ampli-tude), what about other error statistics (e.g. RMSE) and why would a further model cali-bration possibly not sensible? • The effects of tidal flats are analysed in detail but how well are they represented by the model? You might want to give a comparison of hypsometric curves of model and under-lying bathymetric data.

The discussion of the observed changes in tidal asymmetry in term of the interaction between the tidal wave and the basin geometry is going in the right direction, but is too short. You emphasize the importance of understanding the complicated interaction between basin geometry and tides. So why don't you give a thorough discussion on this aspect? For example, you indicate that tidal asymmetry depends on the a/h ratio as well as on the extension of intertidal area and find that ES becomes more ebb-dominant with SLR. So what does this mean with regard to the specific basin geometry of the ES and its significance for the tidal response to SLR?? Please consider further discussion on this aspect.

-Sediment transport One of the main results of Jiang et al. is that the ES will suffer

from sediment loss with SLR. Sed-iment transport is not modelled with a separate sed-iment transport model but inferred from hydrodynamics. This approach is on the one hand a simplification but is a neat way to obtain an estimate for important impacts of SLR. These results can have an important impact on local sed-iment management and the work of authorities in that area. Therefore, a careful model validation and sensi-tivity tests for sediment transport are required to prove reliable results. The validation needs to be generally improved. It is important to understand the complexity of hydro-dynamics and at least mention potentially relevant mechanisms at first before drawing conclusions on sediment transport. For example, tidal asymmetry is not only gener-ated by M2-M4 phase differences, but also due to hypsometric controls and lateral circulation. Please state model limitations more clearly in the discussion.

Please consider the following issues: • The assumption for sediment transport to be dependent on M2 and M4 amplitude and phase only is very crude as it neglects e.g. lateral circulation or density effects. Can you give any measurements for comparison? • Are your results valid for the observed range of sediment grainsizes? At least carry out a sensitivity study on assumptions made (e.g. is settling velocity ws of 1 mm/s representa-tive for the ES). • Why is the system experiencing sediment loss today (p. 6 l.13) if there is a mixed flood/ebb dominance? • Results for Q are presented as averages over ES. What can we infer from that? I guess that is very difficult to interpret, e.g. you have a Q of about -0.35 kg/(m s) for an equal flood and ebb tide dominance (which would locally result in Q = 0). Please revise analysis or explain it. • The authors estimate the net sediment transport Q inferred from the relation of Burchard et al. (2013). For a sea-level rise of 1 m Q would be around -0.8 kg/(m s). Please discuss the effect on the ES, e.g. taking the width of the ES at the mouth of about 10 km one would get an export of sediment of about 8,000 kg/s which relates with density rho 1,600 kg/m$^3$ to a volume loss of 158 Mio. m$^3$ p.a.. Is that sensible? • A steady state topography is assumed for SLR which will not be the case, what kind of feedback mechanisms are to expect?

Even though the results needs further corroboration and discussion on model limitations I like to stress that the conclusions are presented in a clear and comprehensible way.

Specific and technical comments • Abstract: o Clearly written but last sentence is misleading. One would expect way more than just model coupling. o 'our model downscaling approach' implies a novel approach for the model setup. However, downscaling is actually not new in modelling of shelf seas and shallow coastal waters. • Introduction: o Their brief literature review is almost sufficient. However, some more reference to published work using their (coupled) modelling approach in other coastal regions would be appropriate. Are there no previous relevant studies for the ES itself? o Formally I would expect to find references in chronological order. o p.1 l.22 not only salt marsh accretion, but also tidal flat accretion is influenced (those two are not the same) o p.2 l.3 refer also to Pelling et al. (2013), who focused exactly on this aspect (the decrease in tidal amplitude as a consequence of enhanced dissipation in newly inundated areas) o p.2 l.5 Do you mean in shallow ebb-dominant estuaries? o p.2 l.5-8 A high ratio of a/h usually coincides with vast areas of intertidal flats, since large intertidal flats result in a small mean basin depth. Hence, there is NO contrast between the two aspects you describe as you say, too: seaward transport is reduced (first sentence) and there is a transition to flood-dominance (second sentence). This is the same direction of change in tidal asymmetry. o p.2 l.6 sentence mixed up o p.2 l.7 'may cause' o p.2 l.14 Is it really true that most modeling studies simply prescribe SLR as an additional water elevation at the seaward boundary? Please refer to some exam-ples, where it was simply added. o p.2 l.17 Pickering et al. (2017) use a global model and thus are probably no ap-propriate reference for statements on the tidal propagation on shelfs. o p.2 l.23 as said in the general feedback: the downscaling method is already broadly applied for simulating tidal dynamics in coastal waters. • Study site o p. 3 l. 4 water depths • Methods o p.3 l.11-12 Add some more information about the model here. Are the models 2D or 3D? And if it is 3D, what is the vertical resolution? Which quantities are con-sidered, which are relevant to density effects (transport of

salt, heat)? o p.3 l.16 Better: 'The MARS domain extends to deep waters and covers the entire North-West European continental shelf...' This makes it more clear that the model not only extends to deep waters along a few sections of the open boundary, but captures the entire shelf edge.) o p. 3 l. 24 reference Slangen 2014 is missing; restructure sentence o p.4 l.10 Depending on the grain size, the time lag between local suspended sedi-ment concentration and current velocity is not necessarily negligible. With regard to finer fractions (especially silt fractions) settling lag effects are important! • Results o p.4 l.24 The correlation coefficient is not ideal to assess a tidal signal as the tidal wave itself is a very strong signal compared to the error, therefore observed and modeled tidal water elevations always have a relatively strong correlation - also in case of rather low model accuracy. For example, you may add RMSE, which is more appropriate. o p.5 l.8 "the main tidal patters" . . .of what? Tidal current velocity? Please clarify. o p.5 l.11-13 "With SLR, TR increases almost uniformly within ES": You may men-tion that this statement is related to the investigated SLR range up to 2 m and that this is not necessarily true for larger SLR values (e.g. for SLR > 2 m). (The same for p.5 l.19-21) o p.5 l.14 Please indicate more precisely, which region you mean with "adjacent North Sea"? Do you mean only the region directly located seaward from the bar-rier (up to which depth?) or the entire North Sea section within the GETM model or an even larger domain of the North Sea such as the Southern Bight? o p.5 l.16-18 I don't get your point. Why do you mention the fixation of the tidal ba-sin size when talking about the role of tidal range for tidal prism? An increased tidal range will always increase the tidal prims, no matter, if the tidal basin is fixed or not. A non-fixation would only further increase the tidal prism. Also keep in mind that as long as intertidal flats are present in the initial case (your baseline scenario) the tidal prism will always increase with SLR, even if tidal range is not increased (remains constant), because with SLR former tidal flat volume is added to the tidal prism. o p.5 l.18 Figure 7 does not show that. It looks like tidal currents are mainly in-creased on the tidal flats or shoals. How could this be explained? o p.6 l.9-10 and Figure 6b: Your analysis is very difficult to justify. You made an av-erage of residual sediment transport over the entire ES.

What does that tell you about the exchange between ES and the adjacent North Sea? When the tidal asymmetry is 0, I would expect the residual sediment transport to be minimal as well, if it is related to the exchange between ES and the adjacent North Sea. Furthermore, I would expect the intensity of residual sediment transport at SLR of 0.25 m to be roughly the same as in the baseline scenario (SLR of 0), since the magnitude of tidal asymmetry of current velocity is about the same. • Discussion and summary o p.6 l.17-18 Do you have any figure as proof of the value of 30 cm for the high wa-ter at spring tide? As shown in Figure 3 the increase in TR due to a SLR of 1 m is about 0.3 m. If tidal high water increases by an extra of 0.3 m (1.3 m in total), this means that tidal low water is elevated exactly by SLR (1 m in total)? So does that mean that the increase in tidal range by 0.3 m is solely induced by the increase of tidal high water? o p.6 l.19-23 "turnover time": Why do you place this totally different aspect here? It is not logically connected to the rest of the text, neither to the preceding nor to the following. o p.6 l.24 stronger tidal response [of tidal range] to SLR o p. 7 l. 5 'increases faster than' is unclear o p.7 l.6-8 I suggest not to generalize findings from this study, because the barrier is a special geometric feature strongly affecting the tidal dynamics in the ES (just as you say it in line 21). o p.7 l.11-15 A tidal basin with extensive intertidal flats actually corresponds to a high ratio of a/h. The relative importance of these two effects depends on the ra-tio of tidal flat area to channel area within the tidal basin. In your study site (ES) the channel area is much larger than the tidal flat area, suggesting a stronger de-pendence on the a/h ratio. This could explain why tidal asymmetry shifts towards ebb dominance with SLR. o p.7 l.27-28 I agree with you, but did you make any comparison to results of the shelf model (MARS)? If not, how can you conclude that shelf models are less ap-plicable? o p.7 l.31 Are you sure that most studies on this topic neglected this aspect? I guess that most studies actually considered it. o p. 8 l. 16 you certainly take into account 'gravitational force', do you mean tide generating forces within the model

• Figures o The number of figures is adequate but the quality needs to be generally improved. A coastline and at least some geographic information would help. o

Specifically: When you state Depth in m, what is the vertical reference system? Meter below NAP? IS it the same for regional and shelf model? What is the coor-dinate system you are using? I would prefer coordinates instead of model dimen-sions for the axes. o Figure 1: Cannot see gauge locations well. o Figure 3: At least some geographic information would be nice, e.g. show Vlissing-en from Fig. 1 o Figure 4:e) should be amplitude not phase in the o Figure 6: references a) and b) missing o Figure 7: show coastline / land

Please also note the supplement to this comment:
https://www.ocean-sci-discuss.net/os-2019-50/os-2019-50-RC2-supplement.pdf

---

## Referee Comment (RC3) · Mick van der Wegen (Referee) · 31 Oct 2019

Dear Authors;

Thank you for your interesting contribution to the discussions on SLR impact in estuaries. I think you did some decent work in downscaling tidal dynamics and analysing the -potential- impact of SLR on the tidal dynamics. Also you explored the potential impact on sediment transport and tidal flats survival. I think this latter aspect deserves some more clarification and discussion. In the attached document I have made some minor comments. Below I formulate my major concerns. I have no doubt accepting your work when these are adequately addressed.

1) You disregard morphodynamic development. This may be a justified assumption in

the sense that the morphodynamics potentially create an extra and yet unclear dimension to the work. It is good to restrict your efforts sometimes. However, I feel that there will be significant morphodynamic analysis coming 100 years in the ES.

2) You consider coarse sediment only (neglecting a settling lag) whereas muddy sediment will be relevant as well. Mud could import while sand exports. Mud could heighthen flats. A sand export could deepen channels while flats are maintained.

Drawing strong conclusions based on an analysis that disregards morphodynamics and fine sediments seems not justified. I suggest to rephrase the summary and conclusions acknowledging more clearly that morphodynamics and fines were not considered. You may add a discussion on why you disregarded these and what important implications of that assumption could mean to your results (like the heightening of tidal flats).

* I believe it is also the reduced tidal range (and not only the reduced sediment supply) in the ES that makes the intertidal area to erode. Wave action is more concentrated at a speciifc height (in a smaller tidal range) causing more erosion of the tidal flats.

* I am interested in how the ES is implemented in the MArs model: can you explain that a little bit more what the assumptions and implications are of the one way coupling? To what extent does the MARs model include the effect of the ES? Are the GETM boundaries far enough at sea to have no effect of the ES dynamics under SLR?

* I miss conclusions since these are merged in the discussion. Please differentiate into "discussion" and "summary". And maybe add sub-headings in the section that is now called "discussion and summary"

with kind regards

Mick van der Wegen

Please also note the supplement to this comment:

https://www.ocean-sci-discuss.net/os-2019-50/os-2019-50-RC3-supplement.pdf

---

## Editor Comment (EC1) · John M. Huthnance (Editor) · 15 Nov 2019

Dear Authors The three reviewers' scores indicate that your manuscript should be worth publishing in due course. However, as all three recommend "Major Revision" (and are willing to see it again) you should take their comments seriously and respond accordingly. Please incorporate your responses (perhaps summarised) in your revised manuscript so that it is self-contained and readers do not need to refer to the responses in the editorial system. Yours sincerely John Huthnance

---

## Author Comment (AC4) · 13 Jan 2020

Frank Koesters (Referee)
frank.koesters@baw.de

**General comments**

The paper by Jiang et al. presents a modelling study of a tidal bay, here the Eastern Scheldt, on the effects of sea-level rise on hydrodynamics and sediment transport. The study addresses an issue, which is of practical relevance to sediment management especially when considering future sea level rise.

Overall the paper is clearly structured and written in an easily comprehensible but precise way. The authors apply results from a European Shelf model to force a regional Eastern Scheldt model with different boundary conditions representing present-day and sea level rise (SLR) conditions. From these hydrodynamic modelling results they infer effects on sediment transport based on a simplified approach just taking the effect of M2 and M4 tides into account following Burchard et al. (2013). The authors conclude that the Eastern Scheldt will change with SLR from balanced flood-ebb current conditions to ebb dominance which will result in a loss of sediments of the Eastern Scheldt.

The results of Jiang et al. add an interesting case study to previous assessments of SLR effects on coastal regions. Therefore, their work is relevant to the scope of Ocean Science. The modelling study applies state of the art techniques of model coupling to combine large-scale changes of hydrodynamics due to SLR and regional dynamics of a tidal bay.

Based on their modelling results the authors draw the conclusion that the ES will change with SLR from balanced flood-ebb current conditions to ebb dominance and thus a potential loss of sediments. I find it hard to assess the model validity based on the presented results. I recommend a substantial improvement of model validation.

I appreciate the approach to infer sediment transport from hydrodynamic quantities; here the ratio of M2 and M4 tides, but this certainly requires to carefully checking if results are consistent and plausible. Here the authors have to spent considerable more effort.

In view of the complexity of hydrodynamic and sediment transport processes in tidal environments, the paper's very simplified approach to asses residual sediment transport lacks a critical discussion of results.

Response (1): Thanks for the constructive and detailed comments on our manuscript. We have revised the manuscript accordingly. Specifically, we improved the model validation by including additional statistical analysis and expanded the discussion as suggested. We have also interpreted

the results on the sediment transport quantity $Q$ with more caution given the limitations and simplifications of this method. Please see the following responses for detailed revisions.

**Scientific comments**

a) Numerical modelling

- General

Title and wording in the article are misleading when it comes to the term 'downscaling'. The authors imply that the apply a new approach but their methods are neither new nor would I call it downscaling per se. However, their approach is appropriate to address the scientific question how tidal dynamics of the Eastern Scheldt (ES) will change with SLR. The term 'downscaling' in the title is in my view misleading but scope and methods are clarified in the abstract.

Response (2): We have changed "downscaling" to "model coupling" in the title and the text, where necessary. It is indeed not a new approach. Our main point is that it seems an appropriate solution compared to considering SLR only but not taking tidal changes on the shelf into account. It is clarified in Introduction (e.g, from Page 2 Line 34 to Page 3 Line 8 of the "accept-changes" version of the revised manuscript) and Discussion (e.g., Page 10 Lines 22–34 of the "accept-changes" version of the revised manuscript). Hereafter, wherever the page and line numbers occur, we refer to the "accept-changes" version.

- Hydrodynamics

One of the strong points of the paper is the analysis of possible resonance effects even though they do not prove to be important here. A shortcoming of the modelling work is the model validation. This holds for the general approach: not taking meteorological forcing into account is not well suited for a proper hydrodynamic model validation. This is especially important as the authors find that the presence of the storm surge barrier is relevant for SLR induced changes. Unfortunately the storm surge barrier is not resolved therefore they have to proof that they can cover the effect under sea-level rise conditions properly. Please carry out a comparison of model results and measurements outside of the ES to show that the model captures the transformation due to the storm surge barrier narrowing. Moreover, I would suggest carrying at least out a sensitivity study for high mean water levels in the North Sea but for an open storm surge barrier in order to assess if the model adequately resolves higher water levels in the ES.

Response (3): Thanks for the comments. Please see the following four aspects addressing the comment.

1) About the missing meteorological forcing, the GETM setup for the Oosterschelde has been calibrated and validated with realistic meteorological forcing and gravitational circulation in our previous study (Jiang et al., 2019). In the current study, the key point is to simulate the "future"

SLR scenarios. It is difficult to include predicted meteorological forcing and river runoff for the future. Thus, both nested models MARS and GETM are run without meteorological forcing. Despite the absence of meteorological forcing in the model, the coupled model is able to simulate the water elevation with reliable performance, based on which scenario the further SLR runs were conducted.

2) The storm surge barrier is not unresolved. In Methods of the revised manuscript (from Page 4 Line 30 to Page 5 Line 2), we have clarified how it was implemented in our model. The storm surge barrier includes two manmade islands and three openings as shown in Fig. 1. Each tidal opening consists of concrete pillars and steel gates that can be closed under severes stormy conditions (Fig. R1). The bed of the barrier is secured by a sill resulting in a much decreased water depth (Fig. R2). The pillar is around 4 m wide, which is much narrower than the grid size of the model (300 m). The model would need a much finer resolution and smaller time step to resolve the pillars, which is computationally inefficient and unpractical. Given that the pillars accounts for only 8.2% of the overall cross-sectional area based on our calculation, we reduced the depth accordingly (i.e., by ~8.2%) to maintain the realistic cross-sectional area of each tidal opening of the barrier. It turns out that our model setting reasonably captures the realistic water elevation inside the Oosterschelde (Jiang et al., 2019).

[Figure]

A = sill (sediment)
B = tidal water
C = pier
D = steel gate

**Figure R1: The schematic cross-sectional view of the storm surge barrier of the Oosterschelde. Source: Nienhuis and Smaal, 1994.**

[Figure]

**Figure R2: The side view of the storm surge barrier of the Oosterschelde. Source: Wikipedia Oosterscheldekering, https://en.wikipedia.org/wiki/Oosterscheldekering.**

3) The calibration in the North Sea was conducted by Idier et al. (2017). To show the calibration results, we are here extracting the validation results (Figs. R3 and R4) from this paper, which indicate that the European Shelf model simulates the water elevation and major tidal components reasonably well.

[Figure]

1: Dunkerque      5: Helgoland      9: Scheveningen  13: Aberdeen
2: Saint Malo     6: Liverpool      10: Ouistreham   14: Wick
3: Le Conquet     7: Plymouth-Devenport 11: Dover     15: Etretat
4: La Rochelle    8: North Shields  12: Esbjerg      16: Portland

**Figure R3: Computational domain of the MARS model with locations of tide gauges used to validate the model. Source: Idier et al., 2017.**

[Figure]

(a)

(b)

**Figure R4: (a) Modeled versus data-based highest annual tide (for the year 2009) for each site. The numbers refer to the tide gauge names in Fig. R6; (b) modeled versus data-based tidal component amplitudes for 5 selected tidal components. + and • indicate sites with**

relative errors for the highest tide that are smaller or larger than 5%, respectively. Source: Idier et al., 2017.

4) A sensitivity test without the storm surge barrier is a good suggestion but will not help the model validation. The current Oosterschelde is a completely different system from the pre-barrier (i.e., without the storm surge barrier) times not only because of the storm surge barrier. As part of the Delta Works in the late 1980s (https://en.wikipedia.org/wiki/Delta_Works), many other dams and sluices (Fig. R5) are built at approximately the same time cutting the freshwater input of the Oosterschelde, which along with other systems (e.g., the salt-water lake Grevelingen and freshwater lake Haringvliet) became isolated it from other delta networks (Ysebaert et al., 2016). Thus, running the pre-barrier scenario requires extending the model domain to the entire Southwest Dutch Delta including the Rhine, Meuse, and Schelde Rivers and may lend little weight to the model performance in the model setup of this study.

[Figure]

**Figure R5: The Southwest Delta (Netherlands) with the main water basins and the main hydraulic infrastructures related to the Delta Works. Numbers indicate locations 1) Brienenoord, 2) Puttershoek, 3) Bovensluis, 4) Haringvliet center, 5) Steenbergen, 6) Zoom center, 7) Dreischor, 8) Zijpe, 9) Lodijkse Gat, 10) Hammen Oost, 11) Soelekerkepolder, 12) Hansweert. Source: Ysebaert et al., 2016.**

**References**

Idier, D., Paris, F., Le Cozannet, G., Boulahya, F., and Dumas, F.: Sea-level rise impacts on the tides of the European Shelf, Cont. Shelf Res., 137, 56–71, https://doi.org/10.1016/j.csr.2017.01.007, 2017.

Jiang, L., Gerkema, T., Wijsman, J. W., and Soetaert, K.: Comparing physical and biological impacts on seston renewal in a tidal bay with extensive shellfish culture, J. Mar. Syst., 194, 102–110, https://doi.org/10.1016/j.jmarsys.2019.03.003, 2019.

Nienhuis, P. H., and Smaal, A. C.: The Oosterschelde estuary, a case-study of a changing ecosystem: an introduction, Hydrobiologia, 282/283, 1–14, https://doi.org/10.1007/BF00024620, 1994.

Ysebaert, T., van der Hoek, D. J., Wortelboer, R., Wijsman, J. W., Tangelder, M., and Nolte, A.: Management options for restoring estuarine dynamics and implications for ecosystems: A quantitative approach for the Southwest Delta in the Netherlands, Ocean Coast. Manage., 121, 33–48, https://doi.org/10.1016/j.ocecoaman.2015.11.005, 2016.

Further issues concerning the representation of hydrodynamics are:

- Deviations of measurements and model results are rather large (up to 10 % in M4 amplitude), what about other error statistics (e.g. RMSE) and why would a further model calibration possibly not sensible?

- Response (4): We have calculated the RMSDs as shown in the Taylor Diagram in Fig. 2c. Why we do not do further calibration it that the GETM setup and settings are the same as in Jiang et al., 2019 except that the current study is a 2D barotropic run. That is, as discussed in Response (3), the only difference from the calibrated realistic run (Jiang et al., 2019) is the absence of meteorological forcing and gravitational circulation. When the wind forcing is not strong, the modeled water elevation is in good agreement with observation (Figs. 2a and 2b). Thus, most of the deviations from observation very likely originate from the missing winds and other realistic forcing, more than insufficient calibration. We have mentioned this in Section 4.1 of the revised manuscript.

  **References**

  Jiang, L., Gerkema, T., Wijsman, J. W., and Soetaert, K.: Comparing physical and biological impacts on seston renewal in a tidal bay with extensive shellfish culture, J. Mar. Syst., 194, 102–110, https://doi.org/10.1016/j.jmarsys.2019.03.003, 2019.

- The effects of tidal flats are analysed in detail but how well are they represented by the model? You might want to give a comparison of hypsometric curves of model and underlying bathymetric data.

- Response (5): Maybe this was confusing in our text. Our model does not account for the accretion and erosion of tidal flats. Rather a constant bottom topography is assumed in all scenarios. We have clarified it on Page 4 Lines 28–29. The bathymetry of tidal flats is derived from the observational data and does not change in the model run. They are the dataset measured by Rijkswaterstaat routinely and accessible online (http://opendap.deltares.nl/thredds/catalog/opendap/hydrografie/surveys/catalog.html).

- The discussion of the observed changes in tidal asymmetry in term of the interaction between the tidal wave and the basin geometry is going in the right direction, but is too short. You emphasize the importance of understanding the complicated interaction between basin geometry and tides. So why don't you give a thorough discussion on this aspect? For example, you indicate that tidal asymmetry depends on the a/h ratio as well as on the extension of intertidal area and find that ES becomes more ebb-dominant with SLR. So what does this mean with regard to the specific basin geometry of the ES and its significance for the tidal response to SLR?? Please consider further discussion on this aspect.

Response (6): Thanks for the suggestion. We have emphasized the importance of basin geometry in modulating the tides in the Oosterchelde in the revised manuscript. Based on previous studies (Friedrichs and Aubrey, 1994; Hunt, 1964; Jay, 1991; Lanzoni and Seminara, 1998; Savenije and Veling, 2005; van Rijn, 2011), we have identified the basin as a strongly convergent and less strongly dissipative basin by the amplified wave speed and tidal amplitude, the tidal wave being nearly standing waves, and the landward decreasing basin cross-sectional area shown on the newly plotted Fig. 6. This part is detailed in Section 4.1 (Page 6 Lines 19–29). In the third paragraph of Discussion (Page 9 Line 32 to Page 10 Line 5), we explained the strengthened ebb dominance under SLR with the findings of Lanzoni and Seminara (1998). Under SLR conditions in our study, friction is reduced, while convergence of basin geometry does not change since the model does not allow flooding of shorelines. When convergence becomes stronger relative to friction, the basin becomes more distorted and ebb-dominant according to Lanzoni and Seminara (1998).

**References**
Friedrichs, C. T., and Aubrey, D. G.: Tidal propagation in strongly convergent channels, J. Geophys. Res. Oceans, 99, 3321–3336, https://doi.org/10.1029/93JC03219, 1994.
Hunt J. N.: Tidal oscillations in estuaries. Geophys. J. R. Astron. Soc., 8, 440–455, https://doi.org/10.1111/j.1365-246X.1964.tb03863.x, 1964.

Jay, D. A.: Green's law revisited: Tidal long-wave propagation in channels with strong topography, J. Geophys. Res. Oceans, 96, 20585–20598, https://doi.org/10.1029/91JC01633, 1991.

Lanzoni, S., and Seminara, G.: On tide propagation in convergent estuaries. J. Geophys. Res. Oceans, 103, 30793–30812, https://doi.org/10.1029/1998JC900015, 1998.

Savenije, H. H. G., and E. J. M. Veling: Relation between tidal damping and wave celerity in estuaries, J. Geophys. Res., 110, C04007, https://doi.org/10.1029/2004JC002278, 2005.

van Rijn, L. C.: Analytical and numerical analysis of tides and salinities in estuaries; part I: tidal wave propagation in convergent estuaries, Ocean Dynam., 61, 1719–1741, https://doi.org/10.1007/s10236-011-0453-0, 2011.

-Sediment transport

One of the main results of Jiang et al. is that the ES will suffer from sediment loss with SLR. Sediment transport is not modelled with a separate sediment transport model but inferred from hydrodynamics. This approach is on the one hand a simplification but is a neat way to obtain an estimate for important impacts of SLR. These results can have an important impact on local sediment management and the work of authorities in that area. Therefore, a careful model validation and sensitivity tests for sediment transport are required to prove reliable results. The validation needs to be generally improved.

It is important to understand the complexity of hydrodynamics and at least mention potentially relevant mechanisms at first before drawing conclusions on sediment transport. For example, tidal asymmetry is not only generated by M2-M4 phase differences, but also due to hypsometric controls and lateral circulation. Please state model limitations more clearly in the discussion.

Response (7): Thanks for the insightful comment. We should use caution when making the conclusion of the sediment transport of the entire basin under SLR. The revisions regarding to $Q$ are mainly three-fold as follows.

Firstly, as the reviewer suggests, $Q$ in our study is only one component of sediment transport that is affected by tidal asymmetry. Other components include lateral transport, density-driven transport and so on. Clarification has been made in Methods (from Page 5 Line 23 to Page 6 Line 5) and Results (Page 8 Lines 11–23) that our study only aims to look at how changes in tidal asymmetry impact sediment transport, not quantifying the sediment budget with all controlling processes included. The sum of $Q$ over the entire basin that may be misunderstood as the sediment budget has been removed from Fig. 8b.

Secondly, the erosion parameter $\alpha$ and settling velocity $w_s$ in calculating $Q$ depend on the property (grain size) of sediments. When prescribing them, only one class of sand with a specific erosion parameter $\alpha$ and settling velocity $w_s$ is considered. That is to say, our study aims to focus on the hydrodynamic effects (velocity and tidal asymmetry) that can be varied by SLR and affect

sediment transport. The reason of choosing sand is that the Oosterschelde after the Delta Works is mostly sandy according to our along bottom measurements of sediment grain size distribution (Fig. R6) and a previous study (Fig. R7). That said, $Q$ is only an example of how SLR can change the asymmetry-associated transport of this type of sand. We have specified it where $Q$ is discussed in the revised manuscript. Please see from Page 5 Line 23 to Page 6 Line 5.

Thirdly, when calculating $Q$, we have changed from a constant eddy viscosity to the spatially variable eddy viscosity computed in the model. The results are updated in Fig. 7.

[Figure]

**Figure R6: The near-bottom sediment grain size distribution at (a) two stations of the Oosterschelde: (b) OS1 measured on 4 June 2019 and (c) OS7 measured on 6 June 2019. The grain size distribution is measured by the LISST-200X Particle Size Analyzer.**

[Figure]

**Figure R7: Fine sediment content < 53 pm of the subtidal bottom of the Oosterschelde after the completion of the storm-surge barrier and compartment dams. Source: Mulder and Louters, 1994.**

**References**

Mulder, J. P., and Louters, T.: Fine sediments in the Oosterschelde tidal basin before and after partial closure, Hydrobiologia, 282/283, 41–56, https://doi.org/10.1007/BF00024620, 1994.

Please consider the following issues:

- The assumption for sediment transport to be dependent on M2 and M4 amplitude and phase only is very crude as it neglects e.g. lateral circulation or density effects. Can you give any measurements for comparison?

  Response (8): We are not aware of studies comparing the effects of different processes on sediment transport in the Oosterschelde, which probably needs a sophisticated sediment model. Please see Response (7) for our revisions.

- Are your results valid for the observed range of sediment grain sizes? At least carry out a sensitivity study on assumptions made (e.g. is settling velocity ws of 1 mm/s representative for the ES)

  Response (9): Please see Response (7).

- Why is the system experiencing sediment loss today (p. 6 l.13) if there is a mixed flood/ebb dominance?

  Response (10): We have explained this in the paragraph in the revised manuscript. Please see Page 8 Lines 17–20. In the pre-barrier period, the sand erosion and sedimentation of tidal flats were in equilibrium in the Oosterschelde (Mulder and Louters, 1994). The storm surge barrier acts as a barrier of sand import and the reduced tidal currents cannot resuspend and supply sufficient sand to the eroded tidal flats, creating a sand deficit for tidal flats (Eelkema et al., 2012).

  References
  Mulder, J. P., and Louters, T.: Fine sediments in the Oosterschelde tidal basin before and after partial closure, Hydrobiologia, 282/283, 41–56, https://doi.org/10.1007/BF00024620, 1994.
  Eelkema, M., Wang, Z. B., and Stive, M. J.: Impact of back-barrier dams on the development of the ebb-tidal delta of the Eastern Scheldt, J. Coast. Res., 28, 1591–1605, https://doi.org/10.2112/JCOASTRES-D-11-00003.1, 2012.

- Results for Q are presented as averages over ES. What can we infer from that? I guess that is very difficult to interpret, e.g. you have a Q of about -0.35 kg/(m s) for an equal flood and ebb tide dominance (which would locally result in Q = 0). Please revise analysis or explain it.

  Response (11): The quantity $Q$ cannot be used as the sediment budget of the Oosterschelde given the drawbacks mentioned in Response (7). We have removed this from Fig. 8b.

- The authors estimate the net sediment transport Q inferred from the relation of Burchard et al. (2013). For a sea-level rise of 1 m Q would be around -0.8 kg/(m s). Please discuss the effect on the ES, e.g. taking the width of the ES at the mouth of about 10 km one would get an export of sediment of about 8,000 kg/s which relates with density rho 1,600 kg/m³ to a volume loss of 158 Mio. m³ p.a.. Is that sensible?

  Response (12): Please see Responses (11) and (7).

- A steady state topography is assumed for SLR which will not be the case, what kind of feedback mechanisms are to expect?

Response (13): A very good question. It is assumed that topography will not change with SLR. If it does, the spatial bottom roughness can be altered in contrast to the baseline scenario, which may change the local friction and tides. In addition, the convergence can also be changed. It will require a sediment transport and geomorphology model to study all these effects. An example is that the M2 tides in the German Bight are amplified because of the bathymetric changes (Hagen et al., 2019). We have expanded the discussion of the limitation in the last paragraph of the manuscript. Please see Page 11 Lines 18–24.

**References**
Hagen R., Freund J., and Plüß, A.: The impact of natural bathymetry changes, EGU poster, http://doi.org/10.13140/RG.2.2.13292.62083, 2019.

Even though the results needs further corroboration and discussion on model limitations I like to stress that the conclusions are presented in a clear and comprehensible way.

Response (14): Thanks for the positive comment.

Specific and technical comments

- Abstract:

  o Clearly written but last sentence is misleading. One would expect way more than just model coupling.

    Response (15): We have change "downscaling" to "coupling" as mentioned in Response (2). Now on Page 1 Line 18.

  o 'our model downscaling approach' implies a novel approach for the model setup. However, downscaling is actually not new in modelling of shelf seas and shallow coastal waters.

    Response (16): Agreed. Our point is the model coupling approach should be applied more widely in such studies instead of simply rising the sea level of a regional model. This sentence is rephrased. Now on Page 1 Lines 18–19.

- Introduction:

  o Their brief literature review is almost sufficient. However, some more reference to published work using their (coupled) modelling approach in other coastal regions would be appropriate. Are there no previous relevant studies for the ES itself?

Response (17): We have added references of coupled modeling approach in the last but one paragraph of Introduction in the revised manuscript. Please see Page 3 Lines 7–8. The previous literature on the Oosterschelde (we have changed the name "ES" to "Oosterschelde" according to another reviewer's suggestion) is described in Section 2, Study site, specifically, Page 3 Lines 15–22.

o Formally I would expect to find references in chronological order.

Response (18): The references throughout the manuscript have been changed from an alphabetical to chronological order.

o p.1 l.22 not only salt marsh accretion, but also tidal flat accretion is influenced (those two are not the same)

Response (19): We have added tidal flat accretion to the sentence. Now on Page 1 Line 28.

o p.2 l.3 refer also to Pelling et al. (2013), who focused exactly on this aspect (the decrease in tidal amplitude as a consequence of enhanced dissipation in newly inundated areas)

Response (20): It has been added. Now on Page 2 Line 9.

o p.2 l.5 Do you mean in shallow ebb-dominant estuaries?

Response (21): We doubled checked it. It should be flood-dominant here. Will explain it in the next response.

o p.2 l.5-8 A high ratio of a/h usually coincides with vast areas of intertidal flats, since large intertidal flats result in a small mean basin depth. Hence, there is NO contrast between the two aspects you describe as you say, too: seaward transport is reduced (first sentence) and there is a transition to flood-dominance (second sentence). This is the same direction of change in tidal asymmetry.

Response (22): Sorry for the confusion. We need more explanation here in describing the findings of Friedrichs and Aubrey (1988). Starting with one figure from this paper (Fig. R8b), when the ratio $a / h$ is over 0.3, the M2-M4 velocity phase difference is over 270 degrees, which is an indication of flood dominance. The higher $a / h$ is, the higher chance of flood dominance. If $a / h$ is below 0.2, all estuaries tend to be ebb dominant with a M2-M4 velocity phase difference less than 270 degrees. When $a / h$ is between 0.2 and 0.3, the tidal asymmetry depends largely on the ratio of tidal flats to channels $Vs / Vc$. Therefore, if $h$ is increasing faster than $a$ in a flood dominant system under SLR, i.e., $a / h$ decreases, the system is moving to the left, i.e., towards

less flood dominance. If *a* is increasing faster than *h*, the system become more flood dominant. With SLR, *Vs* / *Vc* will decrease, i.e., the system is moving down the y-axis, resulting in a more flood dominant situation. We have rephrased the sentence for clarification. Please see Page 2 Lines 11–16.

[Figure]

**Figure R8: Contour plots of the numerical results of 84 model systems as a function of A/H and Vs/Vc: (a) cross-sectionally averaged velocity M4/M2, amplitude ratio; (b) cross-sectionally averaged velocity 2M2 – M4, relative phase. A, tidal amplitude; H, water depth; Vs, volume of intertidal storage; Vc, volume of channels. Source: Friedrichs and Aubrey, 1988.**

**References**
Friedrichs, C. T., and Aubrey, D. G.: Non-linear tidal distortion in shallow well-mixed estuaries: a synthesis. Estuar. Coast. Shelf Sci., 27, 521–545, https://doi.org/10.1016/0272-7714(88)90082-0, 1988.

o  p.2 l.6 sentence mixed up

Response (23): "in" should be "if", but we have replaced the sentence. Please see Page 2 Lines 11–16.

o  p.2 l.7 'may cause'

Response (24): Sentence replaced.

o  o p.2 l.14 Is it really true that most modeling studies simply prescribe SLR as an additional water elevation at the seaward boundary? Please refer to some exam-ples, where it was simply added.

Response (25): Examples provided. Please see Page 3 Line 2–3.

- o p.2 l.17 Pickering et al. (2017) use a global model and thus are probably no appropriate reference for statements on the tidal propagation on shelfs.

  Response (26): Reference removed. Now on Page 3 Line 4.

- o p.2 l.23 as said in the general feedback: the downscaling method is already broadly applied for simulating tidal dynamics in coastal waters.

  Response (27): The "broad applicability" has been changed to "necessity". Now on Page 3 Line 12.

- Study site

  - o p. 3 l. 4 water depths

    Response (28): "depth" is changed to the plural form. Now on Page 4 Line 2.

- Methods

  - o p.3 l.11-12 Add some more information about the model here. Are the models 2D or 3D? And if it is 3D, what is the vertical resolution? Which quantities are con-sidered, which are relevant to density effects (transport of salt, heat)?

    Response (29): The model is 2D barotropic. It has been added to the Methods. Please see Page 4 Line 30.

  - o p.3 l.16 Better: 'The MARS domain extends to deep waters and covers the entire North-West European continental shelf...' This makes it more clear that the model not only extends to deep waters along a few sections of the open boundary, but captures the entire shelf edge.)

    Response (30): Thanks for the suggestion. The changes are made. Now on Page 4 Lines 14–16.

  - o p. 3 l. 24 reference Slangen 2014 is missing; restructure sentence

    Response (31): The reference is added to the list on Page 17 Lines 26–28. We have rephrased the sentence. Now on Page 4 Lines 23–24.

  - o p.4 l.10 Depending on the grain size, the time lag between local suspended sediment concentration and current velocity is not necessarily negligible. With regard to finer fractions (especially silt fractions) settling lag effects are important!

Response (32): Thanks for the reminder. As mentioned in Response (7), we consider sand here. This part is revised to clarify the application of *Q* in this study. Please see from Page 5 Line 33 to Page 6 Line 4.

- Results

  - p.4 l.24 The correlation coefficient is not ideal to assess a tidal signal as the tidal wave itself is a very strong signal compared to the error, therefore observed and modeled tidal water elevations always have a relatively strong correlation - also in case of rather low model accuracy. For example, you may add RMSE, which is more appropriate.

    Response (33): Thanks for the suggestion. We have shown the Taylor Diagram including correlation coefficients, root mean square deviations, and standard deviations (Fig. 2c).

  - p.5 l.8 "the main tidal patters" …of what? Tidal current velocity? Please clarify.

    Response (34): We have removed this sentence in the revised manuscript.

  - p.5 l.11-13 "With SLR, TR increases almost uniformly within ES": You may mention that this statement is related to the investigated SLR range up to 2 m and that this is not necessarily true for larger SLR values (e.g. for SLR > 2 m). (The same for p.5 l.19-21)

    Response (35): Agreed. We have the defined the range in which these relationships applies. Please see Page 7 Lines 11–12.

  - p.5 l.14 Please indicate more precisely, which r egion you mean with "adjacent North Sea"? Do you mean only the region directly located seaward from the bar-rier (up to which depth?) or the entire North Sea section within the GETM model or an even larger domain of the North Sea such as the Southern Bight?

    Response (36): Indeed unclear here. We have changed it to "the adjacent North Sea in the GETM domain as well as the Southern North Sea calculated by Idier et al. (2017)". Now on Page 7 Lines 17–18.

  - p.5 l.16-18 I don't get your point. Why do you mention the fixation of the tidal basin size when talking about the role of tidal range for tidal prism? An increased tidal range will always increase the tidal prims, no matter, if the tidal basin is fixed or not. A non-fixation would only further increase the tidal prism. Also keep in mind that as long as intertidal flats are present in the initial case (your base-line scenario) the tidal

prism will always increase with SLR, even if tidal range is not increased (remains constant), because with SLR former tidal flat volume is added to the tidal prism.

Response (37): Yes, it is true. This sentence is removed.

o p.5 l.18 Figure 7 does not show that. It looks like tidal currents are mainly increased on the tidal flats or shoals. How could this be explained?

Response (38): Fig. 7 is now Fig. 9 in the revised manuscript. Fig. 9b shows the velocity difference between scenarios with 1 m and 0 m SLR. The difference is positive at most of the basin, indicating the increase of tidal currents in most areas. We explain it in Section 4.2. The M2 phase difference between the seaward to landward ends is reduced (Fig. 5d), which means that it takes shorter time for the M2 tide to penetrate along the basin. This is a combined result of increased water depth, reduced friction, and basin convergence. Please see from Page 7 Line 33 to Page 8 Line 3.

o p.6 l.9-10 and Figure 6b: Your analysis is very difficult to justify. You made an average of residual sediment transport over the entire ES. What does that tell you about the exchange between ES and the adjacent North Sea? When the tidal asymmetry is 0, I would expect the residual sediment transport to be minimal as well, if it is related to the exchange between ES and the adjacent North Sea. Furthermore, I would expect the intensity of residual sediment transport at SLR of 0.25 m to be roughly the same as in the baseline scenario (SLR of 0), since the magnitude of tidal asymmetry of current velocity is about the same.

Response (39): We realize that the quantity $Q$ in our study should not be used as the estimate of sediment budget in the Oosterschelde for reasons given in Response (7). Therefore, we have removed the basin-integrated $Q$ in Fig. 8b, i.e., the previously Fig. 6b. Accordingly, the interpretation of $Q$ is revised in Section 4.3. Please see Page 8 Lines 11–23.

• Discussion and summary

o p.6 l.17-18 Do you have any figure as proof of the value of 30 cm for the high wa-ter at spring tide? As shown in Figure 3 the increase in TR due to a SLR of 1 m is about 0.3 m. If tidal high water increases by an extra of 0.3 m (1.3 m in total), this means that tidal low water is elevated exactly by SLR (1 m in total)? So does that mean that the increase in tidal range by 0.3 m is solely induced by the increase of tidal high water?

Response (40): The tidal range increased by 0.33 m for the 1 m SLR scenario, which is the average increase in tidal range. That is, the mean high water is increased and

mean low water is decreased by half of 0.33 m, i.e., 0.17 m. Here, we are talking about the high water at spring tide, i.e., the maximum high water, which should be the upper limit of coastal defense. Fig. R9 shows the high water at spring tide in the baseline and 1 m SLR scenarios, and their difference. Excluding the 1 m SLR, the high water is elevated for an extra 20–30 cm (Fig. R9c). We have made the clarification in the text. Please see Page 8 Lines 27–30.

[Figure]

**Figure R9: The high water during the spring tide in the baseline and (b) 1 m SLR scenarios; (c) the difference of high water between the 1 m SLR and baseline scenarios.**

- p.6 l.19-23 "turnover time": Why do you place this totally different aspect here? It is not logically connected to the rest of the text, neither to the preceding nor to the following.

  Response (41): Thanks for the comment. Maybe the text is not clear enough. The paragraph is about the impact of SLR on ecosystem functions and management strategies. Prior to this sentence, the impact of increased dike height is mentioned. The second impact is the potential changing ecosystem function. As mentioned in Section 2, the construction of the storm surge barrier almost doubled the flushing time of the basin and affected the exchange with the North Sea and ecosystem functions (e.g., shellfish culture) of the Oosterschelde. Since the tidal range may shift back to the pre-barrier level (Section 4.2), we are curious to see whether the flushing time can be reversed with SLR. This is why the sentence is placed in the paragraph. Now the paragraph is from Page 8 Line 25 to Page 9 Line 3.

- p.6 l.24 stronger tidal response [of tidal range] to SLR

  Response (42): The suggested revision is made. Now on Page 9 Line 4.

- p. 7 l. 5 'increases faster than' is unclear

  Response (43): We have removed this sentence and updated the discussion in this paragraph (Page 9 Lines 4–24).

- p.7 l.6-8 I suggest not to generalize findings from this study, because the barrier is a special geometric feature strongly affecting the tidal dynamics in the ES (just as you say it in line 21).

  Response (44): Sentence removed.

- p.7 l.11-15 A tidal basin with extensive intertidal flats actually corresponds to a high ratio of a/h. The relative importance of these two effects depends on the ratio of tidal flat area to channel area within the tidal basin. In your study site (ES) the channel area is much larger than the tidal flat area, suggesting a stronger dependence on the a/h ratio. This could explain why tidal asymmetry shifts towards ebb dominance with SLR.

  Response (45): This makes sense. Like mentioned in Response (22), the ratio $a / h$ is important. The ratio $V_s / V_c$ in the Oosterschelde is small in the western part but is large in the east. That is probably the reason of spatial variable tidal asymmetry in the basin. We have expanded the discussion on shifts to ebb dominance in the Oosterschelde under SLR. Please see Page 9 Lines 26–32.

- o p.7 l.27-28 I agree with you, but did you make any comparison to results of the shelf model (MARS)? If not, how can you conclude that shelf models are less applicable?

   Response (46): The MARS model does not have a high resolution in the Dutch Delta region. The Oosterschelde is sometimes close (Fig. R10a) and sometimes open (Fig. R10b) to Grevelingen in MARS, while in reality they are isolated by dams and sluices. The water elevation in the Oosterschelde is also not well simulated due to a low spatial resolution. For example, the water elevation in the eastern part is always around 1 m whether at low (Fig. R10a) or high (Fig. R10b) tides. Therefore, a refined local model for the Oosterschelde seems necessary.

[Figure]

**Figure R10: Snapshots of sea surface height simulated by MARS at (a) 2 Jan. 2009 14:15 and (b) 9 May 2009 14:00.**

- p.7 l.31 Are you sure that most studies on this topic neglected this aspect? I guess that most studies actually considered it.

Response (47): This is related to the comment addressed in Response (25). Our study emphasizes that it should be considered and not be neglected.

o   p. 8 l. 16 you certainly take into account 'gravitational force', do you mean tide generating forces within the model

Response (48): We have changed it to "gravitational circulation". Now on Page 11 Line 15.

- Figures
  o   The number of figures is adequate but the quality needs to be generally improved. A coastline and at least some geographic information would help.

  Response (49): We have changed the coordinates to longitude and latitude and added the coastline.

  o   Specifically: When you state Depth in m, what is the vertical reference system? Meter below NAP? IS it the same for regional and shelf model? What is the coordinate system you are using? I would prefer coordinates instead of model dimensions for the axes.

  Response (50): It is NAP. We used the Cartesian coordinate. Now we have changed them to longitude and latitude.

  o   Figure 1: Cannot see gauge locations well.

  Response (51): Fig. 1 is replotted.

  o   Figure 3: At least some geographic information would be nice, e.g. show Vlissingen from Fig. 1

  Response (52): Fig. 3 is replotted with longitude and latitude and Vlissingen is marked in the first panel. Now Fig. 4.

  o   Figure 4:e) should be amplitude not phase in the

  Response (53): Fig. 4 is replotted and corrections made. Now Fig. 5.

  o   Figure 6: references a) and b) missing

  Response (54): It is on the upper right corner. Now Fig. 8.

  o   Figure 7: show coastline / land

  Response (55): We have added the coastline. Now Fig. 9.

---

## Author Comment (AC5) · 13 Jan 2020

Mick van der Wegen (Referee)
m.vanderwegen@un-ihe.org

Dear Authors;
Thank you for your interesting contribution to the discussions on SLR impact in estuaries. I think you did some decent work in downscaling tidal dynamics and analysing the -potential- impact of SLR on the tidal dynamics. Also you explored the potential impact on sediment transport and tidal flats survival. I think this latter aspect deserves some more clarification and discussion. In the attached document I have made some minor comments. Below I formulate my major concerns. I have no doubt accepting your work when these are adequately addressed.

Response (1): Thank you for the constructive comments on our manuscript. We have revised the manuscript as requested and addressed all the comments as follows.

1) You disregard morphodynamic development. This may be a justified assumption in the sense that the morphodynamics potentially create an extra and yet unclear dimension to the work. It is good to restrict your efforts sometimes. However, I feel that there will be significant morphodynamic analysis coming 100 years in the ES.

Response (2): Thanks for the comment. Yes, we assumed a constant bottom topography mainly because of the uncertainties in the future. If the topography does change with time, the spatial bottom roughness can be altered in contrast to the baseline scenario, which may change the local friction and tides. In addition, the convergence can also be changed. It will require a sediment transport and geomorphology model to study all these effects. An example is that the M2 tide in the German Bight is amplified because of the bathymetric changes (Hagen et al., 2019). We have expanded the discussion of the limitation in the last paragraph (Page 11 Lines 15–27) of the "accept-changes" version of the revised manuscript and pointed out the limitation in the abstract (Page 1 Line 13) and conclusions (Page 12 Lines 1–3). Hereafter, wherever the page and line numbers occur, we refer to the "accept-changes" version.

**References**
Hagen R., Freund J., and Plüß, A.: The impact of natural bathymetry changes, EGU poster, http://doi.org/10.13140/RG.2.2.13292.62083, 2019.

2) You consider coarse sediment only (neglecting a settling lag) whereas muddy sediment will be relevant as well. Mud could import while sand exports. Mud could heighthen flats. A sand export could deepen channels while flats are maintained.

Response (3): It is true that the mud transport is not included in the study. We have addressed this comment in Response (13).

Drawing strong conclusions based on an analysis that disregards morphodynamics

and fine sediments seems not justified. I suggest to rephrase the summary and conclusions acknowledging more clearly that morphodynamics and fines were not considered. You may add a discussion on why you disregarded these and what important implications of that assumption could mean to your results (like the heightening of tidal flats).

Response (4): We have revised the abstract and added a conclusion section to acknowledge the assumptions and limitations of this study. For example, the exclusion of morphodynamics and fine sediments are mentioned on Page 1 Line 13 and 17, respectively. The reasons of making these assumptions are described in Methods on Page 4 Lines 28–29 and from Page 5 Line 33 to Page 6 Line 4, respectively. The assumptions are discussed on Page 11 Lines 18–24 and Page 8 Lines 20–23, respectively.

* I believe it is also the reduced tidal range (and not only the reduced sediment supply) in the ES that makes the intertidal area to erode. Wave action is more concentrated at a speciifc height (in a smaller tidal range) causing more erosion of the tidal flats.

Response (5): Thanks for the suggestion. Yes, we agree with it. We have made the changes to suggest that the reduced tidal range also contributes to the erosion of sediment in Sections 2 (Page 3 Lines 20–21) and 4.3 (Page 8 Lines 17–20).

* I am interested in how the ES is implemented in the MArs model: can you explain that a little bit more what the assumptions and implications are of the one way coupling? To what extent does the MARs model include the effect of the ES? Are the GETM boundaries far enough at sea to have no effect of the ES dynamics under SLR?

Response (6): The MARS model does not have a high resolution in the Dutch Delta region. The Oosterschelde is sometimes closed (Fig. R1a) and sometimes open (Fig. R1b) to Grevelingen in MARS, while in reality they are isolated by dams and sluices. The water elevation in the Oosterschelde is also not well simulated due to a low spatial resolution. For example, the water elevation in the eastern part is always around 1 m whether at low (Fig. R1a) or high (Fig. R1b) tides. Therefore, a refined local model for the Oosterschelde is clearly necessary.

[Figure]

**Figure R1: Snapshots of sea surface height simulated by MARS at (a) 2 Jan. 2009 14:15 and (b) 9 May 2009 14:00.**

\* I miss conclusions since these are merged in the discussion. Please differentiate into "discussion" and "summary". And maybe add sub-headings in the section that is now called "discussion and summary"

Response (7): We have added a Conclusion section in the end to summarize the major findings. Limitations and major assumptions are also ackowledged.

with kind regards
Mick van der Wegen

Please also note the supplement to this comment:
https://www.ocean-sci-discuss.net/os-2019-50/os-2019-50-RC3-supplement.pdf

Specific comments from the supplement:
Page 1 Lines 11-13: The conclusions are quite bold (even misleading) given that
1) morphodynamic adaptations are not accounted for.
2) you consider coarse sediment only whereas muddy sediment will be relevant as well
I would rephrase the conclusions as more provisional and within the limitations of your
assumptions (your decent work suggests developments under rough assumptions)

Response (8): This comment is the same as is addressed in Response (4). The abstract is
revised as suggested.

Page 1 Line 14: Cross out "model"

Response (9): It is changed to "model coupling". Now on Page 1 Line 18.

Page 2 Line 30: Actually I believe it is also the reduced tidal range in the ES that makes the
intertidal area to erode. Wave action is more concentrated (in a smaller tidal range) causing
more erosion of the tidal flats

Response (10): Thanks for the suggestion. We have made the changes. See Response (5).

Page 3 Line 26: I am interested in how the ES is implemented in the MArs model: can you
explain that a little bit more?

Response (11): This comment is the same as is addressed in Response (6).

Page 3 Line 31: Can you explain a little bit more what the assumptions and implications are
of the one way coupling? To what extent does the MARs model include the effect of the ES?
Are the GETM boundaries far enough at sea to have no effect of the ES dynamics under
SLR?

Response (12): We have added the explanation in the first paragraph of Section 3 (Page 4
Lines 9–11). One-way coupling in our application means the communication from the larger
(MARS) to smaller (GETM) domain is resolved, but not contrariwise. The MARS model
does not include the effect of the ES. It is not necessary to extend the GETM far enough. The
effect of SLR on tides at the open boundary is transferred from MARS to GETM.

Page 4 Line 10: Is this assmption valid? it only considers (coarse) sand while mud
concentrations are considerable. What are the consequences pls elaborate a little bit more.

Response (13): When calculating $Q$, only one class of sand with a specific erosion parameter $\alpha$ and settling velocity $w_s$ is considered. Our study aims to focus on the hydrodynamic effects (velocity and tidal asymmetry) that can be varied by SLR and affect sediment transport. The reason of choosing sand is that the Oosterschelde after the Delta Works is mostly sandy according to our along bottom measurements of sediment grain size distribution (Fig. R6) and a previous study (Fig. R7). That said, $Q$ is only an example of how SLR can change the asymmetry-associated transport of this type of sand. We have specified it where $Q$ is discussed in the revised manuscript (See from Page 5 Line 33 to Page 6 Line 4). It is also clarified in Section 4.3 that the mud transport is unaddressed in this study, and therefore, this study is not a quantification of sediment budget under SLR. Please see Page 8 Lines 20–23.

[Figure]

**Figure R2: The near-bottom sediment grain size distribution at (a) two stations of the Oosterschelde: (b) OS1 measured on 4 June 2019 and (c) OS7 measured on 6 June 2019. The grain size distribution is measured by the LISST-200X Particle Size Analyzer.**

[Figure]

**Figure R3: Fine sediment content < 53 pm of the subtidal bottom of the Oosterschelde after the completion of the storm-surge barrier and compartment dams. Source: Mulder and Louters, 1994.**

**References**
Mulder, J. P., and Louters, T.: Fine sediments in the Oosterschelde tidal basin before and
        after partial closure, Hydrobiologia, 282/283, 41–56,
        https://doi.org/10.1007/BF00024620, 1994.

Page 5 Line 27: I would add here your nice analysis from page 7 lines 12 to 15.

Response (14): This section covers the changes in tidal range and major components, while tidal asymmetry is not discussed. We would prefer to keep the discussion on tidal asymmetry in Section 5. Now on Page 9 Lines 27–32.

Page 6 Line 13: I think there is more to this. Apart from the lowering due to less sediment supply, an increase in tidal range will reshape the intertidal area to become steeper. The question is also if the flats can follow the SLR, which, under limited sediment supply, is indeed unlikely.
eg see Van der Wegen, M., Jaffe, B., Foxgrover, A., & Roelvink, D. (2017). Mudflat morphodynamics and the impact of sea level rise in South San Francisco Bay. Estuaries and Coasts, 40(1), 37-49.

Response (15): Thanks for the suggestion. We realize that caution should be used here to predict the fate of tidal flats under SLR given the simplification of this model study. More details such as the shape of the flats and the relative strength of erosion and sedimentation as in van der Wegen et al. (2017) should be considered. The conclusion and implication here are revised. Now on Page 8 Lines 20–23.

Page 6 Line 21: but this is in contrast to coarse sediment export. pls explain

Response (16): Thanks for the comment. This is a different process from sediment transport. As found in Jiang et al. (2019), organic matter imported from the North Sea is largely deposited by benthic fauna like a biological pump. The basin is a net sink. In contrast, sediment transport is dominated by physical processes.

**References**
Jiang, L., Gerkema, T., Wijsman, J. W., and Soetaert, K.: Comparing physical and biological impacts on seston renewal in a tidal bay with extensive shellfish culture, J. Mar. Syst., 194, 102–110, https://doi.org/10.1016/j.jmarsys.2019.03.003, 2019.

Page 7 Line 3: "narrowing", cross-sectional convergence?

Response (17): This sentence is removed.

Page 7 Line 4: "reduced; consequencely", reduced. Consequently,

Response (18): This sentence is removed.

Page 7 Line 5: "Into", behind?

Response (19): This sentence is removed.

Page 7 Lines 11-15: Maybe also explain that the reduction in a/h is apparently stronger than the reduction of Vs/Vc

Response (20): We have added the argument in the paragraph. Please see Page 9 Lines 31–32.

Page 7 Lines 11-12: "because of a stronger frictional damping during ebb" I think friedrichs explains this in a different way : A larger a/h causes the tide to propagete faster at HW than at LW; this causes a shorter flood and a tidal assymetry into flood dominance.

Response (21): Agreed. This sentence is rephrased. Now on Page 9 Lines 27–28.

Page 7 Line 12: remove "currents"

Response (22): It is changed to propagation. Now on Page 9 Line 29.

Page 7 Line 14: "less flood- and ebb-dominant state" less flood dominant and less ebb dominant

Response (23): Suggested changes made. Now on Page 9 Line 30.

Page 7 Line 14: "respectively; the shift",  respectively. The

Response (24): Suggested changes made. Now on Page 9 Lines 30–31.

Page 7 Line 25: "including" and cross-sectional convergence

Response (25): Suggested changes made. Now on Page 10 Line 19.

---

## Author Response (AR1)

John M. Huthnance (Editor)
jmh@noc.ac.uk

Dear Authors,

The three reviewers' scores indicate that your manuscript should be worth publishing in due course. However, as all three recommend "Major Revision" (and are willing to see it again) you should take their comments seriously and respond accordingly. Please incorporate your responses (perhaps summarised) in your revised manuscript so that it is self-contained and readers do not need to refer to the responses in the editorial system.

Yours sincerely,
John Huthnance

Dear Editor,

Thank you for your time and efforts in handling our manuscript.

We have completed the revision of our manuscript in response to the comments of three reviewers. Reviewer 1 suggests improving the discussion of tides and sediment transport in estuaries by including insights from previous studies, expanding the model description, calibration, and validation with more requested information and statistical analyses, and using caution to interpret our results on sediment transport. Reviewer 2 raises similar concerns on model validation, the assessment of the whole sediment budget based on the simple quantity $Q$, and the calculation of $Q$. Reviewer 3 recommends specifying and discussing two of the limitations in this study, i.e., not accounting for the changes of bed morphology in the SLR scenario and excluding muddy sediments in the calculations of $Q$.

We have made a major revision based on their comments. The Introduction and Discussion sections have been significantly improved by including the additional insight suggested by reviewers. We have also emphasized in our manuscript that our sediment transport quantity $Q$ is not a quantification of the entire sediment budget in the Oosterschelde, as reminded by all three reviewers. $Q$ is only one component of the transport processes influenced by tidal asymmetry and current velocity. Moreover, only certain sediment grain size, sand, is considered. The limitation of our estimate of $Q$ is mentioned in Abstract, Methods, and Conclusions and it is discussed in Discussion. The model description, calibration, and validation are revised substantially based on the detailed comments.

The revised manuscripts with changes tracked and accepted are returned along with our responses to the reviewer's comments point by point in blue. Please kindly consider our revised manuscript.

Sincerely,
Long Jiang
On behalf of co-authors
In this manuscript, Jiang et al. describe a nested model in which a large regional model (2 km resolution) is downscaled to an estuary in The Netherlands (the Eastern Scheldt). Sea-level scenarios are run and it is shown that tide changes are much bigger in the estuary than in the North Sea. Moreover, increasing sea-level is observed to shift the estuary towards ebb-dominated currents, with implications for sediment transport. Overall, this is an interesting paper with some interesting results. However, the analysis and discussion of estuary tides and sediment could be improved, and many of the important papers and physical insights from the last decade or so could be referenced and used to help interpret the model results. The model is incompletely described, and more error statistics and discussion of sources of uncertainty would be good. In many places there are some additional analyses that could be done that would increase the novelty of the effort. Also, sediment transport in estuaries is complicated, and one usually should not ignore density/salinity effects; therefore, would suggest that the manuscript be more careful in how implications to sediment transport are described, and perhaps frame the discussion of results in terms of hydrodynamic quantities (e.g., relative phase) that strongly suggest that important components of transport have changed.

Response (1): We appreciate the reviewer's constructive comments on our manuscript. The general comments include many specific ones that are addressed below. Specifically, 1) we have cited the papers suggested by the reviewer and included a deeper discussion of the convergent nature of the Oosterschelde and its responses to SLR. 2) The manuscript is extended to include details in model description and extra analyses (e.g., phase difference between horizontal and vertical tides, changes of M2 and M4 velocity, etc.) as suggested by the reviewer. 3) Caution is used to interpret the sediment transport quantity $Q$, which represents only one important component of sediment transport in the basin. $Q$ has been recalculated using the eddy viscosity from the model. The parameters in the $Q$ calculation are further substantiated.

**Specific Comments:**

Line 18: "Global and regional tidal regimes": While a few regional tide changes have been observed or modelled, ocean scale changes to tides have not been. Would suggest removing "global".

Response (2): Thanks for the suggestion. Pickering et al. (2017) did indicate that SLR can induce changes in tides of the global ocean, despite that these changes may be much smaller than in shelf seas and estuaries. Thus, we tend to keep the "global" according to their

findings. Now on Page 1 Line 24 of the "accept-changes" version of the revised manuscript. Hereafter, wherever the page and line numbers occur, we refer to the "accept-changes" version.

**References**
Pickering, M. D., Horsburgh, K. J., Blundell, J. R., Hirschi, J. M., Nicholls, R. J., Verlaan, M., and Wells, N. C.: The impact of future sea-level rise on the global tides, Cont. Shelf Res., 142, 50–68, https://doi.org/10.1016/j.csr.2017.02.004, 2017.

Line 23: The Chernetsky reference is 2010, not 2011.

Response (3): "2011" is changed to "2010". Now on Page 1 Line 28.

Line 25: "Nienhuis and Smaal, 1994": This is a rather old reference. Can you find a few others? There are a number of references about tidal changes and effects on currents, transport, salinity, sediment concentration, oxygen concentration, etc for estuaries such as the Ems, Gironde, Loire, Hudson, Western Scheldt, etc.

Response (4): Three references in the 2010s are added, Zhang et al., 2010; Winterwerp et al., 2013; de Jonge et al., 2014. Now on Page 1 Line 30 and Page 2 Line 1.

**References**
de Jonge, V. N., Schuttelaars, H. M., van Beusekom, J. E., Talke, S. A., and de Swart, H. E.: The influence of channel deepening on estuarine turbidity levels and dynamics, as exemplified by the Ems estuary, Estuar. Coast. Shelf Sci., 139, 46–59, https://doi.org/10.1016/j.ecss.2013.12.030, 2014.
Winterwerp, J. C., Wang, Z. B., van Braeckel, A., van Holland, G., and Kösters, F.: Man-induced regime shifts in small estuaries—II: a comparison of rivers, Ocean Dynam., 63, 1293–1306, https://doi.org/10.1007/s10236-013-0663-8, 2013.
Zhang, W., Ruan, X., Zheng, J., Zhu, Y., and Wu, H.: Long-term change in tidal dynamics and its cause in the Pearl River Delta, China, Geomorphology, 120, 209–223, https://doi.org/10.1016/j.geomorph.2010.03.031, 2010.

"ramifications for residual sediment transport and morphodynamic development": Would suggest also referencing one of the more recent papers out of the Schuttelaars group (maybe the Dijkstra paper on the Western Scheldt) they have thought a lot about tidal asymmetry. Ton Hoitink probably also has some relevant papers, if memory serves.

Response (5): Suggested references (Hoitink et al., 2003; Dijkstra et al., 2019) are included. Now on Page 2 Line 11.

**References**

Dijkstra, Y. M., Schuttelaars, H. M., and Schramkowski, G. P.: Can the Scheldt River Estuary become hyperturbid?, Ocean Dynam., 69, 809–827, https://doi.org/10.1007/s10236-019-01277-z, 2019.

Hoitink, A. J. F., Hoekstra, P., and Van Maren, D. S.: Flow asymmetry associated with astronomical tides: Implications for the residual transport of sediment, J. Geophys. Res. Oceans, 108(C10), https://doi.org/10.1029/2002JC001539, 2003.

"Tidal changes due to SLR": would suggest also referencing the very nice Ensing et al. 2015 paper. There are many other papers on the effects of SLR on tides (another relevant one is Passieri et al., 2016). Somewhere, would suggest also referencing the forthcoming review papers on tide changes Haigh et al., 2019 submitted to Annual Reviews of Geophysics, and Talke & Jay, 2020, Annual Review of Marine Science (https://www.annualreviews.org/doi/pdf/10.1146/annurev-marine-010419-010727).

Response (6): Thanks for the four suggested interesting papers, which are added as references. The review paper in Annual Review of Marine Science is particularly relevant and helpful. Now on Page 2 Lines 18–30.

**References**

Ensing, E., de Swart, H. E., & Schuttelaars, H. M.: Sensitivity of tidal motion in well-mixed estuaries to cross-sectional shape, deepening, and sea level rise, Ocean Dynam., 65, 933–950, https://doi.org/10.1007/s10236-015-0844-8, 2015.

Haigh I. D., Pickering M. D., Green J. A. M., Arbic B. K., Arns A., et al.: The tides they are a changin', Rev. Geophys. In review.

Passeri, D. L., S. C. Hagen, N. G. Plant, M. V. Bilskie, S. C. Medeiros, and K. Alizad: Tidal hydrodynamics under future sea level rise and coastal morphology in the Northern Gulf of Mexico, Earths Future, 4, 159–176, https://doi.org/10.1002/2015EF000332, 2016.

Talke, S. A., and Jay, D. A.: Changing Tides: The Role of Natural and Anthropogenic Factors, Annu. Rev. Mar. Sci., 12, 14.1–14.31, https://doi.org/10.1146/annurev-marine-010419-010727, 2020.

"without considering tidal changes in the shelf seas that may propagate into estuaries/bays.": This is a good point. However, it is also true that tide changes in a bay or estuary could affect basin tides. see the Godin paper on Bay of Fundy, and the similar papers by Arbic, Garret, et al. Would suggest also including this detail here, and acknowledging in the Methods that feedback effects into the ocean are not modeled (or are they?) with your downscaling approach.

Response (7): We did not include the feedback from the Eastern Scheldt to the North Sea in the model downscaling, but we agree with the reviewer. It is necessary to mention this here in Introduction as well as in Methods. We have included the study on tides in Bay of Fundy and the Arbic and Garret paper in the Introduction (Page 3 Line 6). It is also added in the Methods "Our study applies a one-way nesting technique that accounts for the communication from the larger (MARS) to smaller (GETM) domain, but not the other way" (Page 4 Line 9–11).

**References**

Arbic, B. K., and Garrett, C.: A coupled oscillator model of shelf and ocean tides, Cont. Shelf Res., 30, 564–574, https://doi.org/10.1016/j.csr.2009.07.008, 2010.

Ray, R. D.: Secular changes of the $M_2$ tide in the Gulf of Maine, Cont. Shelf Res., 26, 422–427, https://doi.org/10.1016/j.csr.2005.12.005, 2006.

"tidal waves on the shelf are significantly modified in amplitude and phase": would replace "are" with "can be". When there is a steep shelf (e.g., US West Coast), there isn't very much modification that occurs.

Response (8): We agree with and appreciate the suggestion. Replacement made. Now on Page 3 Line 3.

Introduction, general comment: The introduction would be improved by surveying the local changes to tides that have been observed in the North Sea but also in the Western Scheldt, the Rotterdam waterway, etc. See for example Winterwerp et al. 2013, Cai et al. 2013, Hollebrandse 2005, or van Rijn et al 2018. There is an analogy to be made between channel deepening and sea-level rise, though the analogy is not exact. See again the Ensing et al. paper for dynamical insights. There are many changes on historical dredging effects that could be referenced.

Response (9): Two good points! The first suggestion is including overview of tidal changes in surrounding systems. We introduce the North Sea studies in Section 2, the Study Site, and have expanded to incorporate the suggested examples of the Western Scheldt and Rotterdam waterway. Please see Page 3 Lines 23–27. To address the second point, an extra paragraph is added to Introduction reviewing the tidal changes caused by channel deepening and discussing the implications on SLR-induced alterations of tidal properties. Please see Page 2 Lines 18–30.

"are projected to increase mainly due to reduced friction": Isn't the changing amphidrome also a factor? Would suggest commenting on its relative importance.

Response (10): The movement of the amphidromic point is indeed important to the spatial variability of tides in the North Sea under SLR conditions. We have included the amphidrome migration here. Now on Page 3 Lines 29–30.

"tidal wave propagation can be Accelerated": not sure this is the best wording, since this would suggest constantly changing phase speed. Maybe "tidal phase speed is increased"?

Response (11): It is reworded as suggested. Now on Page 4 Line 2.

General comment: Use of acronym "ES" sometimes takes away from the understandability. you could consider just using the word Eastern Scheldt or Oosterschelde.

Response (12): "ES" is replaced with the Oosterschelde throughout the manuscript.

"MARS was forced": could you comment in the text what guided the selection of these 14 constituents? Or in particular, why just one shallow water overtide? I presume M4 was quite small at the 200m isobaths, so is there any point in having it? Would be helpful to frame/ discuss some of these issues, to help clarify the modeling methodology.

Response (13): We took into account all the constituents available in FES2004.
For the M4 component has indeed a small amplitude along the open boundaries of the MARS domain. The amplitude is smaller than 1cm along the open boundaries, except close to the Norway coast where it reached few centimeters. Thus, we could expect our results to be same without forcing the MARS model with the M4 component.
To clarify the methodology, in the manuscript, we specific that we used all the available tidal components in FES2004.

Remove the "The" in "the prescribing both water"

Response (14): The redundant "The" is removed. Please see Page 5 Line 6.

General comment: Am glad you considered variable MSL forcing on the boundary. most studies do not do that.

Response (15): Thanks for the comments.

"Every scenario was run for one year": Can you comment on the consequences of missing the Sa and SSa constituents in your boundary forcing, which are probably larger than the fortnightly constituents you did include? Would be good to state what the magnitude of these constituents are, and what sorts of biases might be introduced by not including them. Or, stated differently, what do your sea-level rise scenarios suggest about seasonal variations in tide amplitudes, and at what point will sea-level rise effects be greater than seasonal variability?

Response (16): If we understand correctly, by boundary forcing the reviewer means the boundary of the MARS model. Since our model is forced by tides only, winds and gravitational circulation that can cause seasonal and interannual variability in tides are not modeled here. Thus, the Sa and SSa should not be a substantial component.

Sa is a radiational wave (it is generated by a cyclic geophysical phenomenon other than gravitational), neglected in our study, consistently with the fact that we neglect all atmospheric forcing. In terms of magnitude, the extraction from the FES2014 database (based on tidal modeling with altimeter and tide gauge data assimilation) at point (3.5°E, 51.75°N) provides the following amplitudes for Sa, Ssa, Msf, and Mf respectively: 0.0018 m, 0.005 m, 0.021 m, 0.0047 m. It suggests that in our study area, neglecting Sa has a negligible effect, while neglecting Ssa have an effect on water level probably smaller than 1 cm. These two

components are smaller than the fortnightly constituents are. However, this is not a firm answer as FES2014 could also contains some errors.

"Vertical eddy viscosity Kv": why did you not use eddy viscosity from the model? Not sure using the same value everywhere makes sense. Also, is this a tidal average? At the very least it would be good to ascertain that your modeled eddy viscosity is consistent with this value. What I would guess is that velocity decreases quite a bit into the estuary (since velocity goes to zero at the head of tides), such that a constant eddy viscosity is a poor representation of reality. This is also a factor that will change with sea-level rise. Hence, might suggest looking into spatial patterns of eddy viscosity, and how they change with SLR. This is usually an easy output in a model, and would be something new (and would give insights into changed frictional effects).

Response (17): Thanks for the suggestion. Using the eddy viscosity from the model makes more sense in this case. Since our 2D barotropic run does not calculate eddy viscosity, we have changed the model run to a 3D run excluding winds and baroclinity. Note that the eddy viscosity is not a constant value but shows spatial variability (Fig. R1). The modeled eddy viscosity is on the order of $0.01$ m$^2$/s, as is applied in our previous computation of $Q$. Despite that, we now plug the model output spatially-variable eddy viscosity in the calculation of $Q$ and update the results in Fig. 7. The eddy viscosity does offer more insight into the hydrodynamic changes in the SLR scenarios. The viscosity over tidal flats are significantly enhanced compared to that over tidal channels. This may increase friction over tidal flats.

[Figure]

**Figure R1: (a) Eddy viscosity in the baseline scenario and (b) the ratio of eddy viscosity between the 1 m SLR and baseline scenarios.**

Constant erosion parameter. While this is used in Graewe et al 2014, is the assumption of a constant erosion parameter justified in an estuary in which sediment properties can be highly variable? Also, is this formulation valid for the cohesive sediments found in estuaries, which behave quite differently than sand? Finally, semi-analytical models in estuaries include both an erosion parameter (somewhat analogous to the one here) and an erodability parameter that is a strong function of location. This is because estuary turbidity maxima form within estuaries, changing sediment availability (i.e., some places have mud banks, others don't). Please look into and discuss more thoroughly the validity of the Graewe formulation within estuaries, and carefully frame what is not included here and what the consequences of that are. There is probably also specific information about the Eastern Scheldt that can be found in the grey literature or similar about sediment sizes, erodability, etc. that could/should be discussed and referenced to help place your results in context.

Response (18): It is true that the erosion parameter $\alpha$ and settling velocity $w_s$ depend on sediment properties. The Oosterschelde sediment after the Delta Works is predominately sandy according to our near-bottom measurements of sediment grain size distribution (Fig. R2) and a previous study (Fig. R3). Therefore, we use a bulk erosion parameter and settling velocity for sand. However, although representative, our parameter cannot be used for all grain size classes. The text before revision reads like we are quantifying the sediment transport in these scenarios, which seems too aggressive given that $Q$ is estimated in such a simplified manner.

The usage of $Q$ needs to be clarified. We are not meant to map the current and future sediment transport here but imply that changing tides can alter sediment transport, or as suggested below, an important component of sediment transport. Therefore, we use the bedload transport of sand as an example of such potential changes and are more cautious about the $Q$ description and interpretation. Please see Page 5 Line 23 to Page 6 Line 5.

[Figure]

**Fig. R2: The near-bottom sediment grain size distribution at (a) two stations of the Oosterschelde: (b) OS1 measured on 4 June 2019 and (c) OS7 measured on 6 June 2019. The grain size distribution is measured by the LISST-200X Particle Size Analyzer.**

[Figure]

**Fig. R3: Fine sediment content < 53 pm of the subtidal bottom of the Oosterschelde after the completion of the storm-surge barrier and compartment dams. Source: Mulder and Louters, 1994.**

**References**

Mulder, J. P., and Louters, T.: Fine sediments in the Oosterschelde tidal basin before and after partial closure, Hydrobiologia, 282/283, 41–56, https://doi.org/10.1007/BF00024620, 1994.

General comment: there are other types of barotropic sediment transport that can be important besides tidal asymmetry (e.g., Tidal return flow, settling lag, etc). Would look at some the papers from the Schuttelaars group. Also, can you back up the assertion that gravitational circulation, internal asymmetry (now called "ESCO": see one of the Dijkstra papers) , and other types of tidal asymmetry are not important in the Eastern Scheldt, ideally with references or measurements? If there is a salinity gradient between ocean and freshwater, then it is at least somewhat important, in some places. There should be some information on this, and the salinity structure in the estuary should be discussed/referenced.

Response (19): Thanks for the suggestion. Yes, indeed these processes may have strong influences on the sediment budget in the Oosterschelde. With negligible freshwater input, the Oosterschelde is more a tidal bay than an estuary. As mentioned above, the primary focus of the paper is more to seek the potential implications of tidal changes than to derive a complete

picture of sediment transport in the basin. We have cited these processes in our revised manuscript. Please see Page 5 Lines 23–26.

"the directional changes in residual sediment transport in different SLR scenarios.": Given the various caveats mentioned above, would frame this as sensitivity of one component of sediment transport to SLR scenarios.

Response (20): A component of sediment transport is a good suggestion. We have rephrased in our revised manuscript.

General comment, methods: Did you account for the infrastructure at the Delta Works that caused tidal amplitude to decrease 13%, as stated earlier? Am not sure a resolution of 300m would be sufficient to model any bridge piers or storm surge structures in an adequate way. However, it is essential to model this infrastructure in some way: it would be incorrect to simply increase the drag coefficient in the entire estuary as a way to obtain realistic tides. In general, some description of the inlet infrastructure would be good (It looks like an Island was built, but there must be other structures as well).

Response (21): The storm surge barrier includes two manmade islands and three openings as shown in Fig. 1. Each tidal opening consist of concrete pillars and steel gates that can be closed under severely stormy conditions (Fig. R4). The bed of the barrier is secured by a sill resulting in a much decreased water depth (Fig. R5). The pillar is around 4 m wide, which is much narrower than the grid size of the model (300 m). The model would need a much finer resolution and smaller time step to resolve the pillars, which is computationally inefficient and unpractical. Given that the pillars accounts for only 8.2% of the overall cross-sectional area based on our calculation, we reduced the depth accordingly (i.e., by ~8.2%) to maintain the realistic cross-sectional area of each tidal opening of the barrier. It turns out that our model setting reasonably captures the realistic water elevation inside the Oosterschelde (Jiang et al., 2019). We did not change the drag coefficient in the bay for calibration purposes, and the same $z_0$ was applied as in the Wadden Sea (Duran-Matute et al., 2014). The description of the storm surge barrier and bottom roughness length scale is expanded in the Methods of the revised manuscript. Please see Page 4 Line 30 to Page 5 Line 2.

[Figure]

A = sill (sediment)
B = tidal water
C = pier
D = steel gate

**Figure R4: The schematic cross-sectional view of the storm surge barrier of the Oosterschelde. Source: Nienhuis and Smaal, 1994.**

[Figure]

**Figure R5: The side view of the storm surge barrier of the Oosterschelde. Source: Wikipedia Oosterscheldekering, https://en.wikipedia.org/wiki/Oosterscheldekering.**

**References**

Duran-Matute, M., Gerkema, T., De Boer, G. J., Nauw, J. J., and Gräwe, U.: Residual circulation and freshwater transport in the Dutch Wadden Sea: a numerical modelling study, Ocean Sci., 10, 611–632, https://doi.org/10.5194/os-10-611-2014, 2014.

Jiang, L., Gerkema, T., Wijsman, J. W., and Soetaert, K.: Comparing physical and biological impacts on seston renewal in a tidal bay with extensive shellfish culture, J. Mar. Syst., 194, 102–110, https://doi.org/10.1016/j.jmarsys.2019.03.003, 2019.

Nienhuis, P. H., and Smaal, A. C.: The Oosterschelde estuary, a case-study of a changing ecosystem: an introduction, Hydrobiologia, 282/283, 1–14, https://doi.org/10.1007/BF00024620, 1994.

A related note: Did not see any information about model calibration in section 3, even though section 3 promised (first paragraph) to discuss calibration. Information about tide stations used (and where to find data), statistics about root mean square error (for the different constituents), and so on is needed to assess how well the model is performing. Some of this information is given at the start of section 4, but it would be good to expand this.

Response (22): We have added the last but one paragraph to include the suggested information in Section 3 and the calibration results in Section 4.1. Please see Page 5 Lines 12–19 and Page 6 Lines 8–18.

General Comment: Was there wetting/drying in the model? This is very important for bathymetries in which there are intertidal flats, as there are here. For example, it can alter tidal amplitudes and tidal velocities. Please discuss whether you have wetting/drying, and the consequences if you do not (based off of known literature).

Response (23): Yes, the model resolves wetting/drying for tidal flats. We have added the description in Methods. Please see Page 5 Lines 4–5.

Section 4
Figure 2:Could you somewhere discuss the relative phase of the water levels (2M2 – M4) in your model, vs. the measured relative phase? This will give some indication about whether you are getting the tidal asymmetry correct.

Response (24): Good point. We have calculated the M2-M4 phase difference of the vertical tides and compared with the observations. Our model represents the direction of tidal asymmetry correctly despite overestimating the extent of flood and ebb dominance. We added a Fig. 3b for illustration. Please also see Page 6 Lines 13–18.

Also, it would be useful if your discussion of the calibration discerns between errors at the ocean boundary and errors that are produced within the estuary. In other words, can you discern between the "external M4" and the "internal M4", as in Chernetsky et al. 2010? In that vein, in might be useful to extend your calibration and discussion to coastal gauges that are outside the estuary (e.g., Den Helder, Vlissingen, and some other nearby coastal gauges). Having only 3 calibration points is a rather small sample size, especially since the wider domain encompasses many tide gauges. It would be useful to know how well the larger model is doing (with comparison statistics).

Response (25): The validation of the MARS model is mostly done by Idier et al. (2017). We are here extracting the validation results (Figs. R6 and R7) from this paper, which indicate that the European Shelf model simulates the water elevation and major tidal components reasonably well.

It would be difficult to completely decompose internally and externally generated errors. We made an effort to run a scenario in which tides from the 1-m SLR MARS model were prescribed to the open boundary but the sea level stays the same as the baseline. The M4 tide from this run was compared from the baseline run. We can see that M4 difference from the

boundary hardly penetrates into the bay (Fig. R8). Therefore, it is likely that internal M4 dominates in the bay.

[Figure]

Figure R6: Computational domain of the MARS model with locations of tide gauges used to validate the model. Source: Idier et al., 2017.

[Figure]

(a)

(b)

Figure R7: (a) Modeled versus data-based highest annual tide (for the year 2009) for each site. The numbers refer to the tide gauge names in Fig. R6; (b) modeled versus data-based tidal component amplitudes for 5 selected tidal components. + and • indicate

**sites with relative errors for the highest tide that are smaller or larger than 5%, respectively. Source: Idier et al., 2017.**

[Figure]

**Figure R8: The difference in M4 amplitude in the GETM domain between a hypothetical model run and the baseline run. In the hypothetical run, the tides in the 1-m SLR scenario of the MARS model was prescribed to the GETM open boundary, while the sea level stayed the same as the baseline run.**

**References**

Idier, D., Paris, F., Le Cozannet, G., Boulahya, F., and Dumas, F.: Sea-level rise impacts on the tides of the European Shelf, Cont. Shelf Res., 137, 56–71, https://doi.org/10.1016/j.csr.2017.01.007, 2017.

General Comment: Can you let us know what the phase between tidal velocity and tidal elevation is at different locations (e.g., for M2), and discuss implications? The phase provides insight into whether there is a Stokes Drift and an associated return flow (see e.g., Moftakhari et al. 2016).

Response (26): The phase difference between horizontal and vertical tides is close to 90º based on our calculation (Fig. R9). Thus, the tidal waves are mostly standing waves and Stokes Drift should be limited. The horizontal and vertical M2 phase difference helps us understand the tidal asymmetry calculated from both water elevation and tidal currents. We have included this in the last but one paragraph of Section 4.1. Please see Page 6 Lines 23–25.

[Figure]

**Figure R9: The M2 phase difference between horizontal and vertical tides (horizontal −
vertical) on a transect from the west to the east of the Oosterschelde.**

These results are interesting. However, projecting into the future is fundamentally a
counterfactual: it's a "what if" scenario that cannot (yet) be proven, yet depends a lot on the
assumptions made in the future projection (flooding vs. no flooding, for example, or the
assumption of no morphological change). Also, would argue that the modeled future tides
depend a lot on how friction was modeled in the estuary, whether and how wetting/drying is
included, etc. Further, small scale infrastructure (tide-gates) and small scale channels might
(and probably do) matter. Some discussion of such uncertainties is needed. Again, the Ensing
et al. paper has some insights.

Obviously one cannot include everything, and the comment above doesn't just pertain to this
paper. However, can you think of ways to address what the consequences of various
modeling decisions are, and discuss how they impact results? For example, how might trends
with MSL change if friction is changed by +/- 10%? What would be the consequence of
random perturbations in bathymetry, or if only the channels (but not the flats) get deeper (i.e,
an assumption of partial morphodynamic adjustment)? Finally, might suggest running the
model with and without the storm surge barrier infrastructure, to see if your model is able to
approximate the historical change to the model. As argued in the Talke & Jay review and
references therein, doing a retrospective model run is helpful in terms of making sure that
your model can at least reproduce past trends (thus increasing confidence in future trends).

Response (27): We agree that predicting the future conditions are counterfactual relying on
many assumptions, uncertainties, and unknowns. These uncertainties should be

acknowledged and discussed in the manuscript. Our manuscript deals more with changes occurring by the "knowns", such as the regional SLR projected by Slangen et al. (2014) and the SLR-induced tidal changes in the shelf seas by Idier et al. (2017), rather than predicting the impacts of the "unknowns".

The concerns about the flooding and non-flooding setting is partially addressed in Response (23). The model setting is based on reality. The coastlines surrounding the Oosterschelde are protected by dikes where flooding is not allowed. The tidal flats can be flooded and exposed during tidal cycles. We discussed this in the last but one paragraph of Section 5 (Page 11 Lines 1–14).

In our study, a spatially constant bottom roughness length scale $z_0$ of 0.0017 m was used following Duran-Matute et al., 2014, which is added to Methods. How the bottom roughness will be affected by changing bed morphology in the future is highly unpredictable and thus we use the same $z_0$ in all SLR scenarios. This concern is recognized as one of the limitations of this study. Changing friction itself by ±10% seems unlikely because friction is nonlinearly related to turbulence and bottom roughness. To decipher how friction is influenced by bed forms and interacted with tidal currents, studies such as Cheng et al. (1999) and Prandle (2004) are necessary in the Oosterschelde. We have mentioned this limitation in the last paragraph of Discussion. Please see Page 11 Lines 18–21. The impacts of anthropogenic perturbation of the regional sea floor is also discussed in this paragraph referring to Ensing et al. (2015). Please see Page 11 Lines 24–27.

Running the pre-barrier simulation for additional model validation is a good suggestion. However, the current Oosterschelde is a completely different system from the pre-barrier times not only because of the storm surge barrier. As part of the Delta Works in the late 1980s (https://en.wikipedia.org/wiki/Delta_Works), many other dams and sluices (Fig. R10) were built at approximately the same time cutting the freshwater input of the Oosterschelde, which along with other systems (e.g., the salt-water lake Grevelingen and freshwater lake Haringvliet) became isolated from other delta networks (Ysebaert et al., 2016). Thus, running the pre-barrier scenario requires extending the model domain to the entire Southwest Dutch Delta including the Rhine, Meuse, and Schelde Rivers and may lend little weight to the model performance in this study.

[Figure]

**Figure R10: The Southwest Delta (Netherlands) with the main water basins and the main hydraulic infrastructures related to the Delta Works. Numbers indicate locations 1) Brienenoord, 2) Puttershoek, 3) Bovensluis, 4) Haringvliet center, 5) Steenbergen, 6) Zoom center, 7) Dreischor, 8) Zijpe, 9) Lodijkse Gat, 10) Hammen Oost, 11) Soelekerkepolder, 12) Hansweert. Source: Ysebaert et al., 2016.**

"under SLR the M4 amplitude decreases outside, while it increases inside ES" – Please explain why.

Response (29): As shown in Fig. R8, the decrease in M4 amplitude in the North Sea is primarily a result of weakened M4 in the larger domain. The external M4 hardly penetrates into the bay. Therefore, the increased M4 amplitude in the basin is mainly a result of stronger tidal distortion.

"Tidal waves in shallow waters propagate at a speed of sqrt(gh)": Actually, this is true only in the inviscid case (i.e., not your case). Would modify your text. Note that friction and convergence can strongly alter the phase speed. For your case, which is most likely weakly convergent and strongly (or moderately) frictional, would expect the phase speed to be somewhat less than sqrt(gh). Would suggest figuring out where in the parameter space mentioned above you are (e.g., by estimating your phase speed or by scaling), and discuss (the phase between velocity and water level also gives you an indication). In general, please look into the literature (e.g., Jay 1991, Friedrichs & Aubrey 1994, Lanzoni & Seminara 1998,

and the many other idealized tide models) and discuss the processes in more detail, and how they affect results.

Response (30): Good suggestion that helps improve the manuscript. It turns out that the Oosterschelde is dominated by convergence more than friction. The main arguments include the greatly decreasing cross-sectional area landwards (Fig. 6), the increasing tidal amplitude landwards (Fig. 4a), a much faster M2 speed (22.7 m/s on average) than $(gh)^{0.5}$ = 8.3 m/s (Section 4.1), and the nearly standing wave of tides (Fig. R7). We have added these results in Section 4.1 (Page 6 Lines 19–29) and discussed our system as a convergent basin among references (Friedrichs and Aubrey, 1994; Hunt, 1964; Jay, 1991; Lanzoni and Seminara, 1998; Savenije and Veling, 2005; van Rijn, 2011). These analyses improve the interpretation of how the system evolves under SLR in Sections 4.2 (e.g., Page 7 Line 33 to Page 8 Line 3) and 5 (e.g., Page 9 Lines 19–21 and 32–33 and Page 10 Lines 1–5).

"decline in bottom friction favors faster wave propagation": without explanation, this doesn't make sense. See comment above on frictional effects.

Response (31): The text here is revised according to the findings in Response (30). Now from Page 7 Line 32 to Page 8 Line 3.

General comment: To what extent is reflection of the tide wave important? Do you see evidence of resonance, e.g., in the phase plots (in near resonance you get a fast phase speed)? It would seem that in addition to changes in friction (and convergence) caused by depth changes, you may have changes in reflection or partial reflection. See e.g., Winterwerp et al., 2013, Familkhalili & Talke 2016, or Ralston et al., 2019. In reflective estuaries, the biggest change in tides is usually seen at the boundary; in estuaries where depth/friction changes matter most and reflection doesn't occur, the maximum tidal change is seen in mid-estuary (see again the Talke & Jay 2020 review). Some discussion on resonance is found later, I see, but some more close analysis is possible. One other idea would be to scale the relative

importance of the convergence term and the friction term, to see if the rise in tide amplitude at the end of the estuary is due to friction that is weaker than convergence (e.g., Friedrichs & Aubrey, 1994).

Response (32): Thanks for the comment. Since the Oosterschelde is an amplifying basin, the phase speed is actually faster than the frictionless wave speed $(gh)^{0.5}$. The quarter-wavelength resonance period is 2 hours, even shorter than the calculated 5.5 hours. Therefore, it is further away from the M2 period. Thus, resonance is even less important than we first thought. The calculation is update in Section 5. Please see Page 9 Lines 7–16.

General comment: Please explain why a transition to ebb dominance occurs. Perhaps the Friedrichs & Aubrey 1988 and Friedrichs & Madsen papers might have some insights.

Response (33): We have discussed the SLR-induced transition to ebb dominance in the paragraph from Page 9 Line 25 to Page 10 Line 13. The changes in tidal asymmetry caused by SLR depends on the competing effect of reduced friction versus submerged tidal flats (Friedrichs et al., 1990). The study by Lanzoni and Seminara (1998) offers insight into strongly convergent and less strongly dissipative basins such as the Oosterschelde. Under SLR conditions in our study, friction is reduced, while convergence of basin geometry does not change since the model does not allow flooding of shorelines. When convergence becomes stronger relative to friction, the basin becomes more distorted and ebb-dominant according to Lanzoni and Seminara (1998).

"The quantity Q is used to estimate the combined effects of tidal current velocity and asymmetry". Before looking at Q, wouldn't it make sense to also plot out the M2 and M4 tidal currents (much like the amplitude plots)? It might also be interesting to see if the tidal ellipse change at all.

Response (34): We have plotted the tidal currents and its change with SLR (Fig. 9). Below we plotted the M2 and M4 velocity along the channel shown in Fig. R9. Only the major axis of the tidal ellipse is shown (Fig. R11) since the minor axis is several orders of magnitude smaller. SLR slightly increase the M2 and M4 velocity (Fig. R11), which is similar to the overall velocity in Fig. 9.

[Figure]

**Figure R11: The major axis of the (a) M2 and (b) M4 tidal ellipses along the transect shown in Fig. R7 in the baseline and 1-m SLR scenarios.**

"the residual transport more than doubles" Again, would be careful about calling "Q" the residual transport. It's perhaps one type of residual barotropic transport, amongst many.

Response (35): We have removed this sentence and $Q$ in Fig. 8b. Through the comments of reviewers, we realize that $Q$ is only one component of tidal transport among many and only considers a typical class of sand. So, it was inappropriate to say $Q$ is the overall residual transport. The purpose of calculating $Q$ is to show that changes in tidal asymmetry may very likely alter the direction of sediment transport in the Oosterschelde. We have revised the entire manuscript to prevent the misleading thoughts that we quantify the overall sediment budget.

"this will not be accompanied by sufficient net sediment import as was in the past" Check grammar of this clause. Would also caution, again, about assuming that this is the only relevant source of transport. All coastal-plain estuaries that I've ever seen have a so-called estuary turbidity maximum that is caused by upstream transport. This is because baroclinic effects (ESCO, gravitational circulation) and settling lag effects are often so important. The paper would be helped by reviewing what is known about ETMs somewhere, both in general and in nearby estuaries (or ideally the Eastern Scheldt). The results presented here (and the way they are framed) would suggest that no ETM forms, which is probably not the case and would likely be greeted with skepticism in the ETM community. For references, see the Burchard et al. 2018 review and references therein.

Response (36): This sentence has been rephrased. We have also clarified that this is just one form of sediment transport. Please see Page 8 Lines 14–23. The Oosterschelde is more a well-mixed tidal bay than an estuary, because it receives negligible freshwater runoff after the Delta Works (Nienhuis and Smaal, 1994; Ysebaert et al., 2016). Gravitational circulation in the Oosterschelde is not as important as in a true estuary like the Western Scheldt. To our knowledge, no studies have found an ETM in the Oosterschelde.

Response (45): These references are added. Now on Page 11 Lines 18.

Figure 1: The surge barrier should be labeled, not just shown with an ellipse. In general, it would be more helpful to describe exactly how much of the channel crosssection is impeded by the storm surge barrier, and how this is modeled.

Response (46): Thanks for the suggestions. We have added the description of the storm surge barrier and how it is modeled in Methods. See Response (21) for details.

Figure 2: Can you explain why only these specific days of tidal modeling are shown? Without explanation it could be interpreted as "cherry picking" a period of time where the fit was good. In general, more statistics on calibration would be good.

Response (47): The editor asked the same question before. We selected a period of weak wind forcing (Fig. R12), as explained in Section 4.1. We have also expanded the calibration with more statistics. See Section 4.1 and Fig. 2 for revisions.

[Figure]

**Figure R12: Wind magnitude over the Eastern Scheldt in the year 2009 (Data source: Royal Netherlands Meteorological Institute). The period of days 175-185 as shown in Figs. 2a and 2b is marked with read dashed lines.**

Figure 3: How are you defining tidal range? There are different ways of doing that, so please specify.

Response (48): Tidal range is defined as the difference between high and low waters in every tidal cycle. Fig. 3 (now Fig. 4) shows the annual average of tidal ranges in all tidal cycles. We have added the definition in the caption.

Figure 4: The effect of the Delta Works is quite stark. Is there an effect of changing inlet cross-sectional area, i..e, as in Passieri et al. 2016? (That paper found variable changes to tides in back-barrier bays of the Gulf of Mexico. See also the Talke & Jay 2020 review for discussion on and references for the "inlet choking effect".

Response (49): Good point. However, in our study, the manmade islands and sill are not flooded by SLR, so it is not the case as in Passieri et al., 2016. We mentioned the choking effect on Page 9 Lines 21–24.

Figure 7: please provide information on how annual average was calculated. Is this based on peak velocity, rms velocity, average of the absolute value, or something else?

Response (50): This is root mean square current speed $(u^2 + v^2)^{0.5}$. We have specified it in the caption. Now Fig. 9.
The paper by Jiang et al. presents a modelling study of a tidal bay, here the Eastern Scheldt, on the effects of sea-level rise on hydrodynamics and sediment transport. The study addresses an issue, which is of practical relevance to sediment management especially when considering future sea level rise.

Overall the paper is clearly structured and written in an easily comprehensible but precise way. The authors apply results from a European Shelf model to force a regional Eastern Scheldt model with different boundary conditions representing present-day and sea level rise (SLR) conditions. From these hydrodynamic modelling results they infer effects on sediment transport based on a simplified approach just taking the effect of M2 and M4 tides into account following Burchard et al. (2013). The authors conclude that the Eastern Scheldt will change with SLR from balanced flood-ebb current conditions to ebb dominance which will result in a loss of sediments of the Eastern Scheldt.

The results of Jiang et al. add an interesting case study to previous assessments of SLR effects on coastal regions. Therefore, their work is relevant to the scope of Ocean Science. The modelling study applies state of the art techniques of model coupling to combine large-scale changes of hydrodynamics due to SLR and regional dynamics of a tidal bay.

Based on their modelling results the authors draw the conclusion that the ES will change with SLR from balanced flood-ebb current conditions to ebb dominance and thus a potential loss of sediments. I find it hard to assess the model validity based on the presented results. I recommend a substantial improvement of model validation.

I appreciate the approach to infer sediment transport from hydrodynamic quantities; here the ratio of M2 and M4 tides, but this certainly requires to carefully checking if results are consistent and plausible. Here the authors have to spent considerable more effort.

In view of the complexity of hydrodynamic and sediment transport processes in tidal environments, the paper's very simplified approach to asses residual sediment transport lacks a critical discussion of results.

Response (1): Thanks for the constructive and detailed comments on our manuscript. We have revised the manuscript accordingly. Specifically, we improved the model validation by including additional statistical analysis and expanded the discussion as suggested. We have also interpreted the results on the sediment transport quantity $Q$ with more caution given the limitations and simplifications of this method. Please see the following responses for detailed revisions.

**Scientific comments**

a) Numerical modelling

- General

Title and wording in the article are misleading when it comes to the term 'downscaling'. The authors imply that the apply a new approach but their methods are neither new nor would I call it downscaling per se. However, their approach is appropriate to address the scientific question how tidal dynamics of the Eastern Scheldt (ES) will change with SLR. The term 'downscaling' in the title is in my view misleading but scope and methods are clarified in the abstract.

Response (2): We have changed "downscaling" to "model coupling" in the title and the text, where necessary. It is indeed not a new approach. Our main point is that it seems an appropriate solution compared to considering SLR only but not taking tidal changes on the shelf into account. It is clarified in Introduction (e.g, from Page 2 Line 34 to Page 3 Line 8 of the "accept-changes" version of the revised manuscript) and Discussion (e.g., Page 10 Lines 22–34 of the "accept-changes" version of the revised manuscript). Hereafter, wherever the page and line numbers occur, we refer to the "accept-changes" version.

- Hydrodynamics

One of the strong points of the paper is the analysis of possible resonance effects even though they do not prove to be important here. A shortcoming of the modelling work is the model validation. This holds for the general approach: not taking meteorological forcing into account is not well suited for a proper hydrodynamic model validation. This is especially important as the authors find that the presence of the storm surge barrier is relevant for SLR induced changes. Unfortunately the storm surge barrier is not resolved therefore they have to proof that they can cover the effect under sea-level rise conditions properly. Please carry out a comparison of model results and measurements outside of the ES to show that the model captures the transformation due to the storm surge barrier narrowing. Moreover, I would suggest carrying at least out a sensitivity study for high mean water levels in the North Sea but for an open storm surge barrier in order to assess if the model adequately resolves higher water levels in the ES.

Response (3): Thanks for the comments. Please see the following four aspects addressing the comment.

1) About the missing meteorological forcing, the GETM setup for the Oosterschelde has been calibrated and validated with realistic meteorological forcing and gravitational circulation in our previous study (Jiang et al., 2019). In the current study, the key point is to simulate the "future" SLR scenarios. It is difficult to include predicted meteorological forcing and river runoff for the future. Thus, both nested models MARS and GETM are run without meteorological forcing. Despite the absence of meteorological forcing in the model, the

coupled model is able to simulate the water elevation with reliable performance, based on which scenario the further SLR runs were conducted.

2) The storm surge barrier is not unresolved. In Methods of the revised manuscript (from Page 4 Line 30 to Page 5 Line 2), we have clarified how it was implemented in our model. The storm surge barrier includes two manmade islands and three openings as shown in Fig. 1. Each tidal opening consists of concrete pillars and steel gates that can be closed under severes stormy conditions (Fig. R1). The bed of the barrier is secured by a sill resulting in a much decreased water depth (Fig. R2). The pillar is around 4 m wide, which is much narrower than the grid size of the model (300 m). The model would need a much finer resolution and smaller time step to resolve the pillars, which is computationally inefficient and unpractical. Given that the pillars accounts for only 8.2% of the overall cross-sectional area based on our calculation, we reduced the depth accordingly (i.e., by ~8.2%) to maintain the realistic cross-sectional area of each tidal opening of the barrier. It turns out that our model setting reasonably captures the realistic water elevation inside the Oosterschelde (Jiang et al., 2019).

[Figure]

A = sill (sediment)
B = tidal water
C = pier
D = steel gate

**Figure R1: The schematic cross-sectional view of the storm surge barrier of the Oosterschelde. Source: Nienhuis and Smaal, 1994.**

[Figure]

**Figure R2: The side view of the storm surge barrier of the Oosterschelde. Source: Wikipedia Oosterscheldekering, https://en.wikipedia.org/wiki/Oosterscheldekering.**

3) The calibration in the North Sea was conducted by Idier et al. (2017). To show the calibration results, we are here extracting the validation results (Figs. R3 and R4) from this paper, which indicate that the European Shelf model simulates the water elevation and major tidal components reasonably well.

[Figure]

**Figure R3: Computational domain of the MARS model with locations of tide gauges used to validate the model. Source: Idier et al., 2017.**

[Figure]

**Figure R4: (a) Modeled versus data-based highest annual tide (for the year 2009) for each site. The numbers refer to the tide gauge names in Fig. R6; (b) modeled versus data-based tidal component amplitudes for 5 selected tidal components. + and • indicate sites with relative errors for the highest tide that are smaller or larger than 5%, respectively. Source: Idier et al., 2017.**

4) A sensitivity test without the storm surge barrier is a good suggestion but will not help the model validation. The current Oosterschelde is a completely different system from the pre-barrier (i.e., without the storm surge barrier) times not only because of the storm surge barrier. As part of the Delta Works in the late 1980s (https://en.wikipedia.org/wiki/Delta_Works), many other dams and sluices (Fig. R5) are built at approximately the same time cutting the freshwater input of the Oosterschelde, which along with other systems (e.g., the salt-water lake Grevelingen and freshwater lake Haringvliet) became isolated it from other delta networks (Ysebaert et al., 2016). Thus, running the pre-barrier scenario requires extending the model domain to the entire Southwest Dutch Delta including the Rhine, Meuse, and Schelde Rivers and may lend little weight to the model performance in the model setup of this study.

[Figure]

**Figure R5: The Southwest Delta (Netherlands) with the main water basins and the main hydraulic infrastructures related to the Delta Works. Numbers indicate locations 1) Brienenoord, 2) Puttershoek, 3) Bovensluis, 4) Haringvliet center, 5) Steenbergen, 6) Zoom center, 7) Dreischor, 8) Zijpe, 9) Lodijkse Gat, 10) Hammen Oost, 11) Soelekerkepolder, 12) Hansweert. Source: Ysebaert et al., 2016.**

- The discussion of the observed changes in tidal asymmetry in term of the interaction between the tidal wave and the basin geometry is going in the right direction, but is too short. You emphasize the importance of understanding the complicated interaction between basin geometry and tides. So why don't you give a thorough discussion on this aspect? For example, you indicate that tidal asymmetry depends on the a/h ratio as well as on the extension of intertidal area and find that ES becomes more ebb-dominant with SLR. So what does this mean with regard to the specific basin

geometry of the ES and its significance for the tidal response to SLR?? Please consider further discussion on this aspect.

Response (6): Thanks for the suggestion. We have emphasized the importance of basin geometry in modulating the tides in the Oosterchelde in the revised manuscript. Based on previous studies (Friedrichs and Aubrey, 1994; Hunt, 1964; Jay, 1991; Lanzoni and Seminara, 1998; Savenije and Veling, 2005; van Rijn, 2011), we have identified the basin as a strongly convergent and less strongly dissipative basin by the amplified wave speed and tidal amplitude, the tidal wave being nearly standing waves, and the landward decreasing basin cross-sectional area shown on the newly plotted Fig. 6. This part is detailed in Section 4.1 (Page 6 Lines 19–29). In the third paragraph of Discussion (Page 9 Line 32 to Page 10 Line 5), we explained the strengthened ebb dominance under SLR with the findings of Lanzoni and Seminara (1998). Under SLR conditions in our study, friction is reduced, while convergence of basin geometry does not change since the model does not allow flooding of shorelines. When convergence becomes stronger relative to friction, the basin becomes more distorted and ebb-dominant according to Lanzoni and Seminara (1998).

-Sediment transport

One of the main results of Jiang et al. is that the ES will suffer from sediment loss with SLR. Sediment transport is not modelled with a separate sediment transport model but inferred from hydrodynamics. This approach is on the one hand a simplification but is a neat way to obtain an estimate for important impacts of SLR. These results can have an important impact on local sediment management and the work of authorities in that area. Therefore, a careful

model validation and sensitivity tests for sediment transport are required to prove reliable results. The validation needs to be generally improved.

It is important to understand the complexity of hydrodynamics and at least mention potentially relevant mechanisms at first before drawing conclusions on sediment transport. For example, tidal asymmetry is not only generated by M2-M4 phase differences, but also due to hypsometric controls and lateral circulation. Please state model limitations more clearly in the discussion.

Response (7): Thanks for the insightful comment. We should use caution when making the conclusion of the sediment transport of the entire basin under SLR. The revisions regarding to $Q$ are mainly three-fold as follows.

Firstly, as the reviewer suggests, $Q$ in our study is only one component of sediment transport that is affected by tidal asymmetry. Other components include lateral transport, density-driven transport and so on. Clarification has been made in Methods (from Page 5 Line 23 to Page 6 Line 5) and Results (Page 8 Lines 11–23) that our study only aims to look at how changes in tidal asymmetry impact sediment transport, not quantifying the sediment budget with all controlling processes included. The sum of $Q$ over the entire basin that may be misunderstood as the sediment budget has been removed from Fig. 8b.

Secondly, the erosion parameter $\alpha$ and settling velocity $w_s$ in calculating $Q$ depend on the property (grain size) of sediments. When prescribing them, only one class of sand with a specific erosion parameter $\alpha$ and settling velocity $w_s$ is considered. That is to say, our study aims to focus on the hydrodynamic effects (velocity and tidal asymmetry) that can be varied by SLR and affect sediment transport. The reason of choosing sand is that the Oosterschelde after the Delta Works is mostly sandy according to our along bottom measurements of sediment grain size distribution (Fig. R6) and a previous study (Fig. R7). That said, $Q$ is only an example of how SLR can change the asymmetry-associated transport of this type of sand. We have specified it where $Q$ is discussed in the revised manuscript. Please see from Page 5 Line 23 to Page 6 Line 5.

Thirdly, when calculating $Q$, we have changed from a constant eddy viscosity to the spatially variable eddy viscosity computed in the model. The results are updated in Fig. 7.

[Figure]

**Figure R6: The near-bottom sediment grain size distribution at (a) two stations of the Oosterschelde: (b) OS1 measured on 4 June 2019 and (c) OS7 measured on 6 June 2019. The grain size distribution is measured by the LISST-200X Particle Size Analyzer.**

[Figure]

**Figure R7: Fine sediment content < 53 pm of the subtidal bottom of the Oosterschelde after the completion of the storm-surge barrier and compartment dams. Source: Mulder and Louters, 1994.**

Even though the results needs further corroboration and discussion on model limitations I like to stress that the conclusions are presented in a clear and comprehensible way.

Response (14): Thanks for the positive comment.

Specific and technical comments

- Abstract:

  o Clearly written but last sentence is misleading. One would expect way more than just model coupling.

    Response (15): We have change "downscaling" to "coupling" as mentioned in Response (2). Now on Page 1 Line 18.

  o 'our model downscaling approach' implies a novel approach for the model setup. However, downscaling is actually not new in modelling of shelf seas and shallow coastal waters.

    Response (16): Agreed. Our point is the model coupling approach should be applied more widely in such studies instead of simply rising the sea level of a regional model. This sentence is rephrased. Now on Page 1 Lines 18–19.

- Introduction:

  o Their brief literature review is almost sufficient. However, some more reference to published work using their (coupled) modelling approach in other coastal regions would be appropriate. Are there no previous relevant studies for the ES itself?

    Response (17): We have added references of coupled modeling approach in the last but one paragraph of Introduction in the revised manuscript. Please see Page 3 Lines 7–8. The previous literature on the Oosterschelde (we have changed the name "ES" to "Oosterschelde" according to another reviewer's suggestion) is described in Section 2, Study site, specifically, Page 3 Lines 15–22.

  o Formally I would expect to find references in chronological order.

    Response (18): The references throughout the manuscript have been changed from an alphabetical to chronological order.

o   p.1 l.22 not only salt marsh accretion, but also tidal flat accretion is influenced (those two are not the same)

    Response (19): We have added tidal flat accretion to the sentence. Now on Page 1 Line 28.

o   p.2 l.3 refer also to Pelling et al. (2013), who focused exactly on this aspect (the decrease in tidal amplitude as a consequence of enhanced dissipation in newly inundated areas)

    Response (20): It has been added. Now on Page 2 Line 9.

o   p.2 l.5 Do you mean in shallow ebb-dominant estuaries?

    Response (21): We doubled checked it. It should be flood-dominant here. Will explain it in the next response.

o   p.2 l.5-8 A high ratio of a/h usually coincides with vast areas of intertidal flats, since large intertidal flats result in a small mean basin depth. Hence, there is NO contrast between the two aspects you describe as you say, too: seaward transport is reduced (first sentence) and there is a transition to flood-dominance (second sentence). This is the same direction of change in tidal asymmetry.

    Response (22): Sorry for the confusion. We need more explanation here in describing the findings of Friedrichs and Aubrey (1988). Starting with one figure from this paper (Fig. R8b), when the ratio $a / h$ is over 0.3, the M2-M4 velocity phase difference is over 270 degrees, which is an indication of flood dominance. The higher $a / h$ is, the higher chance of flood dominance. If $a / h$ is below 0.2, all estuaries tend to be ebb dominant with a M2-M4 velocity phase difference less than 270 degrees. When $a / h$ is between 0.2 and 0.3, the tidal asymmetry depends largely on the ratio of tidal flats to channels $Vs / Vc$. Therefore, if $h$ is increasing faster than $a$ in a flood dominant system under SLR, i.e., $a / h$ decreases, the system is moving to the left, i.e., towards less flood dominance. If $a$ is increasing faster than $h$, the system become more flood dominant. With SLR, $Vs / Vc$ will decrease, i.e., the system is moving down the y-axis, resulting in a more flood dominant situation. We have rephrased the sentence for clarification. Please see Page 2 Lines 11–16.

[Figure]

**Figure R8: Contour plots of the numerical results of 84 model systems as a function of A/H and Vs/Vc: (a) cross-sectionally averaged velocity M4/M2, amplitude ratio; (b) cross-sectionally averaged velocity 2M2 – M4, relative phase. A, tidal amplitude; H, water depth; Vs, volume of intertidal storage; Vc, volume of channels. Source: Friedrichs and Aubrey, 1988.**

  Response (28): "depth" is changed to the plural form. Now on Page 4 Line 2.

- Methods

  o p.3 l.11-12 Add some more information about the model here. Are the models 2D or 3D? And if it is 3D, what is the vertical resolution? Which quantities are considered, which are relevant to density effects (transport of salt, heat)?

  Response (29): The model is 2D barotropic. It has been added to the Methods. Please see Page 4 Line 30.

  o p.3 l.16 Better: 'The MARS domain extends to deep waters and covers the entire North-West European continental shelf...' This makes it more clear that the model not only extends to deep waters along a few sections of the open boundary, but captures the entire shelf edge.)

  Response (30): Thanks for the suggestion. The changes are made. Now on Page 4 Lines 14–16.

  o p. 3 l. 24 reference Slangen 2014 is missing; restructure sentence

  Response (31): The reference is added to the list on Page 17 Lines 26–28. We have rephrased the sentence. Now on Page 4 Lines 23–24.

  o p.4 l.10 Depending on the grain size, the time lag between local suspended sediment concentration and current velocity is not necessarily negligible. With regard to finer fractions (especially silt fractions) settling lag effects are important!

  Response (32): Thanks for the reminder. As mentioned in Response (7), we consider sand here. This part is revised to clarify the application of $Q$ in this study. Please see from Page 5 Line 33 to Page 6 Line 4.

- Results

o  p.4 l.24 The correlation coefficient is not ideal to assess a tidal signal as the tidal wave itself is a very strong signal compared to the error, therefore observed and modeled tidal water elevations always have a relatively strong correlation - also in case of rather low model accuracy. For example, you may add RMSE, which is more appropriate.

Response (33): Thanks for the suggestion. We have shown the Taylor Diagram including correlation coefficients, root mean square deviations, and standard deviations (Fig. 2c).

o  p.5 l.8 "the main tidal patters" …of what? Tidal current velocity? Please clarify.

Response (34): We have removed this sentence in the revised manuscript.

o  p.5 l.11-13 "With SLR, TR increases almost uniformly within ES": You may mention that this statement is related to the investigated SLR range up to 2 m and that this is not necessarily true for larger SLR values (e.g. for SLR > 2 m). (The same for p.5 l.19-21)

Response (35): Agreed. We have the defined the range in which these relationships applies. Please see Page 7 Lines 11–12.

o  p.5 l.14 Please indicate more precisely, which r egion you mean with "adjacent North Sea"? Do you mean only the region directly located seaward from the barrier (up to which depth?) or the entire North Sea section within the GETM model or an even larger domain of the North Sea such as the Southern Bight?

Response (36): Indeed unclear here. We have changed it to "the adjacent North Sea in the GETM domain as well as the Southern North Sea calculated by Idier et al. (2017)". Now on Page 7 Lines 17–18.

o  p.5 l.16-18 I don't get your point. Why do you mention the fixation of the tidal basin size when talking about the role of tidal range for tidal prism? An increased tidal range will always increase the tidal prims, no matter, if the tidal basin is fixed or not. A non-fixation would only further increase the tidal prism. Also keep in mind that as long as intertidal flats are present in the initial case (your base-line scenario) the tidal prism will always increase with SLR, even if tidal range is not increased (remains constant), because with SLR former tidal flat volume is added to the tidal prism.

Response (37): Yes, it is true. This sentence is removed.

o  p.5 l.18 Figure 7 does not show that. It looks like tidal currents are mainly increased on the tidal flats or shoals. How could this be explained?

Response (38): Fig. 7 is now Fig. 9 in the revised manuscript. Fig. 9b shows the velocity difference between scenarios with 1 m and 0 m SLR. The difference is positive at most of the basin, indicating the increase of tidal currents in most areas. We explain it in Section 4.2. The M2 phase difference between the seaward to landward ends is reduced (Fig. 5d), which means that it takes shorter time for the M2 tide to penetrate along the basin. This is a combined result of increased water depth, reduced friction, and basin convergence. Please see from Page 7 Line 33 to Page 8 Line 3.

- o p.6 l.9-10 and Figure 6b: Your analysis is very difficult to justify. You made an average of residual sediment transport over the entire ES. What does that tell you about the exchange between ES and the adjacent North Sea? When the tidal asymmetry is 0, I would expect the residual sediment transport to be minimal as well, if it is related to the exchange between ES and the adjacent North Sea. Furthermore, I would expect the intensity of residual sediment transport at SLR of 0.25 m to be roughly the same as in the baseline scenario (SLR of 0), since the magnitude of tidal asymmetry of current velocity is about the same.

  Response (39): We realize that the quantity $Q$ in our study should not be used as the estimate of sediment budget in the Oosterschelde for reasons given in Response (7). Therefore, we have removed the basin-integrated $Q$ in Fig. 8b, i.e., the previously Fig. 6b. Accordingly, the interpretation of $Q$ is revised in Section 4.3. Please see Page 8 Lines 11–23.

- Discussion and summary

  - o p.6 l.17-18 Do you have any figure as proof of the value of 30 cm for the high water at spring tide? As shown in Figure 3 the increase in TR due to a SLR of 1 m is about 0.3 m. If tidal high water increases by an extra of 0.3 m (1.3 m in total), this means that tidal low water is elevated exactly by SLR (1 m in total)? So does that mean that the increase in tidal range by 0.3 m is solely induced by the increase of tidal high water?

    Response (40): The tidal range increased by 0.33 m for the 1 m SLR scenario, which is the average increase in tidal range. That is, the mean high water is increased and mean low water is decreased by half of 0.33 m, i.e., 0.17 m. Here, we are talking about the high water at spring tide, i.e., the maximum high water, which should be the upper limit of coastal defense. Fig. R9 shows the high water at spring tide in the baseline and 1 m SLR scenarios, and their difference. Excluding the 1 m SLR, the high water is elevated for an extra 20–30 cm (Fig. R9c). We have made the clarification in the text. Please see Page 8 Lines 27–30.

[Figure]

**Figure R9: The high water during the spring tide in the baseline and (b) 1 m SLR scenarios; (c) the difference of high water between the 1 m SLR and baseline scenarios.**

o p.6 l.19-23 "turnover time": Why do you place this totally different aspect here? It is not logically connected to the rest of the text, neither to the preceding nor to the following.

Response (41): Thanks for the comment. Maybe the text is not clear enough. The paragraph is about the impact of SLR on ecosystem functions and management strategies. Prior to this sentence, the impact of increased dike height is mentioned. The second impact is the potential changing ecosystem function. As mentioned in Section 2, the construction of the storm surge barrier almost doubled the flushing time of the basin and affected the exchange with the North Sea and ecosystem

functions (e.g., shellfish culture) of the Oosterschelde. Since the tidal range may shift back to the pre-barrier level (Section 4.2), we are curious to see whether the flushing time can be reversed with SLR. This is why the sentence is placed in the paragraph. Now the paragraph is from Page 8 Line 25 to Page 9 Line 3.

o  p.6 l.24 stronger tidal response [of tidal range] to SLR

Response (42): The suggested revision is made. Now on Page 9 Line 4.

o  p. 7 l. 5 'increases faster than' is unclear

Response (43): We have removed this sentence and updated the discussion in this paragraph (Page 9 Lines 4–24).

o  p.7 l.6-8 I suggest not to generalize findings from this study, because the barrier is a special geometric feature strongly affecting the tidal dynamics in the ES (just as you say it in line 21).

Response (44): Sentence removed.

o  p.7 l.11-15 A tidal basin with extensive intertidal flats actually corresponds to a high ratio of a/h. The relative importance of these two effects depends on the ratio of tidal flat area to channel area within the tidal basin. In your study site (ES) the channel area is much larger than the tidal flat area, suggesting a stronger dependence on the a/h ratio. This could explain why tidal asymmetry shifts towards ebb dominance with SLR.

Response (45): This makes sense. Like mentioned in Response (22), the ratio $a / h$ is important. The ratio $V_s / V_c$ in the Oosterschelde is small in the western part but is large in the east. That is probably the reason of spatial variable tidal asymmetry in the basin. We have expanded the discussion on shifts to ebb dominance in the Oosterschelde under SLR. Please see Page 9 Lines 26–32.

o  p.7 l.27-28 I agree with you, but did you make any comparison to results of the shelf model (MARS)? If not, how can you conclude that shelf models are less applicable?

Response (46): The MARS model does not have a high resolution in the Dutch Delta region. The Oosterschelde is sometimes close (Fig. R10a) and sometimes open (Fig. R10b) to Grevelingen in MARS, while in reality they are isolated by dams and sluices. The water elevation in the Oosterschelde is also not well simulated due to a low spatial resolution. For example, the water elevation in the eastern part is always around 1 m whether at low (Fig. R10a) or high (Fig. R10b) tides. Therefore, a refined local model for the Oosterschelde seems necessary.

[Figure]

**Figure R10: Snapshots of sea surface height simulated by MARS at (a) 2 Jan. 2009 14:15 and (b) 9 May 2009 14:00.**

o p.7 l.31 Are you sure that most studies on this topic neglected this aspect? I guess that most studies actually considered it.

Response (47): This is related to the comment addressed in Response (25). Our study emphasizes that it should be considered and not be neglected.

o p. 8 l. 16 you certainly take into account 'gravitational force', do you mean tide generating forces within the model

Response (48): We have changed it to "gravitational circulation". Now on Page 11
Line 15.

- Figures
  - The number of figures is adequate but the quality needs to be generally improved.
    A coastline and at least some geographic information would help.

    Response (49): We have changed the coordinates to longitude and latitude and
    added the coastline.

  - Specifically: When you state Depth in m, what is the vertical reference system?
    Meter below NAP? IS it the same for regional and shelf model? What is the
    coordinate system you are using? I would prefer coordinates instead of model
    dimensions for the axes.

    Response (50): It is NAP. We used the Cartesian coordinate. Now we have
    changed them to longitude and latitude.

  - Figure 1: Cannot see gauge locations well.

    Response (51): Fig. 1 is replotted.

  - Figure 3: At least some geographic information would be nice, e.g. show
    Vlissingen from Fig. 1

    Response (52): Fig. 3 is replotted with longitude and latitude and Vlissingen is
    marked in the first panel. Now Fig. 4.

  - Figure 4:e) should be amplitude not phase in the

    Response (53): Fig. 4 is replotted and corrections made. Now Fig. 5.

  - Figure 6: references a) and b) missing

    Response (54): It is on the upper right corner. Now Fig. 8.

  - Figure 7: show coastline / land

    Response (55): We have added the coastline. Now Fig. 9.
Thank you for your interesting contribution to the discussions on SLR impact in estuaries. I think you did some decent work in downscaling tidal dynamics and analysing the -potential- impact of SLR on the tidal dynamics. Also you explored the potential impact on sediment transport and tidal flats survival. I think this latter aspect deserves some more clarification and discussion. In the attached document I have made some minor comments. Below I formulate my major concerns. I have no doubt accepting your work when these are adequately addressed.

Response (1): Thank you for the constructive comments on our manuscript. We have revised the manuscript as requested and addressed all the comments as follows.

1) You disregard morphodynamic development. This may be a justified assumption in the sense that the morphodynamics potentially create an extra and yet unclear dimension to the work. It is good to restrict your efforts sometimes. However, I feel that there will be significant morphodynamic analysis coming 100 years in the ES.

Response (2): Thanks for the comment. Yes, we assumed a constant bottom topography mainly because of the uncertainties in the future. If the topography does change with time, the spatial bottom roughness can be altered in contrast to the baseline scenario, which may change the local friction and tides. In addition, the convergence can also be changed. It will require a sediment transport and geomorphology model to study all these effects. An example is that the M2 tide in the German Bight is amplified because of the bathymetric changes (Hagen et al., 2019). We have expanded the discussion of the limitation in the last paragraph (Page 11 Lines 15–27) of the "accept-changes" version of the revised manuscript and pointed out the limitation in the abstract (Page 1 Line 13) and conclusions (Page 12 Lines 1–3). Hereafter, wherever the page and line numbers occur, we refer to the "accept-changes" version.

Response (3): It is true that the mud transport is not included in the study. We have addressed this comment in Response (13).

Drawing strong conclusions based on an analysis that disregards morphodynamics

and fine sediments seems not justified. I suggest to rephrase the summary and conclusions acknowledging more clearly that morphodynamics and fines were not considered. You may add a discussion on why you disregarded these and what important implications of that assumption could mean to your results (like the heightening of tidal flats).

Response (4): We have revised the abstract and added a conclusion section to acknowledge the assumptions and limitations of this study. For example, the exclusion of morphodynamics and fine sediments are mentioned on Page 1 Line 13 and 17, respectively. The reasons of making these assumptions are described in Methods on Page 4 Lines 28–29 and from Page 5 Line 33 to Page 6 Line 4, respectively. The assumptions are discussed on Page 11 Lines 18–24 and Page 8 Lines 20–23, respectively.

* I believe it is also the reduced tidal range (and not only the reduced sediment supply) in the ES that makes the intertidal area to erode. Wave action is more concentrated at a speciifc height (in a smaller tidal range) causing more erosion of the tidal flats.

Response (5): Thanks for the suggestion. Yes, we agree with it. We have made the changes to suggest that the reduced tidal range also contributes to the erosion of sediment in Sections 2 (Page 3 Lines 20–21) and 4.3 (Page 8 Lines 17–20).

* I am interested in how the ES is implemented in the MArs model: can you explain that a little bit more what the assumptions and implications are of the one way coupling? To what extent does the MARs model include the effect of the ES? Are the GETM boundaries far enough at sea to have no effect of the ES dynamics under SLR?

Response (6): The MARS model does not have a high resolution in the Dutch Delta region. The Oosterschelde is sometimes closed (Fig. R1a) and sometimes open (Fig. R1b) to Grevelingen in MARS, while in reality they are isolated by dams and sluices. The water elevation in the Oosterschelde is also not well simulated due to a low spatial resolution. For example, the water elevation in the eastern part is always around 1 m whether at low (Fig. R1a) or high (Fig. R1b) tides. Therefore, a refined local model for the Oosterschelde is clearly necessary.

[Figure]

**Figure R1: Snapshots of sea surface height simulated by MARS at (a) 2 Jan. 2009 14:15 and (b) 9 May 2009 14:00.**

\* I miss conclusions since these are merged in the discussion. Please differentiate into "discussion" and "summary". And maybe add sub-headings in the section that is now called "discussion and summary"

Response (7): We have added a Conclusion section in the end to summarize the major findings. Limitations and major assumptions are also ackowledged.

with kind regards
Mick van der Wegen

Please also note the supplement to this comment:
https://www.ocean-sci-discuss.net/os-2019-50/os-2019-50-RC3-supplement.pdf

Specific comments from the supplement:
Page 1 Lines 11-13: The conclusions are quite bold (even misleading) given that
1) morphodynamic adaptations are not accounted for.
2) you consider coarse sediment only whereas muddy sediment will be relevant as well
I would rephrase the conclusions as more provisional and within the limitations of your
assumptions (your decent work suggests developments under rough assumptions)

Response (8): This comment is the same as is addressed in Response (4). The abstract is
revised as suggested.

Page 1 Line 14: Cross out "model"

Response (9): It is changed to "model coupling". Now on Page 1 Line 18.

Page 2 Line 30: Actually I believe it is also the reduced tidal range in the ES that makes the
intertidal area to erode. Wave action is more concentrated (in a smaller tidal range) causing
more erosion of the tidal flats

Response (10): Thanks for the suggestion. We have made the changes. See Response (5).

Page 3 Line 26: I am interested in how the ES is implemented in the MArs model: can you
explain that a little bit more?

Response (11): This comment is the same as is addressed in Response (6).

Page 3 Line 31: Can you explain a little bit more what the assumptions and implications are
of the one way coupling? To what extent does the MARs model include the effect of the ES?
Are the GETM boundaries far enough at sea to have no effect of the ES dynamics under
SLR?

Response (12): We have added the explanation in the first paragraph of Section 3 (Page 4
Lines 9–11). One-way coupling in our application means the communication from the larger
(MARS) to smaller (GETM) domain is resolved, but not contrariwise. The MARS model
does not include the effect of the ES. It is not necessary to extend the GETM far enough. The
effect of SLR on tides at the open boundary is transferred from MARS to GETM.

Page 4 Line 10: Is this assmption valid? it only considers (coarse) sand while mud
concentrations are considerable. What are the consequences pls elaborate a little bit more.

Response (13): When calculating $Q$, only one class of sand with a specific erosion parameter $\alpha$ and settling velocity $w_s$ is considered. Our study aims to focus on the hydrodynamic effects (velocity and tidal asymmetry) that can be varied by SLR and affect sediment transport. The reason of choosing sand is that the Oosterschelde after the Delta Works is mostly sandy according to our along bottom measurements of sediment grain size distribution (Fig. R6) and a previous study (Fig. R7). That said, $Q$ is only an example of how SLR can change the asymmetry-associated transport of this type of sand. We have specified it where $Q$ is discussed in the revised manuscript (See from Page 5 Line 33 to Page 6 Line 4). It is also clarified in Section 4.3 that the mud transport is unaddressed in this study, and therefore, this study is not a quantification of sediment budget under SLR. Please see Page 8 Lines 20–23.

[Figure]

**Figure R2: The near-bottom sediment grain size distribution at (a) two stations of the Oosterschelde: (b) OS1 measured on 4 June 2019 and (c) OS7 measured on 6 June 2019. The grain size distribution is measured by the LISST-200X Particle Size Analyzer.**

[Figure]

**Figure R3: Fine sediment content < 53 pm of the subtidal bottom of the Oosterschelde after the completion of the storm-surge barrier and compartment dams. Source: Mulder and Louters, 1994.**

Page 7 Line 25: "including" and cross-sectional convergence

Response (25): Suggested changes made. Now on Page 10 Line 19.

[revised manuscript text omitted]

---

## Author Response (AR2)

Topic Editor Decision: Publish subject to technical corrections (04 Feb 2020) by John M. Huthnance

Comments to the Author:

Dear Authors

Thank-you for your extensive revisions which have mostly satisfied the referees. One referee has not responded to the request to re-review but I think you responded reasonably to their comments. You may have seen that the other two are pleased with your response and have only "minor" comments now. Please see below their comments and a few details from myself. You should respond to these; on final publication all these comments will become public and so the quality of your response will be visible. I will treat this last stage as "Technical corrections" meaning that I do not need to see the paper again. However, it will be copy-edited and you should check that the final version retains your intended meaning.
Thank-you for publishing in Ocean Science.

Yours sincerely
John Huthnance

Dear Editor,

We appreciate your efforts in handling our manuscript through two rounds of review processes and suggesting technical corrections below. We are glad that you and the reviewers are satisfied with our revisions. The following comments have been replied and addressed point by point. The "technically corrected" manuscript, both "track-changes" and "accept-changes" versions, is submitted again along with our responses.

Thank you for your help.

Sincerely,
Long Jiang
On behalf of co-authors

Review 1

"Dear Authors;

Thank you for considering my comments and making adjustments accordingly. I think you addressed my questions. I have some minor comments/observations:

- at page 11 lines 13-14 you state: "Another limitation of the study is not considering changes in bed morphology and thus, bottom roughness." I would rephrase this sentence a little bit. Including morphodynamics not only has impact on bed roughness, but also has an effect on tidal propagation (as you rightfully state in the abstract) and output parameters such as intertidal area (your figure 8). Making reference to a poster (Hagen et al., 2019) is quite weak when plenty of peer-reviewed journal publications on models with morphodynamic adaptation are available (and some of which you included already elsewhere). Here I give you some references that include relevant discussions and other references on the matter (some of which are mine, without insisting to refer to these specifically, but merely as inspiration for addressing the issue properly in your work).

Response (1): Thanks a lot for reading our manuscript twice and providing helpful comments. We have rephrased the sentence, to refer to not only bottom roughness but also other hydrodynamic processes in the basin. Please see Page 11 Lines 22–23 of the "accept-changes" version of the revised manuscript.

The suggested references broaden our discussion on the estuarine morphodynamic adaptation with sea-level rise. We have included them in the discussion in this paragraph.

Also, you state in the discussion that you simply do not consider bed level changes, but it would have been nice to find some elaborations in the discussions on the potential impact of bed level changes on your results. You choose not to do that (and it may be challenging), but probably also miss an opportunity to make the work more appealing in that sense.

- Ganju, N. K., & Schoellhamer, D. H. (2010). Decadal-timescale estuarine geomorphic change under future scenarios of climate and sediment supply. Estuaries and Coasts, 33(1), 15-29.
- Elmilady, H. M. S. M. A., van der Wegen, M., Roelvink, D., & Jaffe, B. E. (2019). Intertidal area disappears under sea level rise: 250 years of morphodynamic modeling in San Pablo Bay, California. Journal of Geophysical Research: Earth Surface, 124(1), 38-59.
- Lodder, Q. J., Wang, Z. B., Elias, E. P., van der Spek, A. J., de Looff, H., & Townend, I. H. (2019). Future Response of the Wadden Sea Tidal Basins to Relative Sea-Level rise—An Aggregated Modelling Approach. Water, 11(10), 2198.

Response (2): Thanks for the suggestion. We have expanded the discussion of not considering morphological changes. It is indeed challenging to speculate how bottom topography will change given the uncertainties in sediment sources and grain size distribution. Further, the uncertainties

in bottom topography will translate into how including them can change our results. Although we cannot directly predict the morphologic changes in the Oosterschelde based on studies into other systems, the potential uncertain sources are added to the Discussion to be considered by future studies. Please see Page 11 Line 28–32 of the "accept-changes" version of the revised manuscript.

It would be nice to see some of your explanation of the ES implementation of the MARS model, back in the manuscript.

Response (3): We have explained that the MARS resolution is relatively low in the Oosterschelde and Dutch Delta. This is why the model downscaling is necessary. Please see the first paragraph of Section 3

I am very happy with the nice R1-5 figures in your response!

Response (4): Thanks!

Mick van der Wegen"

Review 2
"I really appreciate the thorough revision of the manuscript. All questions and suggestions have been answered in depth.

A technical note:
Please make sure that the orientation of the x-axis in Figure 6 fits to the description."

Response: Thanks for the positive feedbacks. We have double checked the transect orientation. It is from the west to the east as described in the caption.

Editor (myself)

Page 1 line 15. "basin" -> "bay"?

Response (1): Changed.

Page 1 line 17. You might add that the shift to enhanced sediment export has implications for dredging and/or shoreline/coastal defence management (as you discuss in the paper).

Response (2): We have added "with potential implications for shoreline management"

Page 2 line 23. Omit first "The"

Response (3): "The" is removed. Now on Page 2 Line 24 in the "accept-changes" version of the revised manuscript. All the page and line numbers refer to those in the "accept-changes" version.

Page 2 line 27. Omit "the"

Response (4): "the" is removed. Now on Page 2 Line 28.

Page 3 line 1. ". . sea"

Response (5): "seas" is changed to the single form. Now on Page 3 Line 2.

Page 3 line 5. ". . Additionally, tidal changes . ."

Response (6): "the" is removed. Now on Page 3 Line 6.

Page 3 line 32. "are" -> "is"

Response (7): The suggested change is made. Now on Page 4 Line 1.

Page 4 line 24. Better ". . rate itself does not feature in the model runs). . ."

Response (8): We have moved "itself" forward. Now on Page 4 Line 27.

Page 4 line 29. "or" -> "and".

Response (9): We have replaced "or" with "and". Now on Page 4 Line 32.

Page 5 line 17. ". . analyses were carried out both on observed and on simulated . ."

Response (10): The suggested change is made. Now on Page 5 Line 20.

Page 9 line 33. Omit "such".

Response (11): "such" is removed. Now on Page 10 Line 4.

Page 10 lines 7, 8. I think you need to say how the Ria de Aveiro lagoon and Western Scheldt convergence / friction (or ??) are different from the previous examples so that their "tidal asymmetry is insensitive to SLR . . to deepening".

Response (12): The sentence has been rephrased and moved to between Page 9 Line 33 and Page 10 Line 2.

Page 10 line 34. ". . coupling, and would make an . ."

Response (13): We have revised the sentence as suggested. Now on Page 11 Line 3.

Page 11 lines 7,8. ". . parts of Chesapeake Bay (Lee. . .) and San Francisco Bay . . ."

Response (14): "Bay" is added to the sentence. Now on Page 11 Line 11.

Page 11 line 19. Omit first "The".

Response (15): "The" is removed. Now on Page 11 Line 23.

Page 13 Acknowledgements. I think you might thank all the referees for helpful suggestions improving the paper.

Response (16): We have included the names of three reviewers in the Acknowledgements.